# Engineering natural microbiomes toward enhanced bioremediation by microbiome modeling

Zhepu Ruan [1,2,5], Kai Chen[1,5], Weimiao Cao[1], Lei Meng[1], Bingang Yang[1], Mengjun Xu[1], Youwen Xing[1], Pengfa Li [1], Shiri Freilich[3], Chen Chen[1], Yanzheng Gao [4] ✉, Jiandong Jiang [1] ✉ & Xihui Xu [1] ✉

Engineering natural microbiomes for biotechnological applications remains challenging, as metabolic interactions within microbiomes are largely unknown, and practical principles and tools for microbiome engineering are still lacking. Here, we present a combinatory top-down and bottom-up framework to engineer natural microbiomes for the construction of function-enhanced synthetic microbiomes. We show that application of herbicide and herbicide-degrader inoculation drives a convergent succession of different natural microbiomes toward functional microbiomes (e.g., enhanced bioremediation of herbicide-contaminated soils). We develop a metabolic modeling pipeline, SuperCC, that can be used to document metabolic interactions within microbiomes and to simulate the performances of different microbiomes. Using SuperCC, we construct bioremediation-enhanced synthetic microbiomes based on 18 keystone species identified from natural microbiomes. Our results highlight the importance of metabolic interactions in shaping microbiome functions and provide practical guidance for engineering natural microbiomes.

Microbiomes are ubiquitous in nature and play important roles in almost all biogeochemical cycles occurring on this planet, such as the metabolism of nutrients[1–3], agriculture[4,5], food fermentation[6,7], element cycling[8,9], biofuels[10–12], and pollutant degradation[13–15]. Synthetic microbiomes, which are based on interacting relationships within microbiomes, can perform more complicated tasks with higher efficiency compared to single strains and natural microbiomes, showing promising applications in industry, health, and the environment[16–19]. These synthetic microbiomes provide a new strategy to realize the re-establishment of complex metabolic functions by combining the metabolic capacities of multiple strains, which will help to overcome the limitations of the metabolic capacity of a single strain. In addition, synthetic microbiomes provide a viable option for sharing unwanted metabolic burdens among strains in a community[20]. Although many efforts have been made to construct synthetic strains[21–23], research on synthetic microbiomes is still in its infancy, and many unknowns and challenges have emerged during the engineering of microbiomes. To date, practically applicable principles and tools for natural microbiome engineering are still lacking[24].

Bottom-up and top-down strategies have been proposed to engineer microbiomes[24]. The former involves artificially designing synthetic microbiomes by strain combination based on the

[1]Department of Microbiology, College of Life Sciences, Nanjing Agricultural University, Key Laboratory of Agricultural and Environmental Microbiology, Ministry of Agriculture and Rural Affairs, Nanjing 210095, China. [2]Guangdong Laboratory for Lingnan Modern Agriculture, Guangdong Provincial Key Laboratory of Agricultural & Rural Pollution Abatement and Environmental Safety, College of Natural Resources and Environment, South China Agricultural University, Guangzhou 510642, China. [3]Newe Ya'ar Research Center, Agricultural Research Organization, P.O. Box 1021, Ramat Yishay 30095, Israel. [4]College of Resources and Environmental Sciences, Nanjing Agricultural University, Nanjing 210095, China. [5]These authors contributed equally: Zhepu Ruan, Kai Chen. ✉e-mail: gaoyanzheng@njau.edu.cn; jiang_jjd@njau.edu.cn; xuxihui@njau.edu.cn

understanding of different single strains[25]. The limitation of this strategy is that the targeted strains are usually chosen by experience and intuition based on historical knowledge rather than selecting strains with fundamental benefits obtained from natural microbiomes, which might neglect naturally occurring microbial interactions. In contrast, the top-down strategy starts from natural microbiomes and then carefully optimizes the complex natural microbiomes to display desired functions[26–28]. Although this strategy is easier to carry out, it is less controllable and easily misses information on microbial metabolic interactions. In addition, functional microbiomes obtained by the top-down strategy are usually not simple enough for the application, and the largely unknown metabolic interactions among keystone species of the functional microbiomes hinder their further simplification. Therefore, obtaining a simplified functional microbiome from natural microbiomes and capturing complex interspecies interactions within the functional microbiomes are essential for natural microbiome engineering.

Recently, progress in sequencing technologies has greatly promoted the description and understanding of natural microbiomes in response to environmental perturbations at both the taxonomic and functional levels[29–31], laying the foundations for the identification of keystone species. However, limited information on metabolic activities and interactions among keystones could be inferred by sequencing. Until now, experimental determination of metabolic interactions in a microbiome has remained a major challenge, even for simple consortia with two members[32]. Advances in computational tools such as genome-scale metabolic models (GSMMs) and their simulation algorithms enable the in silico analysis of the metabolic activities of a strain and the interspecies interactions in microbiomes[32–36]. Here, we developed a new microbiome modeling framework called Super Community Combinations (SuperCC) to simulate the performances of different microbiomes. Different from most available multistrain metabolic modeling frameworks that are inclined to simulate cooperative interactions in simple microbial consortia[22,23], SuperCC is focused on comparing the performances of different microbiomes that cover all combinations of sets of given strains. In addition, SuperCC adapts to both syntrophic and competitive consortia and has no limitations of numbers in simulated microbiomes, which is suitable for both simple and complex microbiomes. Furthermore, SuperCC can also provide a new strategy for the computational design of a synthetic cell by learning microbiomes based on metabolic interactions. Such synthetic cells mimicking the functions of synthetic microbiomes could provide a solution for synthetic microbiomes containing plant or animal pathogens as keystones that are not suitable for practical application.

Environmental organic pollutants, including pesticides, pharmaceuticals, industrial chemicals, and many others, have become a severe worldwide problem urgently requiring solutions[37–40]. For example, bromoxynil octanoate (BO), a systemic herbicide, has been increasingly used in the past few years to replace atrazine for postemergence control of annual broadleaved weeds[41]. BO is highly toxic to fish and aquatic invertebrates[42–44] and moderately toxic to earthworms[45]. BO and its metabolic intermediate (3,5-dibromo-4-hydroxybenzoate, DBHB) have been detected in many environmental samples[46,47], and their removal from polluted sites is crucial for environmental safety. The metabolic activities of microbiomes rather than single strains are essential for bioremediation[48]. It should be noted that the compositions of microbiomes and the interactions between their members affect both the growth and degradation efficiency of microbiomes[49,50]. However, microbiomes in nature are usually not functional or have low efficiency for the biodegradation of pollutants.

Here, we present a framework to engineer natural microbiomes and construct synthetic microbiomes based on a balanced combination of top-down and bottom-up strategies to optimize the performance of herbicide biodegradation (Fig. 1). We started with constructing a functional microbiome with enhanced bioremediation efficiency. Then we screened keystones in the functional microbiome to construct a simplified microbiome, which was used to substitute for the complex functional microbiome. Next, we used SuperCC to simulate the performances of the simplified microbiomes with different combinations of keystones. The simulation predicted not only the optimized combination of strains but also microbial metabolic interactions. We subsequently mixed isolated strains to create an optimized synthetic microbiome for testing. Our results provide important insights into microbiome engineering, and the framework of synthetic microbiome/cell construction has a wide range of applications, from the bioremediation of contamination to the biosynthesis of industrial products.

## Results
### Constructing the effective functional microbiome for bioremediation

To test whether herbicide application and/or herbicide-degrader inoculation could efficiently produce a functional microbiome with improved pollutant-biodegrading capability from natural microbiomes, different initial microbiomes collected from three totally different soils were used, including red (pH = 5.0), yellow cinnamon (pH = 7.3) and purple soils (pH = 8.1) (Fig. 1 and Supplementary Fig. 1). We also used two different types of pollutants, including a complex pollutant (here, BO) that can only be degraded by a synergistic consortium (*Pseudoxanthomonas* sp. X-1[51] and *Comamonas* sp. 7D-2[48,52]) and a simple pollutant (here, DBHB) that can be degraded by different single strains (*Comamonas* sp. 7D-2 or *Pigmentiphaga* sp. H8[53]). Correspondingly, three different inocula, including a single strain (H8 or 7D-2), a synergistic consortium (X-1 and 7D-2), and a competitive consortium (H8 and 7D-2), were used.

The initial microbiomes were treated with (1) herbicide (BO or DBHB) application; (2) microbial consortium (combination of strains 7D-2 and X-1, 7D-2&X-1) inoculation; (3) the combination of herbicide application and microbial consortium inoculation (BO&7D-2&X-1 or DBHB&7D-2&H8); and (4) the combination of herbicide application and single-strain inoculation (DBHB&7D-2 or DBHB&H8). To obtain an effective inoculation strategy, we compared the performances of single and repeated inoculation, as well as low- and high-dose of inoculation (Supplementary Fig. 2 and Supplementary Fig. 3). Treatments with high-dose of inoculation repeatedly had a more remarkable influence on bacterial community and showed much higher BO-degrading ability compared to other treatments. Therefore, we used the strategy of repeated inoculation of degrading strains with high doses to decrease the experimental time. The detailed study design is described in the Methods. All treatments markedly improved the degradation efficiency of both BO and DBHB, although the three initial microbiomes showed differences in degradation efficiency (Fig. 2 and Supplementary Fig. 4). Specifically, inoculation of the synergistic consortium (BO&7D-2&X-1 or 7D-2&X-1) for degradation of complex pollutants (here, BO, nondegradable by each single strain of 7D-2 or X-1) was effective. In addition, inoculation with the functional competitive consortium (DBHB&7D-2&H8) was more effective than inoculation with a single functional strain (DBHB&7D-2 or DBHB&H8). These results showed inoculation of degrading consortia was feasible for both simple and complex pollutants. Together, we showed herbicide application and herbicide-degrader inoculation were an efficient top-down method to obtain a functional microbiome with enhanced pollutant-biodegrading capability.

### Reassembly of natural microbiome at the taxonomic and functional levels

To explore the dynamic process of microbiome reassembly driven by herbicide and inoculum applications, we investigated changes in microbiomes over a period of 30 days after different treatments. The $\alpha$-diversity analysis revealed significant decreases in diversity

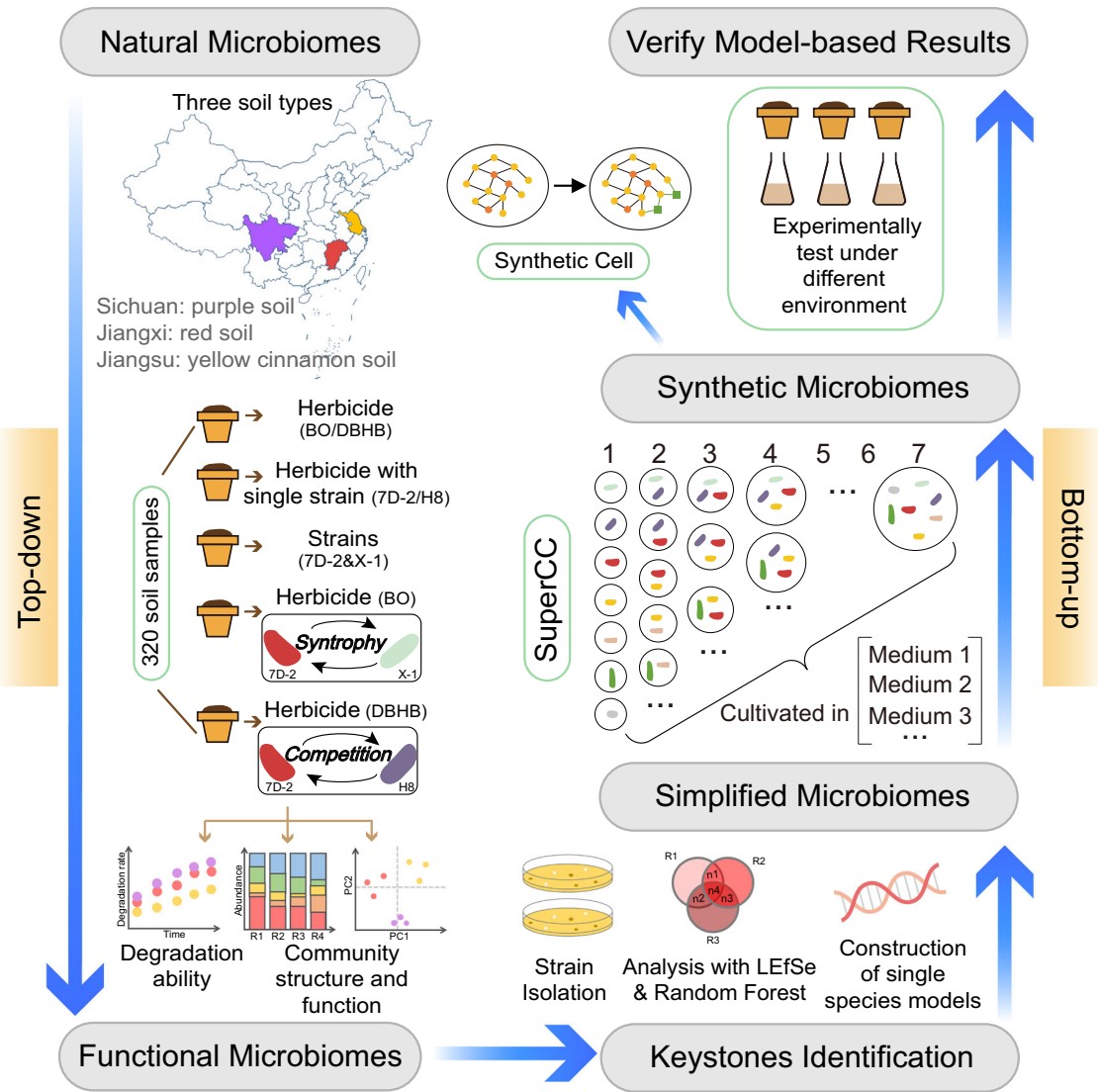

**Fig. 1 | Experimental design and general scheme of synthetic microbiome construction.** The workflow of synthetic microbiome construction is based on a balanced combination of top-down and bottom-up strategies. The workflow starts from engineering natural microbiomes by herbicide application and/or herbicide-degrader inoculation to obtain the functional microbiomes. In the top-down phase, three different soils were treated with alternative combinations of two herbicides (BO and DBHB, representing complex and simple pollutants, respectively) and three kinds of inoculants (single-strain and synergistic or competitive consortia), resulting in 320 soil samples for testing degradation capability and tracing dynamic microbiome successions. Then, potential keystone species were identified by exploring strain abundance shifts combined with strain isolation. The keystones were then used to construct simplified microbiomes to substitute for the complex functional microbiomes. In the bottom-up phase, a newly developed microbiome modeling framework, SuperCC, was used to simulate the performances of simplified microbiomes with different combinations of keystones to optimize the combination of keystones. The synthetic microbiome with an optimal keystone combination was used for further testing and application. Beyond the microbiome simulation, a new computational strategy for synthetic cell design based on learning metabolic interactions of synthetic microbiomes was also provided by SuperCC. BO, bromoxynil octanoate; DBHB, 3,5-dibromo-4-hydroxybenzoate.

(illustrated by a decreased Shannon index) over time during treatments in all soils (Fig. 3a, Supplementary Fig. 5). In nonmetric multidimensional scaling (NMDS) analysis, the initial microbiomes from different soils were separated from each other, revealing great differences among them (Fig. 3b, c). The microbiomes of the herbicide treatment were clustered together with their corresponding initial microbiomes, indicating that slight changes occurred in bacterial composition between the initial and herbicide treatment microbiomes. However, inoculation treatments caused greater changes in microbiomes. More importantly, the Day 30 samples from the inoculation treatments (except for strain H8) were clustered closer than the Day 0 or Day 3 samples (Fig. 3b), and the distances among inoculation treatment (rather than herbicide treatment) microbiomes from different soils decreased over time (Fig. 3d). These results showed a convergent succession of microbiomes at the taxonomic level in the

three soils caused by inoculation, resulting in a more similar bacterial composition in the functional microbiomes.

Metagenomic analyses were performed to gain insights into the functional differences among treated microbiomes (Fig. 4). The Day 30 and Day 0 samples from the BO&7D-2&X-1 treatment were selected to represent the treated and initial microbiomes, respectively. In principal component analyses (PCA) of the functional profiles, the treated microbiomes were clustered together and separated from the initial microbiomes (Fig. 4a). Enzymes involved in the BO biodegradation pathways, including the nitrilase, nitrile hydratase, and nitrile hydroxylation pathways, were traced based on their functional profiles (Fig. 4b). The initial microbiomes had a relatively low abundance of genes encoding BO-degrading enzymes, which was consistent with the low efficiency of BO degradation in the initial soils (Fig. 4c). The treatment increased the abundance of degrading genes, contributing to the enhanced degrading

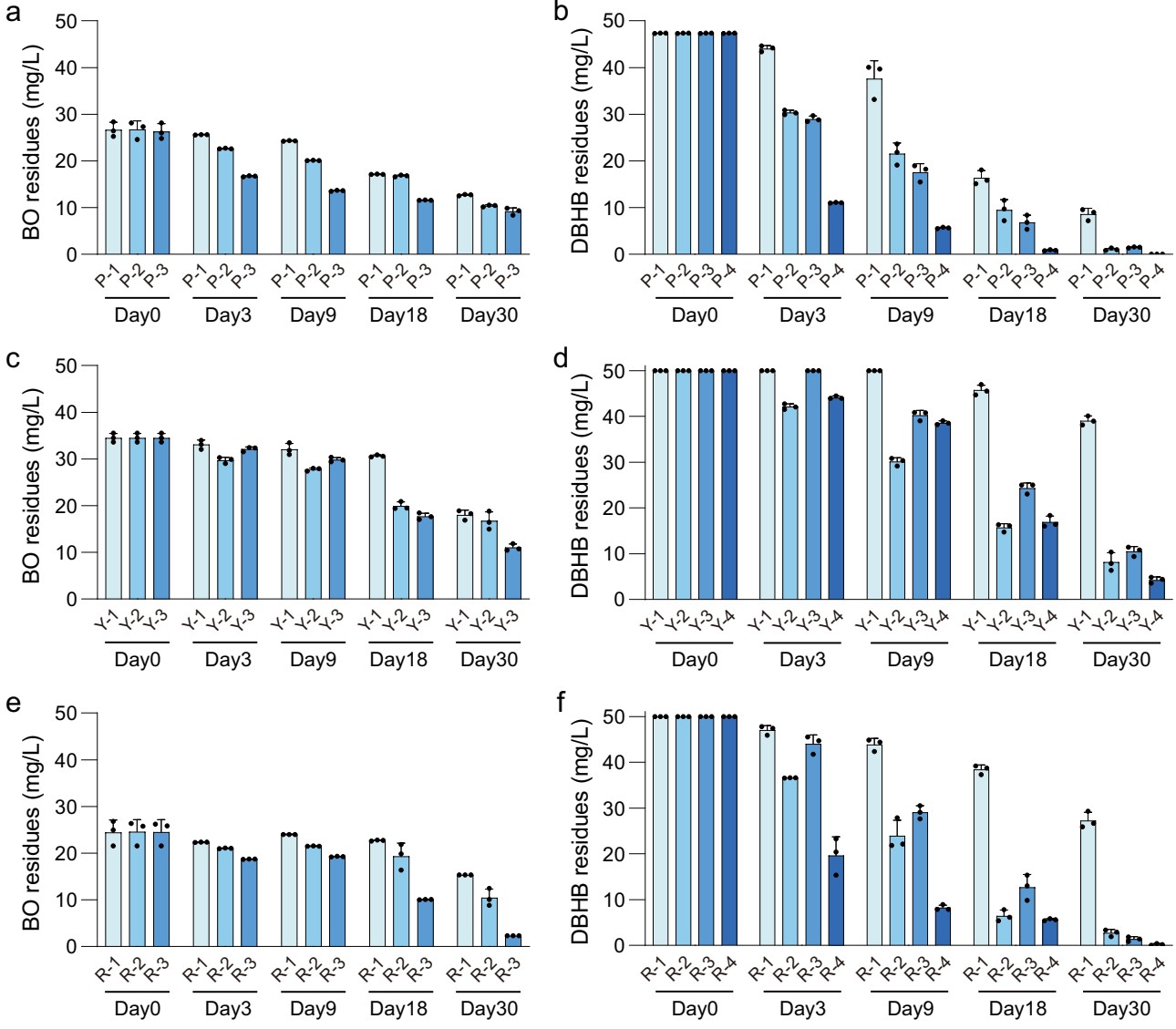

**Fig. 2 | Enhanced degradation efficiency of pollutants by microbiomes driven by herbicide application and bacterial degrader inoculation. a, b** Purple soil (P); **c, d** Yellow cinnamon soil (Y); **e, f** Red soil (R). The degradation efficiency was analyzed by detecting residues of BO (**a, c, e**) or DBHB (**b, d, f**) in MM media supplemented with BO or DBHB (50 mg/L) after degradation by 0.5 g of treated soils for 10 hours. For BO degradation, 1, 2, and 3 represent treatments with BO, the synergistic consortium (7D-2&X-1), and the combination of BO and the synergistic consortium (BO&7D-2&X-1), respectively. For DBHB degradation, 1, 2, 3, and 4 represent treatments with DBHB, DBHB combined with strain 7D-2 (DBHB&7D-2), DBHB combined with strain H8 (DBHB&H8), and DBHB combined with the competitive consortium (DBHB&7D-2&H8), respectively. BO bromoxynil octanoate, DBHB 3,5-dibromo-4-hydroxybenzoate. The data are presented as mean values ± SD (n = 3 biological independent replicates). Source data are provided as a Source Data file.

capability (Fig. 4c), which was consistent with the increased relative abundance of 7D-2 in treated microbiome (Supplementary Fig. 6). Notably, 52%-100% of nitrilase (the key enzyme for BO degradation) in the treated microbiome was from 7D-2, indicating most of the degradation was driven by the inoculated bacteria, especially for the yellow cinnamon soil (Supplementary Fig. 7). Then, we classified these degrading genes taxonomically at the phylum, order, class, and genus levels (Fig. 4d and Supplementary Fig. 8). The dominant phyla containing the degrading genes were similar in the three soils and included *Proteobacteria* and *Actinobacteria* (Fig. 4d). These results demonstrated the convergent succession in microbiomes at the functional level.

## Identification of keystone species for simplifying the functional microbiome

The taxonomic and functional convergence of microbiomes pointed out that specific species in microbiomes that were affected by treatments could functionally modify the microbiome activity for

enhanced degrading efficiency. The acquisition of these specific species (called keystone species) provided an easy way to construct a simplified functional microbiome. To identify keystone species that were affected by treatments, we screened genera and ASVs whose abundances were differentiated along with treatments. LEfSe analysis (LDA > 3.0) revealed a total of 133, 54, and 62 bacterial genera with different abundances in the late phase in BO&7D-2&X-1 compared to the early phase in purple, yellow cinnamon, and red soils, respectively (Fig. 5a). Among them, 27, 6, and 17 genera with significant change in abundance were shared across treatments for the three respective soils. Meanwhile, 67, 71, and 35 genera showed different abundances in the DBHB&7D-2&H8 treatments of the three soils, respectively (Supplementary Fig. 9). However, only a few of these identified genera showed increased abundances over time. For example, only 11 (purple), 11 (red), and 40 (yellow cinnamon) genera were enriched in the BO&7D-2&X-1 treatment, and 11 (purple), 8 (red), and 12 (yellow cinnamon) were enriched in the

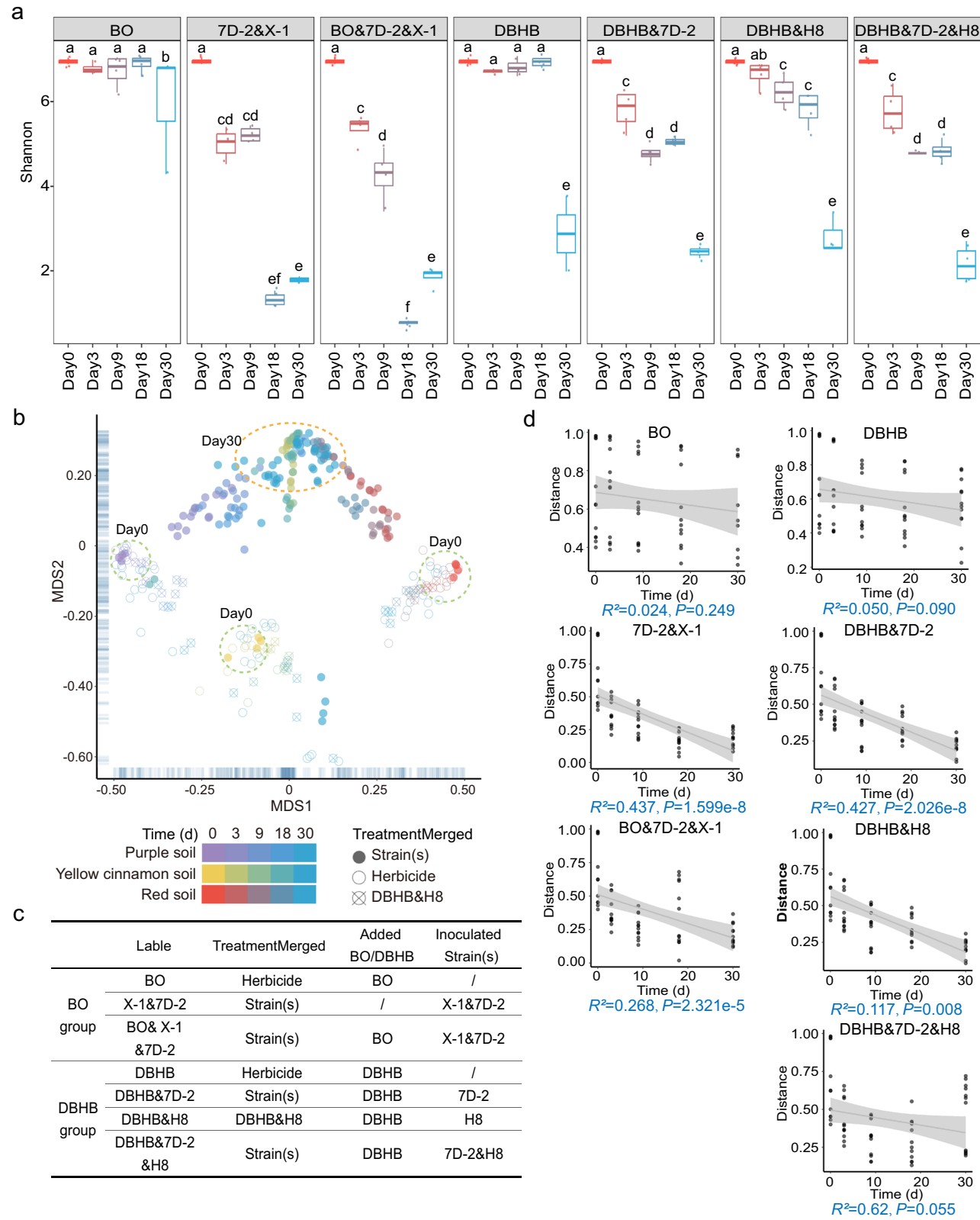

DBHB&7D-2&H8 treatment (Supplementary Fig. 10). More than 40% of these significantly changed genera were same among the three different soils with the same treatments, especially for yellow cinnamon soils (>65%, Fig. 5b), indicating that genera with significant changes in abundance were common among different soils with the same treatments. Specifically, *Bacillus* and *Sphingobacterium* showed significantly increased abundances in all three soils in the BO groups. These results are consistent with the convergent succession of different microbiomes.

In parallel to bacterial community analysis, we carried out independent isolation of BO and DBHB degraders in original and treated soils. In total, 290 typical isolates were isolated and selected for subsequent 16S rRNA gene sequencing to classify their taxonomy and investigate their degradation abilities (Supplementary Table 1). Among

**Fig. 3 | Alterations in microbiome diversity. a** α-diversity levels of microbiomes from purple soils. The α-diversity levels of microbiomes from red and yellow cinnamon soils are shown in Supplementary Fig. 5. The boxplots show the Shannon indices of soils with different treatments ($n = 4$ biological independent replicates). The letters above the boxplots show significant differences between samples at $P < 0.05$ (one-way ANOVA with correction by Tukey's HSD test). The horizontal bars within the boxes represent the medians. The tops and bottoms of the boxes represent the 75th and 25th quartiles, respectively. BO, bromoxynil octanoate; DBHB, 3,5-dibromo-4-hydroxybenzoate. **b** Nonmetric multidimensional scaling (NMDS) plots of beta diversity (Bray–Cutis dissimilarity) by time and treatment. Samples are shaped and color-coded according to the treatments and times. The details of treatments are provided in panel c. **c** Treatments used in the study. **d** Bray–Curtis distances among the microbiomes from different soils at each time point (Days 0-30) decreased with treatment time for the inoculation treatments rather than herbicide treatments. Linear regression line is indicated by the grey line. The 95% confidence interval of the linear regression line is indicated by gray bands. $P$ values are two-sided. Source data are provided as a Source Data file.

them, 133 isolates belonged to the significantly enriched genera identified by LEfSe analysis. Traditionally, these isolates could be used to construct simplified microbiomes to substitute for complex functional microbiomes through experience/intuition or trial-and-error experiments[54–56]. To reduce the time and cost for simplified microbiome construction, community metabolic modeling was used to model community functions and simulate the performances of alternative community combinations. By modeling, the optimization of microbial composition and environmental conditions was predicted, and metabolic interactions that improved community performance were documented. To this end, we used random forest analysis to identify the key ASVs in the significantly changed genera identified by LEfSe. A total of 18 specific key species for modeling were identified through phylogenetic analysis of isolated strains and key ASVs, including *Comamonas, Pseudoxanthomonas, Pigmentiphaga, Pseudarthrobacter, Sphingobacterium, Bacillus, Sphingomonas, Lysinibacillus, Streptomyces, Arthrobacter, Aliihoeflea, Sinomonas, Bradyrhizobium, Acinetobacter, Nocardioides, Achromobacter* and *Pseudomonas* (Fig. 5c, Supplementary Fig. 11, and Supplementary Table 2). GSMMs were constructed for each of the 18 keystones and manually curated (Supplementary Table 3, Supplementary Data 1).

## Interspecies interactions in the inoculated synergistic and competitive consortia

To determine detailed metabolic interactions between strains in the inoculated consortium, including the synergistic (7D-2&X-1) and competitive consortia (7D-2&H8), a two-strain community model was constructed and analyzed (Fig. 6, Supplementary Fig. 12). By simulation, the strain X-1 only transformed BO to bromoxynil, thus could not grow using BO or bromoxynil as the sole carbon source; the strain 7D-2 could degrade bromoxynil, but it was unable to degrade BO into bromoxynil (Fig. 6a). Therefore, the strain 7D-2 could not grow using BO as the sole carbon source, but it could grow using bromoxynil as the sole carbon source (Fig. 6b, Supplementary Fig. 13). However, the synergistic consortium of X-1 and 7D-2 could degrade BO completely by metabolic cooperation, and both strains grew well using BO as the sole carbon source (Fig. 6b, Supplementary Fig. 13). These predictions were supported by experimental validations (Fig. 6c-g). For example, the predictions of no growth of X-1 or 7D-2 for single-culture but growth of both strains for co-cultured were verified by the growth experiment (Fig. 6c). Both strains 7D-2 and H8 could degrade DBHB and grew using DBHB as the sole carbon source (Supplementary Fig. 12). However, the combination of 7D-2 and H8 did not improve the DBHB-degrading efficiency or biomass of the consortium (Supplementary Fig. 12).

For 7D-2&X-1, we predicted mutual exchange fluxes between strains 7D-2 and X-1 (Fig. 6b). Simulations predicted that strain X-1 absorbed BO and secreted bromoxynil, hypoxanthine, D-glucosamine, and L-proline that were consumed by strain 7D-2. In return, strain 7D-2 secreted xanthine, D-mannose, $NH_4^+$, and L-glutamate that were utilized by strain X-1, maintaining the growth of strain X-1 (Fig. 6b). Interestingly, for DBHB degradation, although the two strains H8 and 7D-2 were competitive for DBHB, mutual exchange fluxes were predicted in the process of cogrowth (Supplementary Fig. 12). Strain H8 consumed DBHB and $NH_4^+$ to maintain its growth and secreted fumarate, L-proline, D-glucosamine, D-mannose, and hypoxanthine that were utilized by strain 7D-2 (Supplementary Fig. 12). In return, strain 7D-2 secreted succinate and L-glutamate that were consumed by strain H8 (Supplementary Fig. 12).

The full map of predicted metabolic interactions between 7D-2&X-1 is detailed in Supplementary Fig. 14, describing the metabolic routes leading to production and consumption of the exchange metabolites specified in Fig. 6b. The thermodynamic analysis of the predicted metabolic interaction confirmed its thermodynamic feasibility (Supplementary Table 4, Supplementary Table 5). To verify the predicted exchange fluxes, we used liquid chromatography–mass spectrometry (LC–MS) to detect the exchanged metabolites in co-cultures of these two strains (Supplementary Fig. 14). All the exchanged metabolites were successfully detected in LC–MS. In addition, we tested the growth and BO/bromoxynil/DBHB degradation in monocultures of the three species (X-1, 7D-2, and H8) grown on minimal media, each supplemented by the relevant exchange metabolites. The growth and degradation were enhanced in the supplemented medium for strains X-1, 7D-2, and H8 (Supplementary Fig. 15), which were consistent with predictions. In addition, we detected the exchanged metabolites in monocultures of the three species and compared to those in the co-cultures to test the source of the exchanged metabolites. For 7D-2&X-1, the hypoxanthine was secreted by X-1 (Supplementary Fig. 16); for 7D-2&H8, succinate and L-glutamate were secreted by 7D-2 (Supplementary Fig. 16).

To further validate the validity of the predicted metabolic pathways underlying the mutualism between strains X-1 and 7D-2, gene expression profiling of the two strains was compared in single-cultures versus co-cultures (Supplementary Fig. 14, Supplementary Table 6). The expression levels of most genes encoding enzymes that participate in the synthesis of the secreted metabolites were up-regulated in co-culture compared to single-culture. These results indicate that the strains 7D-2 and X-1 possess all the enzymes required for the metabolic interactions and genes coding these enzymes were expressed during metabolic interactions. Therefore, the results of transcriptomic profiling confirmed the predicted metabolic interaction.

To explore the effect of BO concentration on the relative abundance of strains in the synergistic consortium, we simulated the optimal biomass of strains 7D-2 and X-1 co-cultured under different BO concentrations (Supplementary Fig. 17). With decreasing BO concentration, the biomasses of these two strains decreased correspondingly, but the biomass ratio between strains 7D-2 and X-1 increased sharply (Supplementary Fig. 17). The prediction was experimentally supported: after reaching the maximum biomass, the biomass ratio of the two strains gradually increased over time, along with the decreased BO concentration (Supplementary Fig. 17). The prediction was also consistent with the biomass ratio of strain 7D-2 and strain X-1 in soils in treatment BO-7D-2&X-1, as revealed by 16 S rRNA gene amplicon sequencing (Supplementary Fig. 17). The enhanced soil degradation ability over time resulted in a decrease in the BO concentration in soils, and the ratio of the two strains increased correspondingly over time.

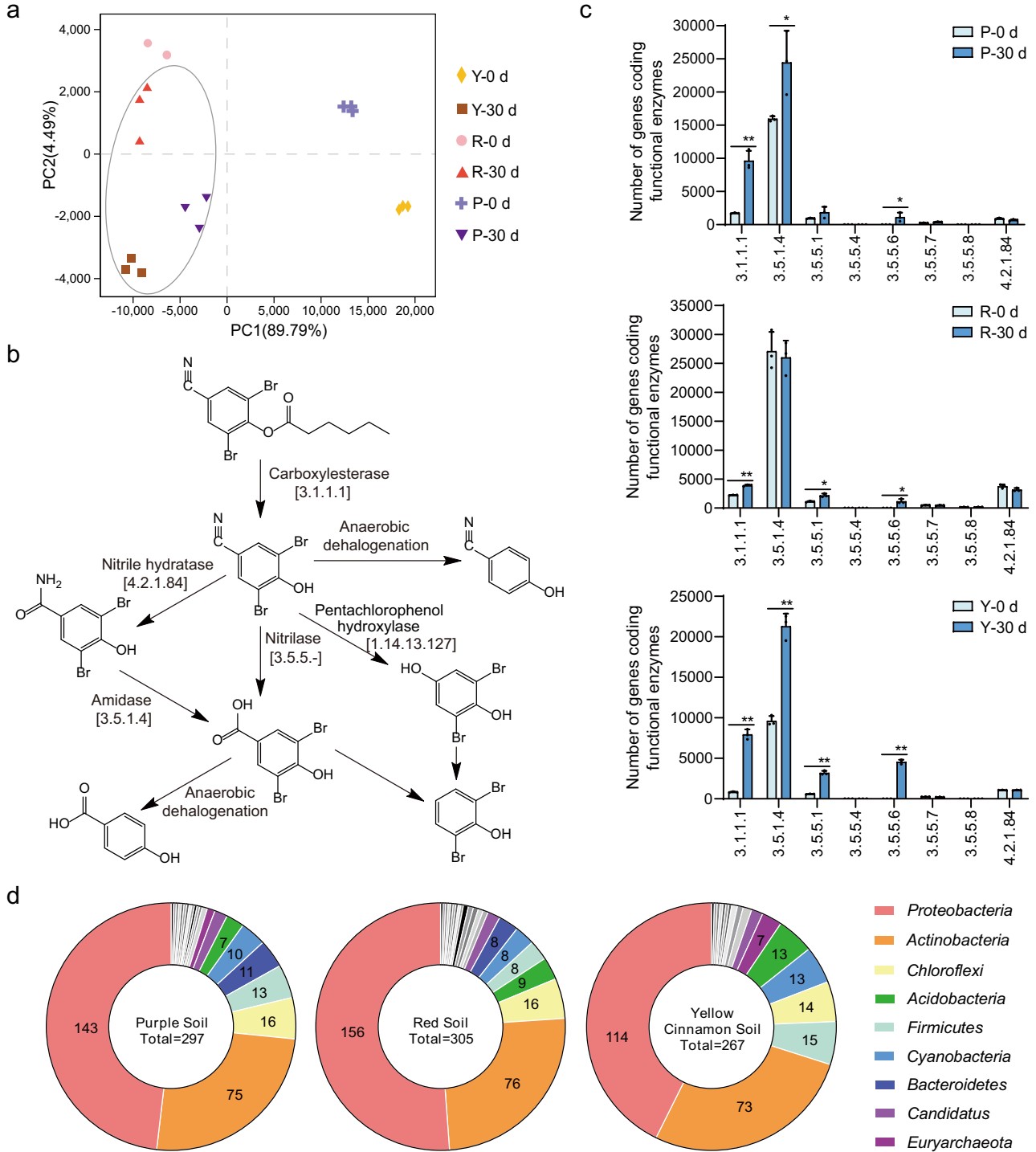

**Fig. 4 | Functional variation in microbiomes. a** Principal component analysis (PCA) of all the metagenomes based on the KEGG module abundance. P, purple soil; Y, yellow cinnamon soil; R, red soil. 0 and 30 d, samples collected at Days 0 and 30 from treatments of BO&X-1&7D-2, respectively. **b** Metabolic pathways of BO and DBHB in the microbiomes. The pathways were reconstructed by functional gene annotation. **c** Abundances of functional genes involved in BO degradation as shown in (b). The data are presented as mean values ± SD ($n = 3$ biological independent replicates). The significance of differences was assessed using a two-sided Student's t-test (**$P < 0.01$; *$P < 0.05$). **d** Taxonomic classification of the functional genes involved in BO degradation as shown in **b**. BO, bromoxynil octanoate; DBHB, 3,5-dibromo-4-hydroxybenzoate. Source data are provided as a Source Data file.

## Predicting performances of combinations of different keystones and characterization of metabolic interactions among keystones with SuperCC

For the simulation of complex microbiomes, we developed a modeling framework, SuperCC, to predict metabolic flux distributions in microbiomes under different nutritional conditions. SuperCC was scalable to a large number of species suitable for both simple and complex microbiomes. To compare the performances of combinations of different keystones, we constructed a set of compartmented community models representing all possible combinations of keystones by SuperCC. Growth simulations were carried out in minimal mineral medium (MM medium) containing BO as the sole nitrogen and carbon source (BO medium) and BO medium supplemented with different nitrogen and carbon sources, including glucose, $NH_4^+$, and $NO_3^-$

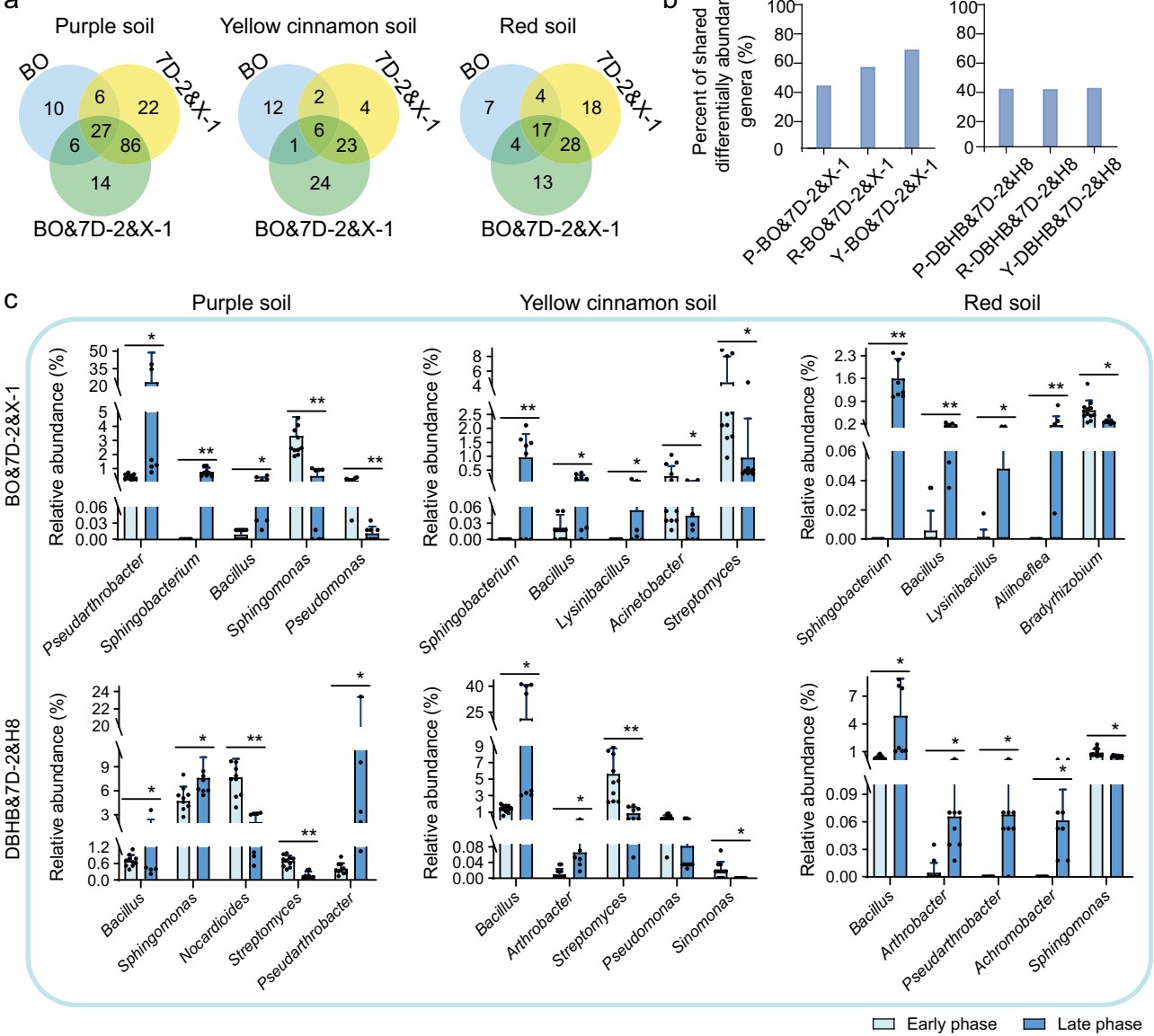

**Fig. 5 | Similarity of abundance shifts in microbiomes driven by demonstration and relative abundances of keystone genera. a** Venn diagram of differentially abundant genera identified by LEfSe analysis, showing that most genera with abundance shifts were shared among inoculation treatments. **b** The percentages of differentially abundant genera shared among different soils. The differentially abundant genera in treatments with a combination of herbicide and inoculation, including BO&7D-2&X-1 and DBHB&7D-2&H8, are shown. P, purple soil; Y, yellow cinnamon soil; R, red soil. BO, bromoxynil octanoate; DBHB, 3,5-dibromo-4-hydroxybenzoate. **c** Relative abundances of keystone genera identified by a combination of LEfSe and random forest analysis. The average abundances of each keystone genus in samples at Days 0-9 (early phase) and Days 18-30 (late phase) are shown. The data are presented as mean values ± SD ($n = 12$ biological independent replicates for early phase; $n = 8$ biological independent replicates for late phase). The significance of differences was assessed using a two-sided Student's t-test (**$P < 0.01$; *$P < 0.05$). Source data are provided as a Source Data file.

(Fig. 7a). With SuperCC, we simulated the performances of communities from the three soils to predict combinations that could enhance community growth. In BO medium, the combination of two strains, including LM5 (for red and yellow cinnamon soils) and P56 (for all three soils), with 7D-2&X-1 exhibited enhanced community growth compared to 7D-2&X-1 only (Fig. 7a). The addition of NH$_4^+$ (BO-NH$_4^+$ medium) promoted community growth, and the combination of AC6&LM5 (for yellow cinnamon soil) with 7D2&X-1 showed the maximal biomass, while strains B2 (for yellow cinnamon soil) and Y13 (purple soil) also promoted the growth of communities. For BO medium supplemented with NO$_3^-$ (BO-NO$_3^-$ medium), no growth promotion was detected except for strain BR1, suggesting that those strains could not assimilate NO$_3^-$ except for strain BR1. Although no growth promotion by the addition of glucose (BO-G medium) was

detected compared to BO medium, the BO medium supplemented with both glucose and NH$_4^+$ (BO-G-NH$_4^+$ medium) markedly improved the community growth, suggesting that a nitrogen source could be an enhancer for BO degradation.

The above predictions were verified experimentally in soils with the three-species consortium (Fig. 7b). We used the combination of 7D-2&X-1 with *Escherichia coli* as a negative control. The combination of 7D-2&X-1 with P56, LM5, B2, or AC6 showed the highest degradation rates in yellow cinnamon soils (>80%), and the combination of 7D-2&X-1 with P56, LM5, or BR1 showed the highest degradation rates in red soils (>80%), consistent with the simulations. In purple soils, we did not detect any strains that could improve the degradation rate by combination, which may be due to the original high degradation rate (>80%) achieved by 7D-2&X-1. The strains E44 and A8 did not show improvement in any

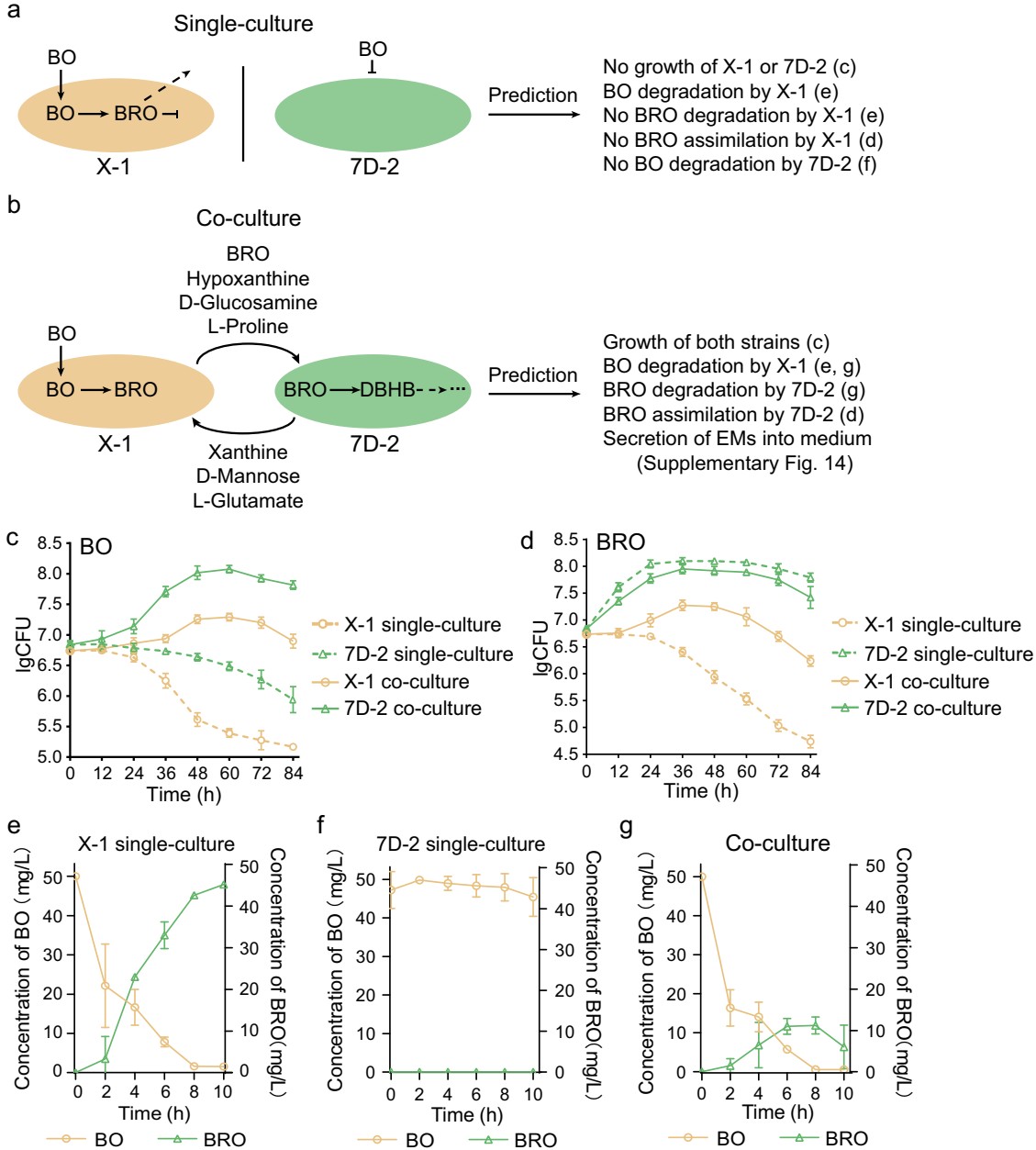

**Fig. 6 | Simulations and experimental validations of metabolic interactions in two-member consortia.** Predicted metabolic interactions between strains 7D-2 and X-1 in single-culture (**a**) and co-culture (**b**) by community modeling. Ten testable predictions were provided for experimental verification of the simulations. The letters in parentheses following the predictions refers to the panels **c**–**g** supporting the corresponding predictions. BO bromoxynil octanoate, DBHB 3,5-dibromo-4-hydroxybenzoate; BRO bromoxynil. **c, d** Comparison of cell growth of 7D-2 and X-1 growing separately versus in co-culture. The log transformed colony-forming unit (CFU) of each strain cultured in medium with BO (**b**) or bromoxynil (**c**) as the sole carbon source is shown. The data are presented as mean values ± SD ($n = 3$ biological independent replicates). **e-g** The degradation ability of BO by X-1 (**e**), 7D-2 (**f**) and co-cultures (**g**). The concentrations of BO and bromoxynil produced during BO degradation in the medium with BO as the sole carbon source are showed. The data are presented as mean values ± SD ($n = 3$ biological independent replicates). Source data are provided as a Source Data file.

soils, which also agreed with the predictions. We also tested growth promotion by adding glucose and/or $NH_4^+$ to MM medium (Fig. 7c). The results showed a marked enhancement of community growth with additional carbon or nitrogen sources, which was also consistent with the prediction. Similar simulations were conducted for communities from DBHB-treated groups. Again, the predictions were consistent with the experimental results (Supplementary Fig. 18-20).

To better understand the metabolic interactions among these keystones and inoculated strains, we predicted exchange fluxes in communities (Supplementary Fig. 21). Exchanges of small molecules such as amino acids and hypoxanthine among strains were detected.

Particular attention was given to strain BR1, which was predicted to utilize $NO_3^-$ and secrete $NH_4^+$ to strain 7D-2 (Supplementary Fig. 21). Through experimentation, we verified that strain BR1 could utilize $NO_3^-$, while strain 7D-2 could not (Fig. 7d). Meanwhile, the combination of strains 7D-2 and BR1 showed much higher biomass than the sum of the biomass of each strain grown separately (Fig. 7d). Similar to the verification for two-strain community models (7D-2&X-1 and 7D-2&H8), we detected the exchanged metabolites in co-cultures of 7D-2&X-1&BR1 to test the perdition of three-strain model (Supplementary Table 7). All the 9 exchanged metabolites were successfully detected in LC−MS (except for 2-oxoglutarate). Then we compared the exchanged

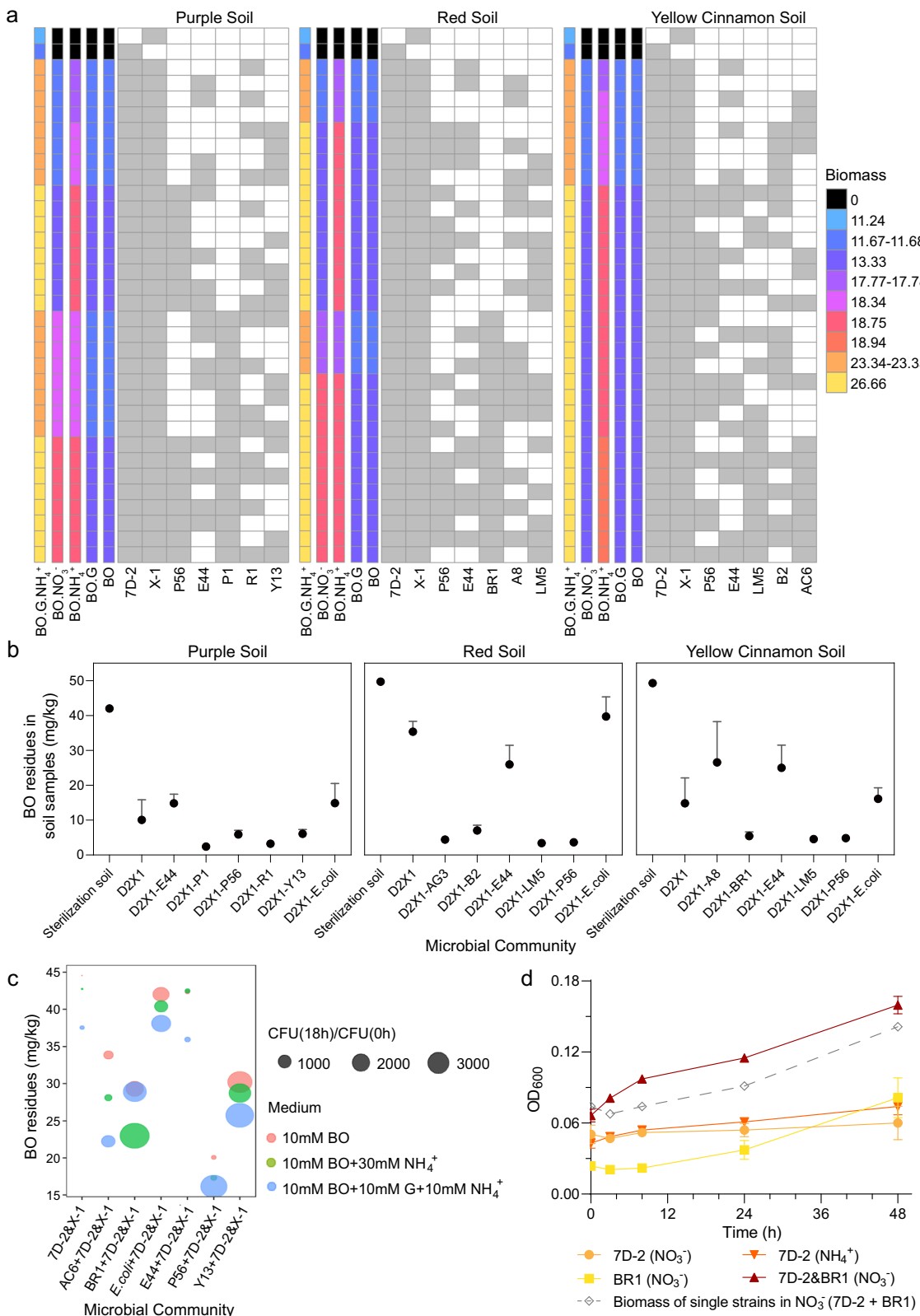

metabolites in monocultures with the co-cultures to test the source of the exchanged metabolites for 7D-2&X-1&BR1. The results showed that fumarate, 4-hydroxybenzoate, succinate, L-lysine, and (R)−3-hydroxybutanoate were exchanged metabolites between 7D-2 and BR1, while hypoxanthine, L-glutamate, and xanthine were between 7D-2 and X-1, which were consistent with the predictions (Supplementary Table 7). Finally, we used DNA stable isotope probing (SIP) combined with

amplicon sequencing to detect the strains that involved in the degradation of BO indicated by the assimilation of $^{13}$C carbon. We used $^{13}$C-labled 4-hydroxybenzoic acid (the intermediate metabolite for BO degradation), as no $^{13}$C-labled BO could be purchased. In the co-cultures of 7 strains (including 7D-2, X-1, P56, E44, LM5, B2, and E. coli), 4 strains (7D-2, X-1, LM5, and B2) were involved in the assimilation of $^{13}$C carbon while 2 strains (E44 and E. coli) were not (Supplementary

**Fig. 7 | Simulations and experimental validations of the performances of different microbiomes. a** Predicted biomass (1/h) of keystone combinations. The gray/white cells in the grids indicate species included/not included in the bacterial combination, respectively. The bars on the left of the grids are indicative of biomass predicted in five media: medium that contains BO as the sole carbon and nitrogen source (BO), BO medium supplemented with glucose (BO.G), BO medium supplemented with $NH_4^+$ (BO.$NH_4^+$), BO medium supplemented with $NO_3^-$ (BO.$NO_3^-$), and BO medium supplemented with glucose and $NH_4^+$ (BO.G.$NH_4^+$). BO, bromoxynil octanoate. **b** Experimental validation of simulations of BO treatments by pot experiments with three-member consortia in three soils separately. All three-member consortia included strains 7D-2 and X-1 together with one selected keystone. The consortia, including strains 7D-2 and X-1 and an exogenous strain (*E. coli*), were used as negative controls. The data are presented as mean values ± SD

($n = 3$ biological independent replicates). **c** Experimental validations of BO degradation and bacterial growth performances in the three media. The dot color represents the medium, and the size represents the growth of the synthetic consortia. **d** Experimental validation of predictions that strain BR1 improves the growth of strain 7D-2 by using $NO_3^-$. Strain 7D-2 could grow in a medium with $NH_4^+$ as the sole nitrogen source ($NH_4^+$) but not in a medium with $NO_3^-$ as the sole nitrogen source ($NO_3^-$). Strain BR1 could use $NO_3^-$ as the sole nitrogen source for growth. The gray line refers to the sum of the biomass of strains 7D-2 and BR1 in $NO_3^-$ medium separately, which is much lower than the biomass of the co-culture of strains 7D-2 and BR1 in the same medium, demonstrating that the co-culture of strains 7D-2 and BR1 improves the growth of both strains in the $NO_3^-$ medium. The data are presented as mean values ± SD ($n = 3$ biological independent replicates). Source data are provided as a Source Data file.

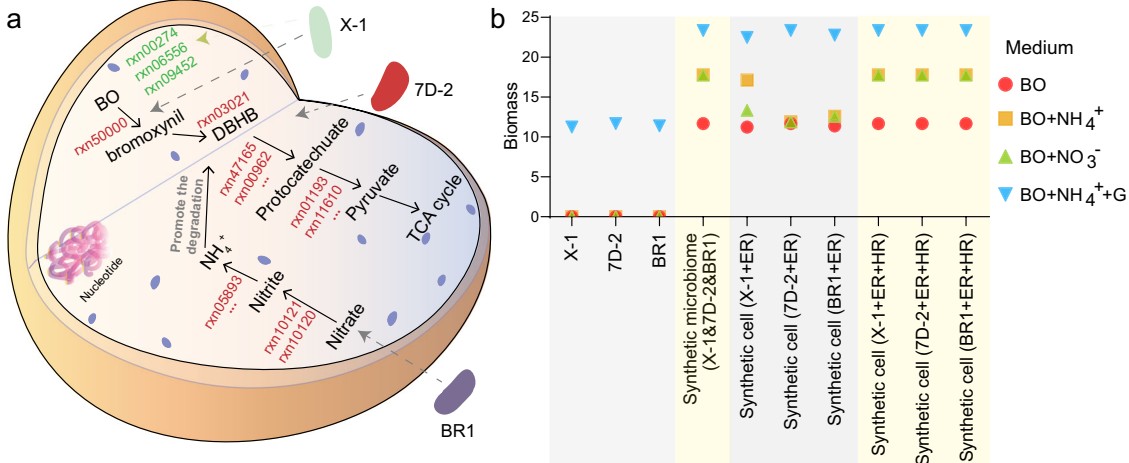

**Fig. 8 | Synthetic cells designed computationally by SuperCC based on learning metabolic networks of functional microbiomes. a** A predicted synthetic cell with key reactions from strains X-1, 7D-2, and BR1 identified by SuperCC, mimicking the metabolic functions of the microbiome of X-1&7D-2&BR1. The key reactions, including essential (red, ER) and helpful reactions (green, HR), are shown. The detailed reaction information is shown in Supplementary Table 8. BO, bromoxynil octanoate; DBHB, 3,5-dibromo-4-hydroxybenzoate. **b** Comparison of the performances of strains (X-1, 7D-2, and BR1), the synthetic microbiome (X-1&7D-2&BR1), and different synthetic cells. X-1 + ER, 7D-2 + ER, and BR1 + ER represent synthetic cells based on chassis cells of X-1, 7D-2, and BR1, respectively. The essential

reactions from the other two strains were added to the chassis cell. For example, X-1 + ER means adding ER from 7D-2 and BR1 to X-1. The above three synthetic cells recovered the function of the synthetic microbiome in a medium containing BO as the sole carbon and nitrogen source (BO) and BO medium supplemented with glucose and $NH_4^+$ (BO + G + $NH_4^+$) but failed in BO medium supplemented with $NH_4^+$ (BO + $NH_4^+$) or $NO_3^-$ (BO + $NO_3^-$). Three additional reactions (HR) helpful for the growth of synthetic cells in BO + $NH_4^+$ or BO + $NO_3^-$ medium were identified in strain X-1, which enabled the synthetic cells to recover the performance of the synthetic microbiome. Source data are provided as a Source Data file.

Fig. 22). The results were agreed with the perditions (except for P56) that strains LM5 and B2 could improve the BO degradation by 7D-2&X-1 while E44 could not. No assimilation of $^{13}C$ carbon was detected for strain P56, which might result from the insufficient amount of $^{13}C$ carbon used in the study and/or the competition among strains P56, LM5, and B2.

### Computational cell design based on mimicking the metabolic network of functional microbiomes

Based on metabolic interactions in the simplified functional microbiomes revealed by SuperCC, we put forward the concept of computational design of synthetic cells by learning functional microbiomes by identifying and adding key metabolic reactions that promote degradation and biomass in the functional microbiomes to a target cell. The computationally designed synthetic cell could achieve the target strain from non-degradable to degradable for biodegradation, from nonsynthetic to synthetic for bioproduction, or from low to high efficiency (Fig. 8a). With SuperCC, 10 reactions were determined to be essential for BO degradation for the 7D-2&X-1 consortium (Supplementary Table 8). By adding the essential reactions to strain 7D-2 or X-1, both computational synthetic 7D-2 and X-1 cells could degrade BO (Fig. 8b). In the 7D-2&X-1&BR1 consortium, three essential reactions for $NO_3^-$ utilization were identified, and

addition of the reactions to the above computational synthetic cells (7D-2 or X-1) helped the synthetic cell utilize $NO_3^-$ (Fig. 8b). Interestingly, the synthetic cells substituting the 7D-2&X-1&BR1 consortium with essential reactions were able to recover the function of the consortium in BO medium rather than in DBHB-$NH_4^+$ or BO-$NO_3^-$ medium. We further identified another three reactions in strain X-1 that enabled the synthetic cells to recover the function of the consortium in DBHB-$NH_4^+$ or BO-$NO_3^-$ medium (Fig. 8b). Although the three reactions were not essential for biomass production, they were helpful for utilizing nitrogen resources. The results also showed the advantages of synthetic microbiomes in the utilization of nitrogen sources based on complex metabolic interactions in microbiomes.

### Discussion

In recent decades, the rapidly increasing number of microbiome studies has greatly improved our understanding of human health and diseases, agricultural production, and environmental remediation[57–59], showing the promising application potential of microbiomes. However, the synthetic microbiome, the basis of taking microbiomes from discovery to application, remains challenging, as the mechanisms underlying microbiome successions and the complex metabolic interactions in microbiomes are still largely unknown. Recent advances in systems biology provide effective methods to understand the

various physiological processes and interactions of microbial strains[60], providing the possibility for the optimal design of synthetic microbiomes. Here, we first showed the reassembly of different microbiomes in response to herbicide and inoculum applications, laying a foundation for functional microbiome construction from a natural microbiome. We then presented a framework to construct synthetic microbiomes based on functional microbiomes and modeling technologies. The framework not only obtained target organisms from natural microbiomes based on microbial interdependences in the natural environments but also captured information on the metabolic interactions in the synthetic microbiome.

The reassembly of natural microbiomes driven by herbicide and inoculation treatments is the basic assumption of the top-down phase in our frameworks to construct a functional microbiome. We showed that these treatments strongly shaped the metabolic function of different natural microbiomes toward enhancing their pollutant-degrading efficiency. Similarly, many other studies have reported nutrition- or host-driven reassembly of natural microbial communities[61–63], indicating that reassembly might be a common feature of microbiome evolution under certain selective pressures. We also showed that treatments with high-dose of inoculation had more remarkable influence on bacterial community compared to those with low-dose inoculation, indicating the effects of inoculation on natural microbiomes might be dose-dependent. Actually, for most bioaugmetation treatments, a final concentration of $10^8$ CFU/g soil is usually used, and the high dose of inoculation causes significant influences on natural microbiomes[64–66]. With decreased concentration ($10^7$ CFU/g soil), the inoculation causes temporary impacts[67], and the inoculation is not the main factor influencing the bacterial community structure compared to the herbicide application[68]. The low-dose of inoculation ($10^6$ CFU/g soil) shows a relatively weak impact on the soil bacterial community[69]. Notably, the initial microbiome from yellow cinnamon soils did not have BO or DBHB degradation ability, but functional microbiomes with high pollutant-degrading efficiency were still obtained. The results demonstrated the feasibility of the bioremediation of new contaminated soils with indigenous microbiomes lacking targeted pollutant-degrading capabilities.

The functional microbiomes directly obtained by top-down strategy are usually not suitable for further applications, as there are generally many unknowns in the functional microbiomes, such as taxa compositions and their interactions in the microbiome[70,71]. These uncertain factors usually result in unstable microbiome structures and/or metabolic efficiency. For example, microbiomes with a better or worse atrazine-degrading efficiency could be derived from the same original microbiome with atrazine-degrading ability in different environments[72]. Thus, it is necessary to construct a simplified microbiome with a known composition and metabolic interactions substituted for a complex functional microbiome. Traditionally, strain isolations followed by trial-and-error experiments were used to construct simplified microbiomes[23,24]. However, the strategy was time-consuming and could miss metabolic information in the functional microbiome. Here, we used sequencing technologies to obtain information on the compositional shifts in microbiomes associated with functional modification for keystone selections. Meanwhile, microbiome modeling was used to characterize the performances of different keystone combinations under various nutritional conditions to optimize simplified functional microbiomes. We showed that this bottom-up pipeline enables the construction of an optimal combination of keystones consisting of both degraders and helper strains. More importantly, these strains were almost all in situ bacteria isolated from soils, giving a stronger environmental adaptabity to the simplified microbiome.

Based on our framework, eighteen species, including three inoculated strains, were selected as potential keystones for simplified functional microbiome construction based on abundance shifts as well as strain isolation. Not surprisingly, the combination of all keystones was not the most cost-effective and degrading-efficient option, possibly because each strain in the combination is resource-consuming. The newly developed SuperCC aimed to quickly establish optimal and simplified combinations and capture the metabolite exchange of combinations in various media. By simulation and experimental validation, four species, *Bacillus* sp. (P56), *Lysinibacillus* sp. (LM5), *Acinetobacter* sp. (AC6), and *Bradyrhizobium* sp. (BR1), were identified to improve the growth of inoculated consortia, thus enhancing metabolic efficiency. The other potential keystones that were not functional showed that the abundance shift driven by demonstration might not be directly involved in functional modification. One possible explanation was the microbiome interactions that led to the abundance shift. Simulations predicted that additional $NH_4^+$ could promote microbiome growth. Furthermore, strain BR1 was predicted to utilize $NO_3^-$, while strain 7D-2 could not, and the combination of strain BR1 with degraders could help the microbiome assimilate $NO_3^-$. These results suggest that $NH_4^+$ or these predicted strains could be used as biostimulation agents.

Currently, the view that antagonistic interactions are ecologically more important than synergistic interactions (such as mutualism) in microbial communities is widely held[73]. However, the function of microbial mutualism in natural environments might be underestimated as metabolic exchanges are difficult to assess within natural systems[73,74]. In the present study, metabolic exchanges were detected between herbicide-degraders and keystone strains in soils by metabolic modeling, showing the application potential of metabolic modeling in exploring mutualism in natural environments. Recently, metabolic modeling has been increasingly used to explore microbial metabolic interactions in both free-living and host-associated natural communities[75–77]. These studies showed metabolic exchanges are ubiquitous in natural microbial communities[73], which is consistent with our results. Besides, we showed the metabolic exchanges enhanced pollutant-biodegrading capability of microbial communities. Similar results have been detected, showing that cross-feeding in microbial communities not only improves survival but also promotes pollutant degradation[78–80].

Exploring and validating metabolic interactions is a challenging task. We developed a modeling tool to predict metabolic interactions and provided a series of testable predictions for experimental validation. These include: (1) the predicted exchanged metabolites could be detected in the medium of co-culture; (2) the strain growth could be improved by the exchanged metabolites in single-culture; (3) SIP experiments could be employed to validate the assimilation of the exchanged metabolites; 4) transcriptome profiling could be used to test the expression of genes required for metabolic reactions. The modeling tool combined with the validation strategy could greatly facilitate the application of metabolic modeling.

Until now, the dominant strategy for microbe application has been synthetic cells designed for specific functions, although the synthetic microbiome has shown promising application prospects[81–84]. The typical workflow to engineer a microbial strain has a number of common steps and requires a large number of decisions on how to improve strain behavior, which are mostly based on a trial-and-error approach[81]. In this study, we provide a new strategy for computational strain design based on mimicking the metabolic network of microbial communities, which could pave the way for efficient strain design workflows to achieve synthetic strains with the capabilities of functional microbial communities. Techniques for genome editing (such as CRISPR[85,86]) and assembly[87] and synthesis of DNA sequences[88,89] are expected to enable the construction of complex synthetic biological systems.

## Methods

### Experimental design and soil sample collection

Three different kinds of soils, including yellow cinnamon, purple, and red soils (0-20 cm soil layer), were collected in June 2018 from cropping fields located in Nanjing (Jiangsu Province; 32° 01′ N, 118° 51′ E), Mianyang (Sichuan Province; 32° 01′ N, 105° 24′ E), and Yingtan (Jiangxi Province; 28° 12′ N, 116° 55′ E), respectively (Fig. 1). Each 1.5 kg of soil sample was ground, passed through a 0.84-mm mesh sieve and placed into a plastic pot. The soil was kept at 40-60% water-holding capacity (WHC). Soil samples were treated by BO or DBHB with four repeats. In the BO group, the microcosm treatments were set as follows: i) BO treatment (BO): adding 2.5 mL of methanol containing 3000 mg/L of BO (the final concentration of BO was 5 mg/kg soil, close to the concentration of field application); ii) inoculation treatment (X-1&7D-2): inoculating two strains X-1 and 7D-2 at a ratio of 1:1, with a final concentration of approximately $2 \times 10^8$ CFU/g soil for each strain, and adding 2.5 mL of methanol unified with other treatments; and iii) bioaugmentation treatment by X-1 and 7D-2 (BO&X-1&7D-2): adding 5 mg/kg of BO and inoculating both strains X-1 and 7D-2 with the same final concentration to treatment ii. Similarly, the microcosm treatments of the DBHB group were iv) DBHB treatment (DBHB): adding 5 mg/kg of DBHB dissolved in 2.5 mL methanol; v) bioaugmentation treatment by 7D-2 (DBHB&7D-2): adding 5 mg/kg of DBHB and inoculating the strain 7D-2 with a final concentration of approximately $2 \times 10^8$ CFU/g soil; (vi) bioaugmentation treatment by H8 (DBHB&H8): adding 5 mg/kg of DBHB and inoculating the strain H8 with same final concentration to treatment v; and (vii) bioaugmentation treatment by 7D-2 and H8 (DBHB&7D-2&H8): adding 5 mg/kg DBHB and inoculating two strains (7D-2&H8, 1:1) with the final concentration to treatment v. The herbicides and inoculating strains were added repeatedly every three days a total of 10 times. Before each repeated addition, 0.5 g of soil was collected to assess the degrading ability of BO or DBHB with high-performance liquid chromatography (HPLC).

The large-scale pot experiments were conducted indoors in November 2018 in Nanjing, China. Soil samples of Days 0, 3, 9, 18, and 30 from each type of soil and treatment were collected and frozen at −80 °C until DNA extraction.

### HPLC analysis

To detect the degrading ability of BO or DBHB by microbial communities with different treatments, 0.5 g of soil was collected from different samples and transferred into fresh MM media containing BO or DBHB (50 mg/L). Following incubating on a shaker at 30 °C for 10 h, 1 mL of the liquid medium was extracted for the detection of residual BO, bromoxynil, or DBHB. BO was extracted from the medium by shaking with an equal volume of dichloromethane for 5 min. The extract (1 mL) was then dried over anhydrous $Na_2SO_4$ and evaporated using a vacuum rotary evaporator at room temperature. The residue was redissolved in 1 mL of methanol and analyzed using a Thermo Scientific Dionex UltiMate 3000 Rapid Separation LC (RSLC) system (Germering, Germany) equipped with a Syncronis $C_{18}$ reversed-phase column (4.6 mm × 250 mm, 5 μm particle size). The mobile phase was pure methanol and the flow rate was 1.0 mL/min. BO was detected at 231 nm, and the BO concentration was determined from the peak area ratio relative to individual standard calibration curves. The column was maintained at 30 °C, and the injection volume was set to 20 μL. Under these conditions, BO exhibited a retention time of 4.8 minutes. For analysis of bromoxynil and DBHB, 1 mL of the medium was centrifuged at 16,000 g for 1 min. The supernatant was then analyzed by HPLC. The mobile phase was acetonitrile/water/acetic acid (50/49.5/0.5, v/v/v), and the flow rate was 1.0 mL/min. Bromoxynil and DBHB were detected at 221 nm and 250 nm, respectively. Concentrations were determined from peak area ratios relative to individual standard calibration curves. The retention times for bromoxynil and DBHB were 9.2 min and 4.8 min, respectively.

### DNA extraction and microbial community sequencing

A total of 348 soil samples were analyzed by 16S rRNA gene amplicon sequencing. Microbial DNA was extracted using the EZNA Soil DNA Kit (Omega Bio-tek, Norcross, GA, USA) according to manufacturer's protocols. The V3-V4 region of the bacteria 16S ribosomal RNA gene was amplified by PCR using the 341F/806R primer set (341F: 5′-CCTAYGGGRBGCASCAG-3′, 806R: 5′-GGACTACNNGGGTATCTAAT-3′). The DNA product was used to construct Illumina Pair-End library following Illumina's genomic DNA library preparation procedure. Then the amplicon library was sequenced on an Illumina Miseq PE250 platform (Shanghai BIOZERON Biotech. Co., Ltd, Shanghai, China) according to the standard protocols. A total of 28 soil samples were found to be contaminated or low in sequencing quality and were deleted, leaving 320 samples for further analysis. After filtration, the sequences were dereplicated and subjected to the DADA2 algorithm[90] to identify indel mutations and substitutions by QIIME2[91]. The phylogenetic affiliation of each 16S rRNA gene sequence (here called amplicon sequence variants, ASVs) was analyzed by the Silva (SSU132) 16S rRNA database using a confidence threshold of 70%[92,93]. In addition to taxonomic composition analysis, we also performed a metagenomic analysis to gain insights into the functional differences between untreated and treated microbiomes with bioaugmentation. A total of 18 soil samples (soil samples from Days 0 and 30 from BO&X-1&7D-2 with three repeats) were selected for metagenomic analysis.

### Statistical analyses and keystone selection

Rarefaction analysis based on Mothur (v1.21.1)[94] was conducted to reveal the Shannon index. $\beta$-diversity was calculated using the "vegan" package (v2.5-7). The microbiome function based on gene profiles from the metagenome was ordinated by PCA using unweighted UniFrac distance with KEGG modules. For each treatment, LEfSe[95] was used to explore the most discriminating genus between early (samples in Days 0, 3, and 9) and late phases (samples in Days 18 and 30). A random forest approach was also used to identify marker ASVs discriminating treatment times using the randomForest package (v4.6-14) in R (v4.0.3). In the random forest model, 80% of the data were used as the training set, 20% were used as the test set, and 100,000 trees were constructed. To reduce the deviation caused by a single run, each model was run 20 times. Keystones of BO&X-1&7D-2 and DBHB&7D-2&H8 were first selected by LEfSe analysis at the genus level, whose abundance in soils was significantly increased or decreased by treatments. Then, a random forest classifier was used to identify the top 30 important ASVs at the species level. Furthermore, the soil strains were isolated and identified to obtain the keystone strains. In total, 290 strains were isolated from the in situ soils by dilution separation methods on Luria-Bertani (LB) agar, and phylogenetic analysis of 16S rRNA genes of the isolates was performed. The ASVs from LEfSe and random forest analysis with isolated strains in soils were used as specific keystones for further synthetic microbiome construction.

Data were analyzed using GraphPad Prism version 8.0 (GraphPad Software, Inc., La Jolla, CA, USA). Graphs of the microbiome data were created using the "ggplot2" (v3.3.0), "pheatmap" (v1.0.12) and "VennDiagram" (v1.6.20) packages.

### Metagenomic sequencing

The DNA was fragmented to an average size of about 400 bp using Covaris M220 (Gene Company Limited, China) for paired-end library construction. The NEXTFLEX Rapid DNA-Seq kit (Bioo Scientific, Austin, TX, USA) was used to construct the paired-end library, where adapters containing the full complement of sequencing primer hybridization sites were ligated to the blunt ends of the fragments. Paired-end sequencing was performed on an Illumina NovaSeq 6000 platform at Majorbio Bio-Pharm Technology Co., Ltd. (Shanghai, China) according to the manufacturer's instructions. Metagenomics data were assembled using MEGAHIT (v1.1.2)[96]. Open reading frames

(ORFs) were predicted from each assembled contig using Prodigal/MetaGene[97] with a minimum length of 100 bp. A non-redundant gene catalog was constructed with CD-HIT (v4.6.1)[98] at 90% sequence identity and coverage. High-quality reads were aligned to this non-redundant gene catalogs to calculate gene abundance with 95% identity using SOAPaligner (v2.21)[99]. Representative sequences from the gene catalog were then annotated for taxonomy using Diamond (v0.8.35)[100] against the non-redundant (NR at NCBI) database with an e-value cutoff of 1e⁻⁵. Further annotations included clustering of orthologous groups of proteins (COG) using Diamond (v0.8.35)[100] against the eggNOG[101] database and KEGG annotations performed against the KEGG[102] database, both with the same e-value cutoff of 1e⁻⁵.

## Whole genome sequencing

The genomic DNA was extracted using a Bacteria DNA Kit (OMEGA) according to the manufacturer's instructions. Whole-genome sequencing of strains X-1, 7D-2, H8, and AT5 was conducted using the Illumina NovaSeq 6000 platform (Shanghai BIOZERON Biotech. Co., Ltd, Shanghai, China). Quality control of the raw paired-end reads was performed using Trimmomatic (v0.36)[103]. Genome assembly was carried out with Unicycler (v0.4.8)[104] using default parameters. Ab initio gene prediction was performed, and gene models were identified using GeneMark[105]. Functional annotation of all gene models was conducted using the blastp against databases such as NR, SwissProt, KEGG, and COG. Additionally, tRNA genes were identified using tRNAscan-SE (v1.23)[106], and rRNA genes were identified using RNAmmer (v1.2)[107].

## Reconstruction of single-species models

A total of eighteen species were selected for model construction. Genomes of four strains were obtained by genome sequencing. Genome sequences of 14 other species were downloaded from public resources (NCBI and JGI[108]). The draft models were first constructed by ModelSEED[109] and then curated using COBRAToolbox-3.0[110,111]. We tested the growth of each strain in MM medium with different carbon and nitrogen sources and cofactors, and these experimental results were used for model curation (Supplementary Fig. 23). Generally, the draft models could not produce every biomass component under specific nutrients in which strain growth was feasible, as proven by experiments. The potentially missing reactions were first identified through an automated gap-filling process[111], and only reactions with gene evidence in the genomes of the strain or its phylogenetically closely related species were retained. Then, we artificially complemented missing reactions by gene/enzyme annotation according to public databases, such as KEGG[102,112], UniProt[113,114], BiGG[115], IMG[116], and MetaCyc[117]. In addition, all reaction IDs from different databases were converted for consistency; elementally imbalanced reactions on the basis of chemical formulae were checked and balanced; and futile loops were removed. For the raw model, all the transport reactions were supported by transporter annotations in the genome. For model curation, a transport reaction was added when it was supported by experimental data, considering the possibility of imperfect genome annotation. For example, the experimental data showing that the strain can assimilate a compound, the transport reaction of the compound should be added into the model. In summary, all transport reactions have genomic or experimental supports. After iterative revision, the final models were able to produce all biomass components in MM medium with alternative carbon and nitrogen sources, which was consistent with the experimental results.

## Microbiome model construction and optimal community combination analysis

For microbiome simulation, we developed a modeling framework called SuperCC for predicting metabolic flux distributions in microbial communities under different nutritional environments. Briefly, we integrated different single-species models into a multicompartment

model. Each single model was considered a distinct compartment, simulating an independent cell where reactions occurred. Additionally, a shared community compartment was established to facilitate the exchange of metabolites among species, mimicking the co-culture medium. Transport reactions were added to enable each cell to absorb or secrete metabolites from the culture medium, while exchange reactions represented the accumulation or consumption of metabolites in the medium obtained from the environment. The number of single-species models in the framework was not limited; thus, SuperCC was scalable to a large number of species suitable for both simple and complex microbiomes. We provided four commonly used scenarios, including: (1) equal abundance for each organism; (2) no limitations for any organisms, meaning that the biomass of each organism could be zero; (3) defining the biomass of a specific organism as the community biomass (used to identify organisms in a community that could improve the growth of the target organism); and (4) any defined abundances. To compare the performances of communities with different organisms, a set of compartmented community models was constructed covering all possible combinations of given organisms. The biomass functions are weighted combinations of molecules that are required for cellular growth and reproduction and are scaled such that the units are 1/h; concentrations are expressed in units of mmol/gDW. The community biomass was defined as linear combinations of biomass of each individual species.

We extended the FBA by integrating different single-species models into a multicompartment model[118]. The fluxes were calculated by the parsimonious FBA method (pFBA)[119], optimizing the biomass function while minimizing the flux of each nutrient exchange reaction through the model. Flux variability analysis (FVA) was used to identify the key reactions that contributed to improving the performance of the community. The key reactions were then added to the model of the target organism to construct a synthetic cell achieving improved function. The mathematical framework of SuperCC is shown below.

All species in the community are represented as $K$. The FBA for predicting maximum growth for each species $k$ in $K$ is described as:

$$\max v_{biomass}^k \text{ subject to}$$

$$\sum S_{ab}^k v_b^k = 0, \forall a \in A^k, \forall b \in B^k \tag{1}$$

$$LB_b^k \leq v_b^k \leq UB_b^k, \forall b \in B^k \tag{2}$$

where:

$S_{ab}^k$ is the stoichiometry for metabolite $a$ in reaction $b$. $v_b^k$ is the flux of reaction $b$, set to −mmol gDW⁻¹h⁻¹ for general metabolic reactions, and h⁻¹ for the biomass reaction. Each metabolite $a$ and reaction $b$ of organism $k$ are in the set of metabolites and reactions represented as $A^k$ and $B^k$. $LB_b^k$ is the lower bound that represents the amount of metabolite absorbed by species $k$. $UB_b^k$ is the upper bound for secreting metabolite $b$. Each reaction is limited by lower bounds ($LB$) and upper bounds ($UB$).

The mass balance of secretions and uptakes of each species in the extracellular space in the microbial community is stated as follows:

$$\left(\sum_{k \in K} v[ex]_a^k\right) + IP_a^k - OP_a^k = 0, \forall a \in A^{com} \tag{3}$$

$$\max \sum_{k \in K} c^k v_{biomass}^k \tag{4}$$

where:

$v[ex]_a^k$ is the flux of the exchange reaction for metabolite $a$ in the metabolic model of species $k$. $IP_a^k$ and $OP_a^k$ are the community import and output rates of metabolites $a$. $A^{com}$ is a set of metabolites shared in the community. The objective function of the community model is defined to contain the sum of the biomass fluxes of each organism in

equation [4]. $c^k$ is a vector of weights indicating how much each $v^k_{biomass}$ contributes.

## Testing the computational predictions

To test the predicted metabolic interactions among strains experimentally, co-cultures of different strains were used to detect secreted metabolites by strain. Strains X-1 and 7D-2 (with $OD_{600} = 0.35$ for each strain, washed three times with sterile water) were inoculated into MM medium supplemented with BO as a sole carbon and nitrogen source at 30 °C for 6 h, and then the predicted metabolites were screened by LC–MS. Similarly, strains 7D-2 and H8 were grown in MM medium supplemented with DBHB and $NH_4^+$. Pure compounds, including hypoxanthine, L-glutamate, xanthine, D-mannose, fumarate, succinate, D-glucosamine, and L-proline, were used as reference standards. The exchanged metabolites in the medium were detected by a LC–MS system (G2-XS QTof, Waters). A 2 μL solution was injected into the UPLC column (2.1 mm × 100 mm, ACQUITY UPLC BEH $C_{18}$ column containing 1.7 μm particles) at a flow rate of 0.4 mL/min. Buffer A consisted of 0.1% formic acid in water, and buffer B consisted of 0.1% formic acid in acetonitrile. The gradient was 5% Buffer B for 1 min, 5–95% Buffer B for 11 min, and 95% Buffer B for 2 min. Mass spectrometry was performed using MSe acquisition mode with a selected mass range of 50–1200 m/z (electrospray ionization in positive or negative ion mode). The ionisation parameters were the following: capillary voltage was 3.0 kV, cone voltage was 30 V, source temperature was 120 °C, and desolvation gas temperature was 400 °C. Collision energy was 20–40 eV. Data was acquired and processed using MassLynx 4.1, with ion chromatograms extracted at a 0.01 Da width. The signal-to-noise ratio thresholds for detection and quantitation were set at 3 and 10, respectively. The chromatographic process facilitated the separation of sample components, which were then identified and quantified based on their mass and retention times relative to known standards (Supplementary Table 9). Each metabolite concentration was determined using individual standard calibration curves.

BO and DBHB degradation by different consortia in both MM medium and in situ soils was measured to test the computational predictions experimentally. Detailed information on the eighteen isolates used is provided in Supplementary Table 3. BO degradation levels by two-member (X-1&7D-2, 1:1) and three-member consortia (combination of A8, AC6, B2, BR1, E44, LM5, P1, P56, R1 or Y13, with X-1&7D-2, 1:1:1) were measured. For DBHB degradation, two-member (H8&7D-2, 1:1) and three-member consortia (combination of AT5, B2, E3, P1, P29, P56, R1, R3, Y13 or Y3, with H8&7D-2, 1:1:1) were measured. The model microorganism *E. coli* was used to construct a negative three-member consortium with X-1&7D-2 or 7D-2&H8. Strains in each combination ($OD_{600} = 0.03$) were inoculated into 20 mL MM medium containing 50 mg/l BO or DBHB and enriched at 30 °C for 18 hours. In addition, the performances of different consortia were also tested in MM medium with BO/DBHB and glucose/$NH_4^+$. The biomass and remaining BO or DBHB were detected. Since BO is insoluble in water-forming emulsions, strain biomass in media with BO was detected by dilution plating on LB agar, while biomass in media with DBHB was detected through $OD_{600}$ using a Thermo Scientific Evolution 220 UV–Visible spectrophotometer. The in situ soil experiments were performed in sterilized soils. The two- and three-member consortia were inoculated into the corresponding soils with a final concentration of approximately $2 \times 10^8$ $CFU \cdot g^{-1}$ soil for each strain. The degradation ability of each combination in soils was detected as described above.

Similar to BO-treated groups (Fig. 7), simulations were conducted and tested for communities from DBHB-treated groups. Growth simulations were carried out in MM medium containing DBHB and $NH_4^+/NO_3^-$ as the nitrogen and carbon sources (DBHB-$NH_4^+/NO_3^-$ medium) and in DBHB-$NH_4^+$ medium supplemented with glucose (DBHB-$NH_4^+$-G medium) (Supplementary Fig. 18). We did not detect

any strains that could improve growth except for strain P56. Again, the predictions were consistent with the experimental results (Supplementary Fig. 19). The weak enhancement of degradation by combining strain P56 may be due to the original high degradation rate by 7D-2&H8, especially in yellow cinnamon soils (>95%). We tested the growth promotion in MM medium with glucose and/or $NH_4^+$ (Supplementary Fig. 20). The results showed no effects or even inhibition of community growth by the combination of 7D-2&H8 with other strains, which was also consistent with the simulations.

## DNA SIP experiment

Since 4-hydroxybenzoic acid is the major intermediate metabolite of BO and could not be degraded by most species (Supplementary Fig. 24), the synthetic microbiome (containing strains X-1, 7D-2, B2, LM5, E44, P56, and *E. coli*) from yellow cinnamon soils was fed with normal 4-hydroxybenzoic acid or $^{13}$C-labled 4-hydroxybenzoic acid to explore metabolic interactions. In this consortium, only strain 7D-2 could degrade 4-hydroxybenzoic acid and *E. coli* was used as a negative control. Due to the ability of strain AC6 to degrade 4-hydroxybenzoic acid, it was excluded from this test. After adjusting the cell density to 0.03 ($OD_{600}$), the cultures were diluted 20 times and then inoculated into 40 mL MM medium containing either 25 mg/L of $^{12}$C 4-hydroxybenzoic acid or 25 mg/L of $^{13}$C-labled 4-hydroxybenzoic acid at 3 % v/v. After 8 h of incubation at 30 °C, approximately 80% of the 4-hydroxybenzoic acid was degraded, and the total DNA was extracted using the FastDNA Spin Kit (Solon, USA) according to the manufacturer's instructions. The experiments were carried out in triplicate. For ultra-high density centrifugation, 1600 μg of DNA from each sample was dissolved in Tris-EDTA (pH 8.0)-CsCl solution, and the final buoyant density was adjusted to 1.85 g/mL. Then, the samples were transferred into a Quick-Seal centrifuge tube (13 × 51 mm; Beckman Coulter, Pasadena, CA). The buoyant density was detected using a digital refractometer (model AR200; Leica Microsystems Inc., Buffalo Grove, IL). After heat sealing and equilibration, the centrifuge tubes were ultra-centrifuged (Optima L-100XP; Beckman Coulter, USA) at 190,000 g (20 °C) for 44 h, and DNA was fractionated in the tube. Subsequently, different fractions were collected by a fraction recovery system (Beckman Coulter). Finally, 14 fractions were collected within each sample, and the fractionated DNA was purified using the Universal DNA Purification Kit (TIANGEN Biotech, Beijing). The effectiveness of density gradient centrifugation was determined by measuring the refractive index of different DNA samples. Fractions 1-2, 3-4, 5-6, 7-8, 9-10, 11-12, and 13-14 were merged for microbial community sequencing.

## Transcriptome analysis

To explore possible physiological explanations for mutualism between X-1 and 7D-2, we conducted gene expression profiling of the two strains using RNA-seq. For X-1, the strain was pre-cultured in LB medium containing 50 mg/L BO for 24 h to ensure a consistent metabolic state for all cells. Subsequently, the pre-cultured X-1 was cultured in a new LB medium ($OD_{600} = 0.2$) containing 50 mg/L BO for 1 h to activate gene expressions. The cells were then collected by centrifugation for further processing. For 7D-2, the strain was pre-cultured in LB medium containing 50 mg/L bromoxynil for 24 h. Similar to X-1, cells of 7D-2 were first cultured in a new LB medium ($OD_{600} = 0.2$) containing 50 mg/L bromoxynil for 1 h, and then collected for further treatments. The following treatments were set as follows: i) X-1 growing in MM medium containing BO for 1.5 h (single-culture of X-1), ii) 7D-2 growing in MM medium containing bromoxynil for 1.5 h (single-culture of 7D-2), and iii) X-1 and 7D-2 co-culturing in MM medium containing BO for 1.5 h (co-cultures). Total RNA was extracted using TRIzol Reagent according to the manufacturer's instructions. RNA-seq strand-specific libraries were prepared following TruSeq RNA sample preparation Kit from Illumina (San Diego, CA, USA), using 5 μg of total RNA.

The paired-end libraries were sequenced using the Illumina NovaSeq 6000 sequencing platform at Shanghai BIOZERON Biotech. Co., Ltd (Shanghai, China). The raw reads were trimmed and quality-controlled using Trimmomatic[103]. Then clean reads were aligned separately to the reference genome using Rockhopper software[120] which was also used to calculate gene expression levels with default parameters. The expression level for each transcript was calculated using the transcripts per kilobase million (TPM) method. Significantly differential expression genes (DEGs) were identified using edgeR[121] with a false discovery rate (FDR) < 0.05 and an absolute of fold change ≥ 2.

## Reporting summary

Further information on research design is available in the Nature Portfolio Reporting Summary linked to this article.

## Data availability

All amplicon sequencing data have been deposited in the NCBI Sequence Read Archive (SRA) under the accession number PRJNA799911. The metagenomic sequence data have been deposited in the NCBI SRA under the accession number PRJNA799881. The RNA-sequencing data have been deposited in the NCBI SRA under the accession number PRJNA1073009. The GenBank accession numbers for genome sequences of *Pseudoxanthomonas* sp. X-1, *Pigmentiphaga* sp. H8, *Arthrobacter* sp. AT5, and *Comamonas* sp. 7D-2 are VCHZ00000000, CP033966, CP136441, and CP094238, respectively. The GenBank accession number for plasmid of *Comamonas* sp. 7D-2 is KC771559. The data that support this study are available within the article and its Supplementary Information files. Source data are provided with this paper.

## Code availability

The code used for the analysis is available on GitHub (https://github.com/ruanzhepu/superCC)[122].

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

## Acknowledgements

This work was supported by grants of the National Key Research and Development Program of China (2018YFA0901200 to X.X., J.J. and K.C.), the National Natural Science Foundation of China (41977120 to X.X.; 42177031 to X.X.; and 41925029 to Y.G.), and the National Key Research and Development Program of China (2023YFE0110800 to Y.G.). We thank Professor Yun Tian and Ph.D. Yuqian Li from Xiamen University for sharing the strain *Aliihoeflea aestuarii* N8.

## Author contributions

Z.R., K.C., X.X., Y.G. and J.J. conceived the overall study design. Z.R. and K.C. performed soil treatment and sampling. Z.R., K.C., W.C., L.M., B.Y., M.X., and Y.X. carried out the experiments. Z.R. and X.X. carried out bioinformatics analysis. Z.R. and X.X. reconstructed models and performed modeling analysis. X.X., Y.G. and J.J. supervised the research. Z.R., X.X. and J.J. wrote the manuscript. X.X., C.C. K.C., P.L., S.F., Y.G. and J.J. revised the manuscript. All authors read and approved the final manuscript.

## Competing interests

The authors declare no competing interests.
