## [Peer Review File · Nature Communications]

Engineering natural microbiomes toward enhanced bioremediation by microbiome modelingReviewer #1 (Remarks to the Author):

The manuscript by Raun et al. presents a framework for developing synthetic microbiomes that have improved degradation capabilities of two herbicides (bromoxynil and 3,5-dibromo-4-hydroxybenzoate; abbreviated BO and DBHB, respectively). The approach consists of enriching natural soil-derived microbiomes under different treatments (e.g., presence of herbicide or bioaugmentation of strains), identifying keystone species using metagenomics and statistical analysis, isolating keystone species, and assembling synthetic consortia using a genome-scale modeling pipeline (termed SuperCC) that have improved herbicide degradation rates. Using this approach, the authors show that different natural microbiomes converged taxonomically and functionally across different treatments towards improved BO/DBHB degradation and that inoculation of degrading consortia resulted in higher BO/DBHB degradation rates than herbicide treatments. They then identify 18 keystone species from across their enrichment cultures and use genome-scale modeling to identify interspecies interactions (e.g., metabolite exchanges) and rationally design 2-3 member synthetic communities with enhanced BO/DBHB degradation. Finally, the authors use their modeling pipeline to identify the set of reactions across microbiome members that could be added to a synthetic cell to achieve efficient BO/DBHB degradation.

While I find the overall approach presented by Raun et al. to be promising for engineering microbiomes towards enhanced bioremediation, I have several major issues with the manner in which experiments and results are presented, and with the modeling analysis, as outlined below:

1. A key metric relied upon in the study is degradation rate of either BO or DBHB although very few details are provided on how this was determined. For example, Figure 2 shows degradation rates for different timepoints across treatments. Were soil subsamples from the different treatments brought into fresh media to determine these rates at each timepoint? How were rates normalized relative to total cell density? This is particularly important to control, as it's well known that a higher cell density will result in higher degradation. Therefore, specific degradation rates versus overall degradation rates should be reported throughout the manuscript.

2. Several hypotheses are put forward on metabolite exchanges in the microbiomes that improved degradations rates based on genome-scale modeling. For example, the exchange of various sugars and amino acids (Fig 7 & Fig S12). However, no physiological explanations are provided as to why these exchanges improve degradation, although these should be apparent from the modeling. Can the authors provide this?

3. The metabolomic data presented to validate the proposed metabolite exchanges is very poorly presented and described. Actual concentrations (not peak areas) should be presented and controls for the blank media and cultures growing separately should be provided. Moreover, full details on how metabolites were identified (e.g., based on accurate mass plus retention time of labelled standards) should be described in detail in the methods.

4. Relatedly, an additional caveat is that the measurement of a metabolite doesn't imply they are being exchanged. Isotope tracing experiments would be required at a minimum to support this. This should be highlighted as a limitation of the study.

5. I have issue with the way the model was formulated, which was based on a shared "community" biomass objective function for all species. Imposing such an objective on the model often results in nonsensical metabolite exchange solutions because species are artificially rewarded for the growth of other species. Moreover, this approach ignores kinetic differences (e.g., substrate affinity, maximum substrate uptake) which are well known to impact microbial interactions. A more realistic simulation would have been to use dFBA (PMID: 20668487) where an objective function is solved independently for each species to avoid spurious interactions. When the authors take such an approach do they still obtain the same predicted interactions?

Additional comments:

- In general, the figures need to be described in more detail in the figure legend

- Figure 2 - how was % degradation measured?
- Figure 3a – what do lowercase letters at top of plot mean?
- Page 9, line 169-184 For inoculum treatments where species diversity decreased, did the inoculated strains increase in abundance?
- Line 167 – what does “domestication efficiency” mean? Do you mean degradation efficiency?
- Line 182-184 this is highly speculative and could suggest many things. Are there previous studies that support this claim?
- Fig 3 - It would be helpful if a clear breakdown of the samples and their corresponding treatment were provided in a table. In particular, Fig 3b/c – there are several icons (shape with color) that are identical – what do they represent, replicates?
- In all samples methanol was also added – how much of the available carbon did this represent and did it contribute to the convergence of taxa observed?
- From metagenomic/16S rRNA amplicon analysis, did a common taxa emerge across all enrichments? It would be useful to highlight the core taxa that become abundant and converged overtime.
- Fig 4 - How was the number of genes normalized to sequencing depth?
- Fig 4d – what was the change in phyla and taxa from day 0 to day 30? That would be better evidence of convergent succession versus just day 30 day. Can the data be presented at a resolution lower than phyla?
- Fig 5d – it is not clear from Fig5d how the authors concluded that inoculation treatment influenced the microbiome more than profoundly than herbicide treatment. Also, what exactly is meant by “influenced” – similarity/dissimilarity in taxonomic composition?
- Fig 6C – what about abundance shifts in soils only inoculated with herbicides?
- Fig 7a - What was the actual concentration of metabolite (not area’s) detected in the samples? Can the authors also show measurements from soil control samples without culture and immediately after inoculation?
- Fig 8 - What is the physiological rationale for these organisms to secrete these compounds?
- Figure 7g – what is abundance ratio? Why not show the relative abundance of this strain in the community?
- Page 15, line 306-308 – were transport reactions supported by transporter annotations in the genome? Or was everything allowed to be exchanged? How was this validated?
- Fig 8b / S10 – were degradation rates normalized to the amount of microbial cells added? More details on experiments needed. Were these experiments replicated? I don’t see any error bars

Reviewer #2 (Remarks to the Author):

In this manuscript, the authors use a structured approach for targeted microbiome application based on domesticating bacterial communities from natural soil samples and use in-silico approaches to optimize the functional value of the derived synthetic microbiomes. They use different treatments to isolate the influence domesticating the microbial consortium either via addition of two different herbicides or the inoculation of bacteria with desired functions (as individual species, competing species or synergistic species). They further use LEfSe analysis to identify keystone species in the soil samples and use a compartmentalized community FBA approach to evaluate the metabolic flux exchange between different members of the community. As I am no expert on bacterial molecular approaches and diversity, I cannot assess the approach and quality of the actual metagenomic analysis.

Overall, this is a potentially impactful paper that would benefit from clarification about the data and treatments that have been used for specific Figures etc. as disentangling the actual findings is otherwise cumbersome.

My main concern with the experimental system is the high addition of inoculated cells. Adding 2×10^8 CFU/g soil for each species ten times over 30 days is a total of 4×10^9 CFU/g soil. Considering that typical numbers in soil are 10^{10} cells/g soil, this results in a replacement of 40% of the soil community. I fear that this high number of added cells biases the degradation efficiency of the communities (Figure 2), governs the loss of diversity (Figure 3 panel a), alters the calculated Bray-Curtis distance (Figure 4) and artificially enriches the abundance of functional genes (Figure 4 panel c). The question then becomes if this is a true domestication, i.e. will this

approach be suitable for real in-situ bioremediation using the functional community since the amount of applied bacteria needed to remediate a single ton of soil would be astronomical. Testing the success of inoculating a lower abundance of cells would be an interesting avenue and/or if a single, initial inoculation is sufficient to stimulate the growth of the degrading bacteria.

Throughout the manuscript, the authors talk about the degradation rate of the herbicide by the different treatment approaches. However, from Figure 2 (where the unit on the y-axis is %) and from the methods, I assume that the unit in this Figure is the percentage of remaining herbicide from the last application. In this case, defining this as a degradation rate (which should be a quantity per time) is misleading and may result in quite some confusion for the readers. I would suggest to use a bar plots instead of connected lines (they are not directly linked as there is continuous reapplication of species and herbicide).

Next to the overall very high application of bacterial cells every three days, a second complication when interpreting the diversity results (Figure 3 panel a) and Bray-Curtis distances (Figure 3 c&d) stems from the uncertainty in the overall diversity loss due to the change in conditions simply based on the experimental system. As outlined in the methods section, the experiments were prepared by grinding the soil and transferring into plastic pots where the hydration conditions are kept constant at 40-60% water holding capacity. An important control for this experiment would be to look at the change in diversity when neither species or herbicide is added to the pots. Such a drastic change in conditions for bacterial cells can automatically result in a loss of diversity and disentangling the actual effect of domestication from the application of species/disruption in their habitat is difficult. Figure 3 panel a shows that there was a control at T0 (I assume before application of the treatments) but this is then not repeated for the subsequent time points.

Finally, the authors do not include the information about which treatments are used in the mapping of functional genes (Figure 4 panel c). In case this is for the treatments where different individual strains, the competitive or synergistic community is added over time, I am not surprised that the final community contains far higher abundance of these function genes. Due to the very high application of inoculated cells, I believe the data that would support the domestication of a functional community should be taken from the treatment where only herbicide is added to the soil.

Smaller comments:

Line 21: Long sentence – consider rephrasing.

Line 38: The phrase suggests that human health is a biogeochemical cycle?

Line 52: Sentence is somewhat convoluted.

Line 207: Important to state which treatments were used to generate Figure 4c.

Line 555: What is the effect of using these different formulations of biomass parametrizations?

Line 589: When inoculating the strains for metabolite exchange measurements, were the single cell dilutions (OD600 of 0.35) washed before co-inoculation?

Line 626: I could not find the optimized Cb models on the provided GitHub page.

Line 947: The statement that the Bray-Curtis distances are increasing with time is somewhat misleading. The overall correlation across all soils is rather weak and for some soils there is a clear decrease (e.g. Y in DBHB&7D-2&H8).

Line 958: There is no information on which treatments were used for the generation of panel C. Is this a combination of all? Or only a subset? How do the treatments where domestication was performed by addition of only herbicide compare to domestication via addition of specific strains?

Comments on Figures:

In many figures, purple is misspelled as "pruple"

Figure 2: The connected lines and designation as "degradation rate" is somewhat misleading. Present as bar plots?

Figure 3: Most of the datapoints are outside of the 95% CI in the linear fits. Should these be done for the individual soils? (The trend between soils are often different).

Figure 8: What is the unit of biomass?

Reviewers' comments:

Reviewer #1 (Remarks to the Author):

The manuscript by Ruan et al. presents a framework for developing synthetic microbiomes that have improved degradation capabilities of two herbicides (bromoxynil and 3,5-dibromo-4-hydroxybenzoate; abbreviated BO and DBHB, respectively). The approach consists of enriching natural soil-derived microbiomes under different treatments (e.g., presence of herbicide or bioaugmentation of strains), identifying keystone species using metagenomics and statistical analysis, isolating keystone species, and assembling synthetic consortia using a genome-scale modeling pipeline (termed SuperCC) that have improved herbicide degradation rates. Using this approach, the authors show that different natural microbiomes converged taxonomically and functionally across different treatments towards improved BO/DBHB degradation and that inoculation of degrading consortia resulted in higher BO/DBHB degradation rates than herbicide treatments. They then identify 18 keystone species from across their enrichment cultures and use genome-scale modeling to identify interspecies interactions (e.g., metabolite exchanges) and rationally design 2-3 member synthetic communities with enhanced BO/DBHB degradation. Finally, the authors use their modeling pipeline to identify the set of reactions across microbiome members that could be added to a synthetic cell to achieve efficient BO/DBHB degradation.

While I find the overall approach presented by Raun et al. to be promising for engineering microbiomes towards enhanced bioremediation, I have several major issues with the manner in which experiments and results are presented, and with the modeling analysis, as outlined below:

1. A key metric relied upon in the study is degradation rate of either BO or DBHB although very few details are provided on how this was determined. For example, Figure 2 shows degradation rates for different timepoints across treatments. Were soil subsamples from the different treatments brought into fresh media to determine these rates at each timepoint? How were rates normalized relative to total cell density? This is particularly important to control, as it's well known that a higher cell density will result in higher degradation. Therefore, specific degradation rates versus overall degradation rates should be reported throughout the manuscript.

Answer: Thanks for the reviewer's comments. We have revised the Methods and Supplementary Materials to give the details of methods to determine the degradation rate.

We agree with the reviewer that a higher cell density will result in higher degradation. Consistently, we compared the degradation rates of treatments with low- and high-dose of inoculation in the revised manuscript, which showed that high-dose of inoculation had more remarkable influence on bacterial community and showed much higher BO-degrading ability. We used the strategy with high domestication

strength (i.e. high-dose of inoculation) to reduce time and accelerate microbiome designing. Besides, we would like to describe the aim of the domestication experiment. We intended to obtain the in-situ strains that interacting with degraders (i.e. supporting strains), which provide candidates for constructing synthetic microbiomes. Therefore, the degradation efficiency, and microbial community diversity and abundance of functional genes were detected to identify the alteration caused by our degradation treatment to select supporting strains. Furthermore, the indigenous degraders (not the inoculated strains) were detected for the purple and red soils, not the yellow cinnamon soils; and single strains and consortia were used as inocula in the study. Therefore, it is not safe to compare the degrading rates after normalization relative to total cell density (as uncertainty of degraders). In the revised manuscript, we have focused on the residues of herbicides, which are safe to compare and show the effectiveness of demonstration.

On Lines 427-429, we write:

“0.5 g of soil was collected to assess the degrading ability of BO or DBHB with high-performance liquid chromatography (HPLC). The detection methods are described in the Supplementary Materials”

On Supplementary Materials, we write:

“To detect the degrading ability of BO or DBHB by microbial community with different treatments, 0.5 g of soil was collected from different treatments and brought into fresh MM media supplemented with BO or DBHB (50 mg/L). After incubating on a shaker at 30 °C for 10 h (160 rpm), 1 mL of the liquid medium was taken from the medium to detect the residues of BO, bromoxynil or DBHB. All treatments were performed in triplicate by HPLC.”

On Figure 2, we write:

“

Fig. 2 Enhanced degradation efficiency of pollutants by microbiomes driven by

domestication. The residues of BO and DBHB are shown. ...”

2. Several hypotheses are put forward on metabolite exchanges in the microbiomes that improved degradations rates based on genome-scale modeling. For example, the exchange of various sugars and amino acids (Fig 7 & Fig S12). However, no physiological explanations are provided as to why these exchanges improve degradation, although these should be apparent from the modeling. Can the authors provide this?

Answer: Thanks for the reviewer’s comments. In the revised manuscript, we have provided results of additional experiments showing that the exchanged metabolites can improve the growth of degraders, and enhance the degradation (Fig. S9). The results indicated the exchanged metabolites could be additional carbon and/or nitrogen source for degraders which increase the biomass of degraders and finally improve degradation. The information has been included in the revised manuscript. On Lines 233-237, we write:

“In addition, we tested the growth and BO/bromoxynil/DBHB degradation in monocultures of the three species (X-1, 7D-2, and H8) grown on minimal media, each supplemented by the relevant exchange metabolites. The growth and degradation were enhanced in the supplemented medium for strains X-1, 7D-2, and H8 (Fig. S9), which were consistent with predictions.”

On Fig. S9, we write:

“

Supplementary Fig. S9. In vitro measurements of growth and fraction of BO/DBHB left in medium containing BO/DBHB as a sole nitrogen and carbon source vs. the same

medium supplemented by exchange metabolites. The growth and degradation was enhanced in the supplemented medium for strains X-1, 7D-2, and H8, indicating the exchanges could be additional carbon and/or nitrogen source for degraders which increased the biomass of degrader and finally improve the degradation.”

3. The metabolomic data presented to validate the proposed metabolite exchanges is very poorly presented and described. Actual concentrations (not peak areas) should be presented and controls for the blank media and cultures growing separately should be provided. Moreover, full details on how metabolites were identified (e.g., based on accurate mass plus retention time of labelled standards) should be described in detail in the methods.

Answer: Thanks for the reviewer’s comments. According to the reviewer’s comments, the concentration of each metabolite has been provided in the revised Fig. 7. The controls for the blank media have also been provided in Fig. 7. Besides, results for cultures growing separately have been provided in Table S3-Table S9. The detailed methods to identify metabolites have been provided in Methods and Supplementary Materials.

On Lines 527-528, we write:

“The details of LC–MS analysis are described in the Supplementary Materials.”

On Figure 7, we write:

“

Fig. 7 Simulations and experimental validations of metabolic interactions in two-member consortia. (a) Prediction of exchanged metabolites between strains 7D-2 and X-1. (b) Identification of the predicted exchanged metabolites via LC–MS in a

coculture of strains 7D-2 and X-1. Medium with BO as the only carbon and nitrogen source was used for both simulation and experimental validation. ...

On Supplementary Materials, we write:

“3. LC-MS analysis

...The sample solution was initially flowed into the chromatograph. The varying interactions between different substances and the stationary and mobile phases within the chromatograph resulted in distinct migration rates, ultimately achieving the separation of sample components. Based on the relative molecular mass and peak time of each component, they were matched with the target standards to determine the metabolites in the sample. A corresponding concentration standard curve was prepared for each standard to be matched (Supplementary Table S11), and the concentration was determined from the peak area ratio relative to individual standard calibration curves.”

On Supplementary Tables, we write:

“**Supplementary Table S11.** Detailed information about standard metabolites detected by LC-MS.

Metabolites	Mass	Retention time (min)	Positive/negative ion mode
D-Glucosamine	180.0872	0.79	+
L-Lysine	147.1134	0.75	+
L-Proline	116.0712	0.90	+
Xanthine	153.0413	1.29	+
L-Glutamate	147.0610	1.52	+
Hypoxanthine	137.0463	1.19	+
(R)-3-Hydroxybutanoate	103.0395	1.67	-
Fumarate	115.0031	1.54	-
D-mannose	179.0566	0.85	-
Succinate	117.0188	1.57	-

”

4. Relatedly, an additional caveat is that the measurement of a metabolite doesn't imply they are being exchanged. Isotope tracing experiments would be required at a minimum to support this. This should be highlighted as a limitation of the study.

Answer: Thanks for the reviewer's comments. According to the reviewer's comments, we have added results of many additional experiments to support the conclusion that the metabolites were exchanged, including the isotope tracing experiments, co-culture experiments and metabolomics profiling.

We firstly detected the exchanged metabolites in monocultures of different species and compared to those in the co-cultures to test the source of the exchanged metabolites. Take the consortium X-1&7D-2 as an example (Table S4). D-glucosamine was not detected in monocultures of X-1 or 7D-2, but was detected in co-culture of X-1 and 7D-2. Besides, 7D-2 were added to the monocultures of X-1 (after culture of X-1 for 4h, with X-1 in the culture), and we detected the D-glucosamine. However, when 7D-2 were added to centrifuged monocultures of X-1

(with X-1 removed from the culture by centrifuge), no D-glucosamine were detected. The results showed D-glucosamine was secreted by strain X-1 instead of 7D-2. Similar experiments were also performed for most of the exchanged metabolites. By these results, we showed the source of the exchanged metabolites.

Secondly, we tested the ability of corresponding strains to assimilate the exchanged metabolites. To this end, we compared the growth of strains in medium containing BO/DBHB as a sole nitrogen and carbon source vs. the same medium supplemented by exchange metabolites. The growth was enhanced in the supplemented medium for strains X-1, 7D-2, and H8, indicating the exchanges are available resources for assimilation (Fig. S9, please see our response to Comment 2).

Finally, to test if exchange metabolites were actually being exchanged, we used DNA stable isotope probing (SIP) and amplicon sequencing to detect the strains that involved in the assimilation of ^{13}C carbon. We used ^{13}C -labeled 4-hydroxybenzoic acid (the intermediate metabolite for BO degradation), as no ^{13}C -labeled BO could be purchased. We firstly showed each strain could not assimilate 4-hydroxybenzoic acid except for 7D-2, which is same for BO (Fig. S18). The strains were co-cultured in MM medium with ^{13}C -labeled 4-hydroxybenzoic acid as sole nitrogen and carbon source. The results showed that 4 strains (7D-2, X-1, LM5, and B2) were involved in the assimilation of ^{13}C carbon while 2 strains (E44 and *E. coli*) were not (Fig. S15, Supplementary Materials). These results showed the exchange metabolites are exchanged among strains as predicted. All these information have been included in the revised manuscript.

On Lines 233-242, we write:

“we tested the growth and BO/bromoxynil/DBHB degradation in monocultures of the three species (X-1, 7D-2, and H8) grown on minimal media, each supplemented by the relevant exchange metabolites. The growth and degradation were enhanced in the supplemented medium for strains X-1, 7D-2, and H8 (Fig. S9), which were consistent with predictions. In addition, we detected the exchanged metabolites in monocultures of the three species and compared to those in the co-cultures to test the source of the exchanged metabolites. For 7D-2&X-1, the hypoxanthine and D-Glucosamine were secreted by X-1 (Table S3, S4) while xanthine and L-glutamate were secreted by 7D-2 (Table S5); for 7D-2&H8, succinate and L-glutamate were secreted by 7D-2 (Table S6, S7) while D-Glucosamine was secreted by H8 (Table S8).”

On Lines 298-315, we write:

“Similar to the verification for two-strain community models (7D-2&X-1 and 7D-2&H8), we detected the exchanged metabolites in cocultures of 7D-2&X-1&BR1 to test the perdition of three-strain model (Table S9). All the 9 exchanged metabolites were successfully detected in LC-MS (except for 2-oxoglutarate). Then we compared the exchanged metabolites in monocultures with the co-cultures to test the source of the exchanged metabolites for 7D-2&X-1&BR1. The results showed that fumarate, 4-Hydroxybenzoate, succinate, L-lysine, (R)-3-Hydroxybutanoate were exchanged metabolites between 7D-2 and BR1, while hypoxanthine, L-glutamate, xanthine were between 7D-2 and X-1, which were consistent with the predictions (Table S9). Finally, we used DNA stable isotope probing (SIP) combined with amplicon sequencing to

detect the strains that involved in the degradation of BO indicated by the assimilation of ^{13}C carbon (Supplementary Materials). We used ^{13}C -labeled 4-hydroxybenzoic acid (the intermediate metabolite for BO degradation), as no ^{13}C -labeled BO could be purchased. In the co-cultures of 7 strains (including 7D-2, X-1, P56, E44, LM5, B2, and *E. coli*), 4 strains (7D-2, X-1, LM5, and B2) were involved in the assimilation of ^{13}C carbon while 2 strains (E44 and *E. coli*) were not (Fig. S16). The results were agreed with the perditions (except for P56) that strains LM5 and B2 could improve the BO degradation by 7D-2&X-1 while E44 could not. No assimilation of ^{13}C carbon was detected for strain P56, which might result from the insufficient amount of ^{13}C carbon used in the study and/or the competition among strains P56, LM5, and B2.” On Supplementary Materials, we write:

“DNA stable isotope probing experiment

Since 4-hydroxybenzoic acid is the major intermediate metabolite of BO and could not be degraded by most species (Fig. S18), the synthetic microbiome (containing strains X-1, 7D-2, B2, LM5, E44, P56, and *E. coli*) from yellow cinnamon soils was fed with normal 4-hydroxybenzoic acid or ^{13}C -labeled 4-hydroxybenzoic acid to explore the metabolic interactions. In this consortium, only strain 7D-2 could degrade 4-hydroxybenzoic acid and *E. coli* was used as a negative control. Due to the ability of strain AC6 to degrade 4-hydroxybenzoic acid, it was excluded from this test.

After adjusting the cell density to 0.03 (OD_{600}), the cultures were diluted 20 times and then inoculated into 40 mL MM medium containing either 25 mg/L of ^{12}C -labeled 4-hydroxybenzoic acid or 25 mg/L of ^{13}C -labeled 4-hydroxybenzoic acid at 3 % v/v. After 8 h of incubation at 30 °C, approximately 80% of the 4-hydroxybenzoic acid was degraded, and the total DNA was extracted using the FastDNA Spin Kit (Solon, USA) according to the manufacturer’s instructions. The DNA purity and concentration were determined by a NanoDrop 2000UV–vis spectrophotometer (Thermo Fisher Scientific, Shanghai, China). The experiments were carried out in triplicate.

The subsequent ultra-high density centrifugation layering process refers to the method described previously. Briefly, about 1600 μg of DNA from each sample was dissolved in Tris-EDTA (pH 8.0)-CsCl solution, and the final buoyant density was adjusted to 1.85 g/mL. Then, the samples were transferred into a Quick-Seal centrifuge tube (13 \times 51 mm; Beckman Coulter, Pasadena, CA). The buoyant density was detected using a digital refractometer (model AR200; Leica Microsystems Inc., Buffalo Grove, IL). After heat sealing and equilibration, the centrifuge tubes were ultra-centrifuged (Optima L-100XP; Beckman Coulter) at 45,000 rpm (20 °C) for 44 h, and DNA was fractionated in the tube. Subsequently, the different fractions were collected by a fraction recovery system (Beckman Coulter). Finally, 14 fractions (364 μL each) were collected within each sample, and the fractionated DNA was purified using the Universal DNA Purification Kit (TIANGEN Biotech, Beijing). The effectiveness of density gradient centrifugation was determined by measuring the refractive index of different DNA samples. Fractions 1-2, 3-4, 5-6, 7-8, 9-10, 11-12, and 13-14 were merged for microbial community sequencing.”

On Supplementary Figures and Tables, we write:

“

Supplementary Fig. S16. Relative abundances of ASVs in fraction 7 to 10 of normal 4-hydroxybenzoic acid and ¹³C-labeled 4-hydroxybenzoic acid samples after fractionation. Fraction 7-8 and 9-10 were merged and were marked as the “heavy DNA”.

Supplementary Table S4. LC-MS analysis of the metabolite D-glucosamine in consortium samples of BO groups. BO, bromoxynil octanoate; OAN, octoic acid & NH₄⁺; Bro, bromoxynil. X-1&7D-2&BO, the two strains were inoculated at the same time. (X-1&BO-4h)+7D-2, the strain X-1 was cultured in the MM medium containing BO for 4 hours, and then the strain 7D-2 was added. Excretion of (X-1&OAN)+7D-2&Bro, the strain X-1 was cultured in the MM medium containing BO for 4 hours and centrifuged to remove the bacteria, and then the strain 7D-2 and bromoxynil were added. All final cultures were tested after 4 hours of culture.

	X-1&7D-2&BO	(X-1&BO-4h)+7D-2	Excretion of (X-1&OAN)+7D-2&Bro	X-1&BO	7D-2&Bro
D-Glucosamine	+	+	-	-	-

”

Supplementary Tables S3-S9 have been provided in Supplementary Materials.

5. I have issue with the way the model was formulated, which was based on a shared “community” biomass objective function for all species. Imposing such an objective on the model often results in nonsensical metabolite exchange solutions because species are artificially rewarded for the growth of other species. Moreover, this approach ignores kinetic differences (e.g., substrate affinity, maximum substrate uptake) which are well known to impact microbial interactions. A more realistic simulation would have been to use dFBA (PMID: 20668487) where an objective function is solved independently for each species to avoid spurious interactions. When the authors take such an approach do they still obtain the same predicted interactions?

Answer: Thanks for the reviewer's comments. We agree with the reviewer that the shared "community" biomass might lead to nonsensical metabolite exchange solutions. In our modeling framework, we used the parsimonious FBA method (pFBA) to calculate the fluxes to avoid nonsensical metabolite exchange solutions. By pFBA, the biomass was optimized while minimizing the flux of each nutrient exchange reaction through the model. Besides, our modeling framework provide four commonly used scenarios, including 1) equal abundance for each organism; 2) no limitation for any organisms, meaning that the biomass of each organism is allowed to be zero; 3) defining biomass of a specific organism as community biomass (used for identifying organisms in a community that could improve the growth of the target organism); and 4) any defined abundances. The scenarios could be selected and used according to different aims of the modeling, and also could help to avoid nonsensical metabolite exchange solutions.

Again, we agree with the reviewer that dFBA is another method to avoid spurious interactions. Actually, we did used dFBA to simulate the performance of a simple microbial community and got a pretty good result (Xu *et al.*, ISME Journal, 2019; PMID: PMC6331595). The disadvantage of dFBA is that dFBA needs large amounts of computation. During dFBA, the total time period is often subdivided into predefined discrete time intervals (usually > 1000 or more), and biomass flux for each species should be optimized using the standard FBA for each time intervals. Therefore, dFBA is not suitable for simulating complex microbiome containing a large number of species. In this manuscript, we aimed to develop a modeling framework, i.e. SuperCC, which is suitable for both simple and complex microbiome. By SuperCC, we could get results as accurate as dFBA but with much less computation amount and computation time.

Additional comments:

- In general, the figures need to be described in more detail in the figure legend

Answer: Thanks for the reviewer's comments. All figure legends has been revised thoroughly to provide more details.

Figure 2 - how was % degradation measured?

Answer: Thanks for the reviewer's comments. In the revised Figure 2, "residues of herbicides" were has been used instead of "degradation rates". We have included the detailed methods to measure the residues of herbicides in the revised manuscript.

On Lines 427-429, we write:

"0.5 g of soil was collected to assess the degrading ability of BO or DBHB with high-performance liquid chromatography (HPLC). The detection methods are described in the Supplementary Materials"

On Supplementary Materials, we write:

"To detect the degrading ability of BO or DBHB by microbial community with different treatments, 0.5 g of soil was collected from different treatments and brought into fresh MM media supplemented with BO or DBHB (50 mg/L). After incubating on a shaker at 30 °C for 10 h (160 rpm), 1 mL of the liquid medium was taken from the

medium to detect the residues of BO, bromoxynil or DBHB. All treatments were performed in triplicate by HPLC.”

- Figure 3a – what do lowercase letters at top of plot mean?

Answer: Thanks for the reviewer’s comments. The lowercase letters refer to different treatments. In the revised Figure 3, the lowercase letters have been removed to avoid misunderstanding and make the Figure concise.

- Page 9, line 169-184 For inoculum treatments where species diversity decreased, did the inoculated stains increase in abundance?

Answer: Thanks for the reviewer’s comments. The relative abundances of inoculated stains (7D-2 and X-1) have been provided in Fig. S11. These results showed that the abundance of 7D-2 remarkably increased and those of X-1 increased slightly. On Fig. S11, we write:

“

Supplementary Fig. S11. The relative abundances of strain 7D-2 and X-1 in treatments of inoculation. -2 and -3 represent treatments with the synergistic consortium (7D-2&X-1), and the combination of BO and the synergistic consortium (BO&7D-2&X-1), respectively.”

- Line 167 – what does “domestication efficiency” mean? Do you mean degradation efficiency?

Answer: Thanks for the reviewer’s comments. We have removed “domestication efficiency” in the revised manuscript to avoid misunderstanding, and the relative sentence has been rephrased.

On Lines 142-143, we write:

“These results showed inoculation of degrading consortia was feasible for both simple and complex pollutants.”

- Line 182-184 this is highly speculative and could suggest many things. Are there previous studies that support this claim?

Answer: Thanks for the reviewer’s comments. We agree with the reviewer that the conclusion is speculative. Few previous studies have reported it or similar results.

Besides, the conclusion is not the main point of the manuscript. Therefore, we have removed the conclusion in the revised manuscript.

- Fig 3 - It would be helpful if a clear breakdown of the samples and their corresponding treatment were provided in a table. In particular, Fig 3b/c – there are several icons (shape with color) that are identical – what do they represent, replicates?

Answer: Thanks for the reviewer’s comments. According to the reviewer’s comments, we have revised Figure 3 to provide a table showing breakdown of the samples and their corresponding treatment. The icons have also been revised in Figure 3 to make the figure clear.

On Figure 3, we write:

“

Fig. 3 Alterations in microbiome diversity during domestication. (a) α -diversity levels of the soil microbiomes. The boxplots show the Shannon indices of the three different soils with different treatments. The letters above the boxplots show significant differences between samples. The color of the boxplots represents the sampling time and treatments. (b) Nonmetric multidimensional scaling (NMS) plots of beta

diversity (Bray–Cutis dissimilarity) by time and treatment. Samples are shaped and color-coded according to the treatments and times. The details of treatments are provided in panel c. (c) Treatments used in the study. (d) Bray–Curtis distances among the microbiomes from different soils at each time point (Days 0-30) decreased with domestication time for the inoculation treatments rather than herbicide treatments. The gray area represents the 95% confidence interval.”

- In all samples methanol was also added – how much of the available carbon did this represent and did it contribute to the converge of taxa observed?

Answer: Thanks for the reviewer’s comments. Methanol was used in the study as it is the solvent of herbicides studied, and this is a common method for herbicide treatments, especially for those with poor water solubility. A total of 2.5 mL methanol was added to 1500 g soils (1.67×10^{-3} mL/g). The soils were then mixed to vaporize the methanol for 2 h. Therefore, very few methanol had been kept in soils and has few contributions to the converge of taxa observed.

- From metagenomic/16S rRNA amplicon analysis, did a common taxa emerge across all enrichments? It would be useful to highlight the core taxa that become abundant and converged overtime.

Answer: Thanks for the reviewer’s comments. The comment taxa emerged (or decreased) across all enrichments have been provided in the revised Figure 5. For examples, 27, 6, and 17 genera with significant change in abundance are shared across treatments for purple, yellow cinnamon, and red soils. Besides, more than 40% of these significantly changed genera are the same among the three different soils with the same treatments, especially for yellow cinnamon soils (> 65%), indicating that genera with significant changes in abundance were common among different soils with the same treatments. Specifically, *Bacillus* and *Sphingobacterium* showed significantly increased abundances in all three soils in the BO groups. The information has been included in the revised manuscript.

On Lines 184-198, we write:

“LEfSe analysis (LDA > 3.0) revealed a total of 133, 54, and 62 bacterial genera with different abundances in the late phase in BO&7D-2&X-1 compared to the early phase in purple, yellow cinnamon, and red soils, respectively (Fig. 5a). Among them, 27, 6, and 17 genera with significant change in abundance were shared across treatments for the three respective soils. ...More than 40% of these significantly changed genera were same among the three different soils with the same treatments, especially for yellow cinnamon soils (> 65%, Fig. 5b), indicating that genera with significant changes in abundance were common among different soils with the same treatments. Specifically, *Bacillus* and *Sphingobacterium* showed significantly increased abundances in all three soils in the BO groups. These results are consistent with the convergent succession of different microbiomes during domestication.”

- Fig 4 - How was the number of genes normalized to sequencing depth?

Answer: Thanks for the reviewer's comments. The method to normalize the number of genes to sequencing depth has been included in the Supplementary Materials.

On Supplementary Materials, we write:

“Metagenomic sequencing

DNA extract was fragmented to an average size of about 400 bp using Covaris M220 (Gene Company Limited, China) for paired-end library construction. Paired-end library was constructed using NEXTFLEX® Rapid DNA-Seq (Bioo Scientific, Austin, TX, USA). Adapters containing the full complement of sequencing primer hybridization sites were ligated to the blunt-end of fragments. Paired-end sequencing was performed on an Illumina NovaSeq 6000 platform at Majorbio Bio-Pharm Technology Co., Ltd. (Shanghai, China) according to the manufacturer's instructions (www.illumina.com). Metagenomics data were assembled using MEGAHIT (<https://github.com/voutcn/megahit>, version 1.1.2).

Open reading frames (ORFs) from each assembled contig were predicted using Prodigal/MetaGene (<http://metagene.cb.k.u-tokyo.ac.jp/>). The predicted ORFs with a length ≥ 100 bp were retrieved and translated into amino acid sequences using the NCBI translation table (http://www.ncbi.nlm.nih.gov/Taxonomy/taxonomyhome.html/index.cgi?chapter=tgen_codes#SG1). A non-redundant gene catalog was constructed using CD-HIT (<http://www.bioinformatics.org/cd-hit/>, version 4.6.1) with 90% sequence identity and 90% coverage. High-quality reads were aligned to the non-redundant gene catalogs to calculate gene abundance with 95% identity using SOAPaligner (<http://soap.genomics.org.cn/>, version 2.21).

Representative sequences of non-redundant gene catalog were aligned to NR database with an e-value cutoff of $1e^{-5}$ using Diamond (<http://www.diamondsearch.org/index.php>, version 0.8.35) for taxonomic annotations. Cluster of orthologous groups of proteins (COG) annotation for the representative sequences was performed using Diamond (<http://www.diamondsearch.org/index.php>, version 0.8.35) against eggNOG database with an e-value cutoff of $1e^{-5}$. The KEGG annotation was conducted using Diamond (<http://www.diamondsearch.org/index.php>, version 0.8.35) against the Kyoto Encyclopedia of Genes and Genomes database (<http://www.genome.jp/keeg/>) with an e-value cutoff of $1e^{-5}$.”

- Fig 4d – what was the change in phyla and taxa from day 0 to day 30? That would be better evidence of convergent succession versus just day 30 day. Can the data be presented at a resolution lower than phyla?

Answer: Thanks for the reviewer's comments. In this study, only 18 soil samples (soil samples from Days 0 and 30 from BO&X-1&7D-2 with three repeats) were selected for metagenomic analysis. So we showed the changes between Day30 and Day0 obtained by metagenome sequencing. The samples used here are less than those used for amplicon sequencing covering whole time period. The figure showing the data at genus/order/class level has been included in the Supplementary Materials.

On Figure S5, we write:

“

Supplementary Fig. S5. Taxonomic classification of the functional genes involved in BO degradation at the order (A), class (B), and genus (C) level. The bar and venn diagrams show taxonomic classification of the functional genes involved in BO degradation as shown in Fig.4b. The results indicate that bacteria with the functional genes involved in BO degradation were shared among different soils after treatments.”

- Fig 5d – it is not clear from Fig5d how the authors concluded that inoculation treatment influenced the microbiome more than profoundly than herbicide treatment. Also, what exactly is meant by “influenced” – similarity/dissimilarity in taxonomic composition?

Answer: Thanks for the reviewer’s comments. In the revised manuscript, Figure 5 and its relative parts have been removed as these are not the main points of the manuscript. More importantly, removing these parts could help to make the manuscript much more concise.

- Fig 6C – what about abundance shifts in soils only inoculated with herbicides?

Answer: Thanks for the reviewer’s comments. We have included the information showing the abundance shifts in soils only inoculated with herbicides (Figure S8). The results showed that most species were not significant changed /or declined. In contrast, these keystone genera were significantly increased in soils inoculated with herbicides and strains as shown in revised Fig. 5C.

On Figure S8, we write:

“

Supplementary Fig. S8. Relative abundances of 18 keystone genera in soils only inoculated with herbicides. The average abundances of each keystone genus in samples at Days 0-9 (early phase) and Days 18-30 (late phase) are shown. ** $P < 0.01$; * $P < 0.05$. P, purple soil; Y, yellow cinnamon soil; R, red soil.”

- Fig 7a - What was the actual concentration of metabolite (not area's) detected in the samples? Can the authors also show measurements from soil control samples without culture and immediately after inoculation?

Answer: Thanks for the reviewer's comments. We have included the actual concentration of metabolite in the revised figure 7. The information for control samples have also been included in revised figure 7. Please see our response to comment 3.

- Fig 8 - What is the physiological rational for these organisms to secrete these compounds?

Answer: Thanks for the reviewer's comments. The results of compound secretion were predicted by modeling, and then verified by experiments. The results showed the advantages of modeling in study microbial metabolic interactions. Until now, experimental determination of metabolic interactions in a microbiome has remained a major challenge, even for simple consortia with two members. Thus, little study focused on the complex metabolic interaction in microbiome. Although the physiological rational for these organisms to secrete these compounds are not clear until now, it is a good start to study it. In the future, we will focused on the physiological rational of compound secretion to get a better understanding of the microbial interactions.

- Figure 7g – what is abundance ratio? Why not show the relative abundance of this strain in the community?

Answer: Thanks for the reviewer's comments. We have changed "abundance ratio" to "biomass ratio" in the revised manuscript. Generally, the biomass decreased or increased with the carbon source decreased or increased for one strain. We assumed that strain abundance in a consortia should also be changed along with the amount change of carbon source, but might in a different way. So the simulation here was focused on the effects of BO concentration on the biomass changes of the consortia 7D-2&X-1. As expected, the relative abundance of two strains in the community increased and decreased along with BO increased or decreased. However, a more interesting result was got when we compared the biomass ratio of strain 7D-2 and X-1 in the consortia, which showed non-linear changes for 7D-2 and X-1. Therefore, we showed the biomass ratio rather than biomass directly in community.

- Page 15, line 306-308 – were transport reactions supported by transporter annotations in the genome? Or was everything allowed to be exchanged? How was this validated?

Answer: Thanks for the reviewer's comments. For raw model, all the transport reactions are supported by transporter annotations in the genome. For model curation, a transport reaction was added when it was supported by experimental data, considering the possibility of imperfect genome annotation. For example, the experimental data shows that the strain can assimilate a compound, and the transport reactions of the compound should be added in the model. In summary, all transport reactions have genomic or experimental support. The information has been included in the Supplementary Materials.

- Fig 8b / S10 – were degradation rates normalized to the amount of microbial cells added? More details on experiments needed. Were these experiments replicated? I don't see any error bars

Answer: Thanks for the reviewer's comments. As described in our response to the major comments 1 above, we have used "residues of herbicides" instead of "degradation rate" in the revised manuscript. All experiments were performed in triplicate, and the error bars has been include in the revised Figure 7 (the original Figure 8).

Reviewer #2 (Remarks to the Author):

In this manuscript, the authors use a structured approach for targeted microbiome application based on domesticating bacterial communities from natural soil samples and use in-silico approaches to optimize the functional value of the derived synthetic microbiomes. They use different treatments to isolate the influence domesticating the microbial consortium either via addition of two different herbicides or the inoculation of bacteria with desired functions (as individual species, competing species or synergistic species). They further use

LEfSe analysis to identify keystone species in the soil samples and use a compartmentalized community FBA approach to evaluate the metabolic flux exchange between different members of the community. As I am no expert on bacterial molecular approaches and diversity, I cannot assess the approach and quality of the actual metagenomic analysis.

Overall, this is a potentially impactful paper that would benefit from clarification about the data and treatments that have been used for specific Figures etc. as disentangling the actual findings is otherwise cumbersome.

Answer: Thanks for the reviewer's comments. We have revised the manuscript thoroughly to make the manuscript more concise and focused, and to clarify the data and treatment, which is much easier to understand.

My main concern with the experimental system is the high addition of inoculated cells. Adding 2×10^8 CFU/g soil for each species ten times over 30 days is a total of 4×10^9 CFU/g soil. Considering that typical numbers in soil are 10¹⁰ cells/g soil, this results in a replacement of 40% of the soil community. I fear that this high number of added cells biases the degradation efficiency of the communities (Figure 2), governs the loss of diversity (Figure 3 panel a), alters the calculated Bray-Curtis distance (Figure 4) and artificially enriches the abundance of functional genes (Figure 4 panel c). The question then becomes if this is a true domestication, i.e. will this approach be suitable for real in-situ bioremediation using the functional community since the amount of applied bacteria needed to remediate a single ton of soil would be astronomical. Testing the success of inoculating a lower abundance of cells would be an interesting avenue and/or if a single, initial inoculation is sufficient to stimulate the growth of the degrading bacteria.

Answer: Thanks for the reviewer's comments. In response, we would like to show the results of our preliminary experiment to determine the experimental system. In this study, the aim of the domestication experiment is to obtain the in-situ strains that interacting with degraders (i.e. supporting strains), which provide candidates for constructing synthetic microbiomes. Actually, the functional supporting strains in soils are not widespread, and most strains in soils are not the focus in this study. Based on this purpose, we need to find a strategy with high domestication strength to reduce time and accelerate microbiome designing. Thus, we compared the performances of single and repeated domestication, as well as low- and high-dose of inoculation. Not surprisingly, the results showed that treatments with high-dose of inoculation repeatedly had more remarkable influence on bacterial community and showed much higher BO-degrading ability. Therefore, we used the domestication strategy of repeated inoculation of degrading strains with high dose to decrease the domestication time (Fig. S2, S3).

We agree with reviewer that high number of added cells might bias the degradation efficiency of the communities, governs the loss of diversity, alters the calculated Bray-Curtis distance and artificially enriches the abundance of functional genes. However, in this study, the degradation efficiency, and microbial community diversity

and abundance of functional genes were detected to identify the alteration caused by our degradation treatment to select supporting strains. On this background, the precise degradation process was not the focus of the study. Therefore, the biases caused by the high number of added cells do not influence the main conclusion of the study. On the contrary, the biases indicated the high efficiency of our domestication system, showing its application potential.

We also agree with the reviewer that it is not suitable for real bioremediation if high amount of applied bacteria is needed. Actually, it is just the difficulty and challenge for almost all the bioremediation process, as the exogenous degrader are usually failed to thrive in soils. The aim of the study is trying to solve this problem by using a synthetic microbiome instead of the single strains. We used the high amount of degraders in domestication process (rather than real bioremediation) to identify supporting strains, not in the true bioremediation process. Here, we used synthetic microbiome for real in-situ bioremediation. By construct a synthetic microbiome with degraders and supporting strains, we showed that the synthetic microbiome could improve the degrading efficiency. The information has been included in the revised manuscript.

On Lines 130-135, we write:

“To obtain effective domestication strategy, we compared the performances of single and repeated domestication, as well as low- and high-dose of inoculation (Fig. S2, S3). Treatments with high-dose of inoculation repeatedly had more remarkable influence on bacterial community and showed much higher BO-degrading ability compared to other treatments. Therefore, we used the domestication strategy of repeated inoculation of degrading trains with high dose to decrease the domestication time.”

On Figure S2, we write:

“

Supplementary Fig. S2. *Degradation ability of bromoxynil octanoate (BO) at day 30 by different treated microbiomes. Control, only regular water spray; Domestication once: inoculating two strains X-1 and 7D-2 at a ratio of 1:1, with a final concentration of approximately 2×10^6 CFU/g soil for each strain, and adding 5 mg/kg of BO only once; Continuous domestication (low): BO (5 mg/kg) and inoculating strains (2×10^6 CFU/g soil) were added repeatedly every three days for 10 times; Continuous domestication (high): BO (5 mg/kg) and inoculating strains (2×10^8 CFU/g soil) were added repeatedly every three days for 10 times.”*

On Figure S3, we write:

Supplementary Fig. S3. Microbial diversity and community composition during domestication with different treatments. (a) α -diversity of the soil microbiomes. Inoculating once, inoculating two strains X-1 and 7D-2 at a ratio of 1:1, with a final concentration of approximately 2×10^6 CFU/g soil for each strain, and adding 5 mg/kg of BO only once; Continuous inoculation, BO (5 mg/kg) and inoculating strains (2×10^6 CFU/g soil) were added repeatedly every three days for 10 times. (b) Principal coordinates analysis (PCoA) with Bray–Curtis distances of bacterial communities. No significant clusters were detected for treatments and times. (c) Heatmaps showing relative abundances of genera (left) and families (right) identified in soils with different treatments. No significant difference was detected among treatments and times.”

Throughout the manuscript, the authors talk about the degradation rate of the herbicide by the different treatment approaches. However, from Figure 2 (where the unit on the y-axis is %) and from the methods, I assume that the unit in this Figure is the percentage of remaining herbicide from the last application. In this case, defining this as a degradation rate (which should be a quantity per time) is misleading and may result in quite some confusion for the readers. I would suggest to use a bar plots instead of connected lines (they are not directly linked as there is continuous reapplication of species and herbicide).

Answer: Thanks for the reviewer's comments. According to the reviewer's suggestions, we have used a bar plots rather than line plot in the revised Figure 2. We have also used "residues of herbicides" instead of "degradation rate" in the revised Figure 2. Besides, to make the treatments and data more clearly, we have revised Methods and Supplementary Materials to provide the details of methods to determine the residues of herbicides.

On Lines 427-429, we write:

"0.5 g of soil was collected to assess the degrading ability of BO or DBHB with high-performance liquid chromatography (HPLC). The detection methods are described in the Supplementary Materials"

On Supplementary Materials, we write:

"To detect the degrading ability of BO or DBHB by microbial community with different treatments, 0.5 g of soil was collected from different treatments and brought into fresh MM media supplemented with BO or DBHB (50 mg/L). After incubating on a shaker at 30 °C for 10 h (160 rpm), 1 mL of the liquid medium was taken from the medium to detect the residues of BO, bromoxynil or DBHB. All treatments were performed in triplicate by HPLC."

Next to the overall very high application of bacterial cells every three days, a second complication when interpreting the diversity results (Figure 3 panel a) and Bray-Curtis distances (Figure 3 c&d) stems from the uncertainty in the overall diversity loss due to the change in conditions simply based on the experimental system. As outlined in the methods section, the experiments were prepared by grinding the soil and transferring into plastic pots where the hydration conditions are kept constant at 40-60% water holding capacity. An important control for this experiment would be to look at the change in diversity whendded to the pots. Such a drastic change in conditions for bacterial cells can automatically result in a loss of diversity and disentangling the actual effect of domestication from the application of species/disruption in their habitat is difficult. Figure 3 panel a shows that there was a control at T0 (I assume before application of the treatments) but this is then not repeated for the subsequent time points.

Answer: Thanks for the reviewer's comments. We agree with the reviewer that the change of bacterial community diversity might result in the pot experiments and might disentangling the actual effect of domestication. However, based on our

preliminary experiment (as shown in our response above), the treatments with low-dose of inoculation repeatedly and high-dose of inoculation once caused slight changes in community diversity compared to the initial community. We detected significant changes for treatments with high-dose of inoculation repeatedly. These results showed that the main force driving changes in bacterial community was the treatments with high-dose of inoculation repeatedly, rather than the pot experiments alone.

Consistently, when compared between herbicide application vs. herbicide application & strain inoculation, we found that herbicide application & strain inoculation had more remarkable influence on bacterial community. Besides, we got similar results from three different soils. Therefore, it is safe to say that out domestication process caused significantly changes in bacterial community.

We have tried to repeat the control sample T0 for the subsequent time points. However, the original samples were collected and treated in 2018, and it is not possible to obtain the same sample as the bacterial community changes along with time. So we don't include the repeated control T0 sample in the revised manuscript.

Finally, the authors do not include the information about which treatments are used in the mapping of functional genes (Figure 4 panel c). In case this is for the treatments where different individual strains, the competitive or synergistic community is added over time, I am not surprised that the final community contains far higher abundance of these function genes. Due to the very high application of inoculated cells, I believe the data that would support the domestication of a functional community should be taken from the treatment where only herbicide is added to the soil.

Answer: Thanks for the reviewer's comments. We have included the information about treatments used in the mapping of functional genes in the revised Figure 4. Same to the reviewer, we are also not surprised that the final community contains far higher abundance of these function genes as we inoculated degraders in soils. In response, we would like to describe our purpose of this research. We would like to construct a synthetic microbiome with degraders and supporting strains. Considering the in situ degraders are usually difficult (even impossible) to isolate, we used exogenous degraders for bioremediation. To improve the survival and degrading efficiency, in situ supporting strains were used for construct synthetic microbiome. We then screened the in situ strains that might interact with degraders when herbicides exist in soils as candidates. These strains were got sequencing results showing the abundance changes along with degraders during domestication process. These candidates were then used for modeling and optimal synthetic microbiome construction. Therefore, we chose the functional community from the treatment with combination of herbicide and degrader inoculation.

On Lines 444-445, we write:

“A total of 18 soil samples (soil samples from Days 0 and 30 from BO&X-1&7D-2 with three repeats) were selected for metagenomic analysis.”

On Lines 847-848, we write:

“...0 and 30 d, samples collected at Days 0 and 30 from treatments of BO&X-1 & 7D-2, respectively.”

Smaller comments:

Line 21: Long sentence – consider rephrasing.

Answer: Thanks for the reviewer's comments. The sentence has been rephrased.

On Lines 24-27, we write:

“However, engineering natural microbiomes, the basis of the application of microbiomes, still remain challenging. The obstacles include metabolic interactions within microbiomes are largely unknown, and practical principles and tools for microbiome engineering are still lacking.”

Line 38: The phrase suggests that human health is a biogeochemical cycle?

Answer: Thanks for the reviewer's comments. The sentence has been rephrased.

On Lines 41-43, we write:

“Microbiomes are ubiquitous in nature and play important roles in almost all biogeochemical cycles occurring on this planet, such as metabolism of nutrients¹⁻³, agriculture^{4,5}, food fermentation^{6,7}, element cycling^{8,9}, biofuels¹⁰⁻¹², and pollutant degradation¹³⁻¹⁵.”

Line 52: Sentence is somewhat convoluted.

Answer: Thanks for the reviewer's comments. The sentence has been rephrased.

On Line 55, we write:

“Bottom-up and top-down strategies have been proposed to engineer microbiomes²⁴.”

Line 207: Important to state which treatments were used to generate Figure 4c.

Answer: Thanks for the reviewer's comments. The information has been included.

On Lines 847-848, we write:

“0 and 30 d, samples collected at Days 0 and 30 from treatments of BO&X-1 & 7D-2, respectively.”

Line 555: What is the effect of using these different formulations of biomass parametrizations?

Answer: Thanks for the reviewer's comments. To make superCC user-friendly, we provide four commonly used scenarios. The four commonly used scenarios could meet the needs of most users. For example, the users could chose 1 or 2 for designing the optimal microbial community, and chose 3 for computational cell design based on mimicking the metabolic network of functional microbiomes. Professional users might try scenario 4 to defined parameters they like.

On Supplementary Materials, we write:

“We provide four commonly used scenarios, including 1) equal abundance for each organism; 2) no limitation for any organisms, meaning that the biomass of each organism is allowed to be zero; 3) defining biomass of a specific organism as

community biomass (used for identifying organisms in a community that could improve the growth of the target organism); and 4) any defined abundances.”

Line 589: When inoculating the strains for metabolite exchange measurements, were the single cell dilutions (OD600 of 0.35) washed before co-inoculation?

Answer: Thanks for the reviewer’s comments. The single cell was washed three times with sterile water before co-inoculation.

On Line 521, we write:

“..., washed three times with sterile water...”

Line 626: I could not find the optimized Cb models on the provided GitHub page.

Answer: Thanks for the reviewer’s comments. The model files have been provided in Supplementary Materials.

Line 947: The statement that the Bray-Curtis distances are increasing with time is somewhat misleading. The overall correlation across all soils is rather weak and for some soils there is a clear decrease (e.g. Y in DBHB&7D-2&H8).

Answer: Thanks for the reviewer’s comments. The statement and Figure 4d have been removed in the revised manuscript to make the manuscript more concise and focused, as they are not the main points.

Line 958: There is no information on which treatments were used for the generation of panel C. Is this a combination of all? Or only a subset? How do the treatments where domestication was performed by addition of only herbicide compare to domestication via addition of specific strains?

Answer: Thanks for the reviewer’s comments. The information has been included in the revised Figure 4. In revised Figure 4, we compared samples collected at Days 0 and 30 from treatments of BO&X-1&7D-2.

On Lines 444-445, we write:

“A total of 18 soil samples (soil samples from Days 0 and 30 from BO&X-1&7D-2 with three repeats) were selected for metagenomic analysis.”

On Lines 847-848, we write:

“...0 and 30 d, samples collected at Days 0 and 30 from treatments of BO&X-1&7D-2, respectively.”

Comments on Figures:

In many figures, purple is misspelled as “pruple”

Answer: Thanks for the reviewer’s comments. All the figures have been checked and revised thoroughly.

Figure 2: The connected lines and designation as “degradation rate” is somewhat misleading. Present as bar plots?

Answer: Thanks for the reviewer's comments. The bar plots has been used in Figure 2.

Figure 3: Most of the datapoints are outside of the 95% CI in the linear fits. Should these be done for the individual soils? (The trend between soils are often different).

Answer: Thanks for the reviewer's comments. According to the reviewer's suggestions, we have revised the figure to show all the samples rather than individual soils.

Figure 8: What is the unit of biomass?

Answer: Thanks for the reviewer's comments. We have included the unit of biomass and compound concentrations of modeling in the revised manuscript.

On Line 874, we write:

"...Predicted biomass (1/h) of keystone combinations."

On Supplementary Materials, we write:

"The biomass functions are weighted combinations of molecules that are required for cellular growth and reproduction and are scaled such that the units are 1/h; concentrations are expressed in units of mmol /gDW. The community biomass was defined as linear combinations of biomass of each individual species."

Reviewer #1 (Remarks to the Author):

I thank the authors for their detailed response. An outstanding issue I still have is the lack of a basic physiological explanation for the proposed mutualism. For example, in Figures 6A and 6C, the authors show X-1 and 7D-2 degrading Bromoxynil octanoate to HBr and DBHB to HBr, respectively. How does each cell generate energy for biomass growth under these scenarios? Moreover, during the proposed mutualism, how does the model predict growth of each member is actually improved (i.e., why would an organism secrete say glutamate or mannose, which would be very costly)?

It would also be beneficial to show a basic comparison with the model and experiments in the main text (if not provided) on the growth rate and max OD of each organism growing separately versus in co-culture, together with the impact of pollutant degradation. This would be a simple and clear way of illustrating the benefits of the mutualism.

Finally, it would be very helpful to provide more context on what is known about mutualism in soil microbiomes – for example, are there previous examples that detail mutualistic bacterial interactions in soils and how they benefit productivity or pollutant degradation? How do the interactions proposed in the manuscript here compare?

Reviewer #2 (Remarks to the Author):

I commend the authors for the additional experimental and analysis work on this manuscript. There are considerable improvements concerning clarification in the experimental procedures and data presentation. However, I still fear that the major driver of most observed results and conclusions stems from the high application of bacteria and not true domestication. I do not agree with the statement that “the biases caused by the high number of added cells do not influence the main conclusion of the study. On the contrary, the biases indicated the high efficiency of our domestication system, showing its application potential.” However, I believe that during the top-down part of the manuscript, it is simply important to clarify that a lot of the degradation is driven by the inoculated species (which the authors do in one of the sentences), and no further experimental data is required.

Some suggestions (first from the response to authors, then general):

- The authors have recalculated the metric from degradation rate in percent to concentrations which I applaud. However, why is there a lot of BO at the start of the experiment for the control (where no BO is added at all), same for Figure 2 groups 2 (where bacteria are added but no BO/DBHB)? In the methods, it states that 5 mg/kg is added, whereas all experiments show around 30 to 50 mg/kg residual DBHB or BO – could the authors clarify?
- SI Figure 2: The authors reverted back to the degradation rate as a percentage. I would change this again to the remaining herbicide as a concentration or degradation efficiency in percentage. Also, would the correct control experiments here not be a single inoculation but the same BO application and no inoculation but the same BO application?
- In the initial paragraph of the results, introduce what the different strains are doing metabolically (i.e., why does BO need a synergistic community to degrade?). This will also help to understand Fig. 4b.
- The functional genes in Fig. 4 come from a great diversity of species/taxonomic classification. Is there a way to quantify how many actual cells in the day 30 samples come from the different taxons? And see how many are NOT from the inoculated proteobacteria? This would be interesting to see how much of the degradation is driven by the enriched, resident community.
- I would include a definition of what the authors mean by “domestication of microbiomes” in the introduction. This is important to understand the remaining manuscript. For macro-flora and fauna, there are three typical pathways of domestication, but a definition of the actual process would be good since it is used a lot.
- Generally, I would add one paragraph to the discussion about the high inoculation rate and how this drives the community succession etc. in the initial top-down approach. I agree with the authors that this does not implicate the remaining aspect of the manuscript (bottom-up

community design and validation) but could raise questions from the readership.

We thank the editorial team and the referees for the positive feedback and the constructive report for our manuscript entitled "Engineering natural microbiomes toward enhanced bioremediation by microbiome modeling" (NCOMMS-22-11760B). In the revised version, we now address all the concerns raised by the referees. Please find below a point-by-point response providing a full account of our modifications (bold font style represents comments of reviewers, our response follows, and revised text is brought in italic where relevant). We have also uploaded a file which compares the original submission with the revised one.

REVIEWER COMMENTS

Reviewer #1 (Remarks to the Author):

I thank the authors for their detailed response. An outstanding issue I still have is the lack of a basic physiological explanation for the proposed mutualism. For example, in Figures 6A and 6C, the authors show X-1 and 7D-2 degrading Bromoxynil octanoate to HBr and DBHB to HBr, respectively. How does each cell generate energy for biomass growth under these scenarios? Moreover, during the proposed mutualism, how does the model predict growth of each member is actually improved (i.e., why would an organism secrete say glutamate or mannose, which would be very costly)?

Answer: Thanks for the reviewer's comments. In the revised manuscript, we have included the physiological explanation for the proposed mutualism. We have provided results of additional experiments showing more detailed mutualism between X-1 and 7D-2 (Fig. 6 and Supplementary Fig. 13). The strain X-1 only transformed BO to bromoxynil but it was not capable of degrading bromoxynil, and X-1 could not grow using BO as the sole carbon source. The strain 7D-2 could degrade bromoxynil, but it was unable to degrade BO into bromoxynil. Therefore, 7D-2 could not grow using BO as the sole carbon source, which was same to X-1. However, the synergistic consortium of X-1 and 7D-2 could degrade BO completely by metabolic cooperation, and both strains grew well using BO as the sole carbon source. These results showed that 7D-2 generated energy by assimilating bromoxynil which was produced by X-1, while X-1 generated energy by assimilating metabolites which was produced by 7D-2. We have also included predictions by modeling showing that 7D-2 and X-1 could not grow using BO as the sole carbon source while the co-cultures could grow with the same medium (Supplementary Fig. 11). The predicted results were consistent with experimental results, revealing mutualism between 7D-2 and X-1 enabled both strains to grow in the medium where growth was feasible for single-cultures. By modeling, we further

illustrated details of exchanged metabolites between X-1 and 7D-2. The results showed 7D-2 assimilated bromoxynil, hypoxanthine, D-proline, and D-Glucosamine produced by X-1, and X-1 assimilated xanthine, D-mannose and L-glutamate produced by 7D-2. All these predicted results were verified experimentally. Besides, similar experiments were also performed for the interactions between 7D-2 and H8 to explain their interactions physiologically (Fig. 6). We showed that both 7D-2 and H8 could degrade DBHB and grow using DBHB as the sole carbon source. Besides, the combination of 7D-2 and H8 did not improve the DBHB degradation or biomass, showing the competitive relation for DBHB assimilation. We have also included the BO-degradation pathway by X-1 and 7D-2, to make their relationship easy to understand. The information has been included in the revised manuscript.

We agree with the reviewer that secreting metabolites such as glutamate or mannose are costly. In the revised manuscript, we have included additional results obtained by RNA-seq of 40 samples, which explored the possible causes of secretion of these costly metabolites in the mutualism between 7D-2 and X-1 (Supplementary Fig. 13). The samples included single-cultured X-1 and 7D-2 in medium with/without herbicide, co-cultured X-1 and 7D-2 in medium with/without herbicide, and single-cultured X-1 and 7D-2 in medium with/without exchanged metabolites. For X-1, we found that the gene expression variation was induced by herbicide rather than exchanged metabolites or co-cultures, indicating these metabolites might be by-products during the BO-degradation. In contrast to X-1, the gene expression variation in 7D-2 was mainly induced by both exchanged metabolites and co-cultures instead of herbicide, indicating the metabolite secretion by 7D-2 might be specifically induced. These results showed one possible explanation for secretion of glutamate or mannose by 7D-2, which was that 7D-2 adapted to the co-cultures by regulating gene expression. The regulation of gene expression by 7D-2 enabled it surviving in the medium where growth was feasible for single-cultures. The information has been included in the revised manuscript.

On Lines 213-222, we write:

“The strain X-1 only transformed BO to bromoxynil, but it was not capable of degrading bromoxynil (Fig. 6a). Besides, the strain X-1 could not grow using BO or bromoxynil as the sole carbon source (Fig. 6b, c). The strain 7D-2 could degrade bromoxynil, but it was unable to degrade BO into bromoxynil (Fig. 6a, f). Therefore, the strain 7D-2 could not grow using BO as the sole carbon source, but it could grow using bromoxynil as the sole carbon source (Fig. 6b, c). However, the synergistic consortium of X-1 and 7D-2 could degrade BO completely by metabolic cooperation (Fig. 6a, f), and both strains grew well using BO as the sole carbon source (Fig. 6b). Besides, both strains 7D-2 and H8 could degrade DBHB and grew using DBHB as the sole carbon source

(Fig. 6d, e). However, the combination of 7D-2 and H8 did not improve the DBHB-degrading efficiency or biomass of the consortium (Fig. 6d, e).”

On Lines 225-227, we write:

“By modeling, we showed that the strains 7D-2 and X-1 could not grow using BO as the sole carbon source while the co-culture could grow with the same medium (Supplementary Fig. 11), which was consistent with experimental results.”

On Lines 251-260, we write:

“To explore possible physiological explanations for mutualism between strains X-1 and 7D-2, gene expression profiling of the two strains was compared among single-cultures with herbicide, single-cultures with exchanged metabolites, and co-cultures with herbicide (Supplementary Fig. 13). For the strain X-1, the gene expression variation was mainly induced by BO rather than exchanged metabolites or co-cultures (Supplementary Fig. 13), indicating these exchanged metabolites might be by-products during BO-degradation by X-1. In contrast to X-1, the gene expression variation in 7D-2 was mainly induced by both exchanged metabolites and co-cultures instead of herbicide (Supplementary Fig. 13), indicating the metabolite secretion by 7D-2 might be specifically induced. These results showed that the strain 7D-2 adapted to co-cultures by regulating gene expression, which enabled it surviving in the medium where growth was feasible for single-cultures.”

On Supplementary Methods we write:

“Transcriptome analysis

To explore possible physiological explanations for mutualism between X-1 and 7D-2, gene expression profiling of the two strains was explored by RNA-seq. The analysis was performed in 10 treatments with four repeats including a total of 40 samples. For X-1, we pre-cultured the strain in LB medium containing 50 mg/L BO for 24 h to get a consistent metabolic state for all cells. After that, the pre-cultured X-1 was cultured in a new LB medium ($OD_{600} = 0.2$) containing 50 mg/L BO or exchanged metabolites (i.e. xanthine, mannose, and glutamate) for 1 h to activate gene expressions. The cells were collected by centrifugation and then treated with various media for another 1.5 h. For 7D-2, we pre-cultured the strain in LB medium containing 50 mg/L bromoxynil for 24 h. Similar to X-1, cells of 7D-2 were first cultured in a new LB medium ($OD_{600} = 0.2$) containing 50 mg/L bromoxynil or exchanged metabolites (i.e. hypoxanthine, proline, glucosamine, and bromoxynil) for 1 h, and the cells were then collected and treated with various media for another 1.5 h. The treatments were set as follows: i) single-cultures with/without herbicide, including (1) X-1 first growing in MM medium containing BO then in new MM medium without BO, (2) X-1 first growing in MM medium containing BO then in new MM medium with BO, (3) 7D-2 first growing in MM medium containing bromoxynil then in new MM medium without bromoxynil, (4) 7D-2 first growing in MM medium containing bromoxynil then in new MM medium

with bromoxynil; ii) single-cultures with/without exchanged metabolites, including (5) X-1 first growing in MM medium containing exchanged metabolites then in new MM medium without exchanged metabolites, (6) X-1 first growing in MM medium containing exchanged metabolites then in new MM medium with exchanged metabolites, (7) 7D-2 first growing in MM medium containing exchanged metabolites then in new MM medium without exchanged metabolites, (8) 7D-2 first growing in MM m medium containing exchanged metabolites then in new MM medium with exchanged metabolites; and iii) co-cultures with/without herbicide, including (9) X-1 and 7D-2 first co-culturing in MM medium containing BO then in MM medium without BO, and (10) X-1 and 7D-2 first co-culturing in MM medium containing BO then in MM medium with BO. The samples were finally centrifuged, frozen in liquid nitrogen for 10 minutes, and stored at -80 °C.

Total RNA was extracted from the tissue using TRIzol® Reagent according the manufacturer's instructions (Invitrogen) and genomic DNA was removed using DNase I (TaKara). RNA quality was determined using 2100 Bioanalyser (Agilent) and quantified using the ND-2000 (NanoDrop Technologies). High-quality RNA samples were used to construct sequencing libraries.

RNA-seq strand-specific libraries were prepared following TruSeq RNA sample preparation Kit from Illumina (San Diego, CA), using 5µg of total RNA. Libraries were size selected for cDNA target fragments of 200-300 bp on 2% Low Range Ultra Agarose followed by PCR amplified using Phusion DNA polymerase (NEB) for 15 PCR cycles. After quantified by TBS380, paired-end libraries were sequenced by Illumina NovaSeq 6000 sequencing platform at Shanghai Biozeron Biological Technology Co., Ltd (Shanghai, China). The raw reads were trimmed and quality controlled by Trimmomatic (version 0.36). Then clean reads were separately aligned to reference genome with orientation mode using Rockhopper software which was also used to calculate gene expression levels with default parameters.

To identify significantly differential expression genes (DEGs), the expression level for each transcript was calculated using transcripts per kilobase million (TPM) method. edgeR was used for differential expression analysis. The DEGs between two treatments were selected using the following criteria: i) the logarithmic of fold change was greater than 2 and the false discovery rate (FDR) should be less than 0.05. To understand the functions of DEGs, KEGG pathway analysis were carried out by KOBAS (<http://kobas.cbi.pku.edu.cn/kobas3>).

On Fig. 6, we write:

“

Fig. 6 Simulations and experimental validations of metabolic interactions in two-member consortia. a Comparison of the degradation ability of BO by individual strains of 7D-2 or X-1 versus in co-cultures. The concentrations of BO and bromoxynil produced during BO degradation in the medium with BO as the sole carbon source are showed. **b, c** Comparison of cell growth of 7D-2 and X-1 growing separately versus in co-culture. The log transformed colony-forming unit (CFU) of each strain cultured in medium with BO (b) or bromoxynil (c) as the sole carbon source is showed. **d, e** The residues of DBHB (d) and cell biomass (e) during DBHB degradation by individual strains of 7D-2 and H8 or co-cultures. **f** BO degradation pathway by strains X-1, 7D-2 and H8. **g** Prediction of exchanged metabolites between strains 7D-2 and X-1....”

On Supplementary Fig. 11, we write:

“

Supplementary Fig. 11. Simulations of the biomass of strain X-1, 7D-2 or H8 growing separately versus in co-cultures (X-1&7D-2 or 7D-2&H8). The simulations were performed in the BO or DBHB&NH₄⁺ medium.”

On Supplementary Fig. 13, we write:

“

Supplementary Fig. 13. Comparison of significantly up-regulated genes in strains 7D-2 and H8 with various treatments. *a* Number of significantly up-regulated genes in strain X-1. X-1_BO, comparing X-1 growing with BO versus those without BO; X-1_M, comparing X-1 growing with exchanged metabolites versus those without exchanged metabolites; X-1_Com, comparing X-1 growing in co-cultures with 7D-2 in medium with BO versus those in medium without BO. *b* Number of significantly up-regulated genes in strain 7D-2. 7D-2_BO, comparing 7D-2 growing with bromoxynil versus those without bromoxynil; 7D-2_M, comparing 7D-2 growing with exchanged metabolites versus those without exchanged metabolites; 7D-2_Com, comparing 7D-2 growing in co-cultures with X-1 in medium with BO versus those in medium without BO. *c* Comparison of the significantly up-regulated genes from (a) based on KEGG annotation. *d* Comparison of the significantly up-regulated genes from (b) based on KEGG annotation. The numbers of annotated KEGG pathway are showed.”

It would also be beneficial to show a basic comparison with the model and experiments in the main text (if not provided) on the growth rate and max OD of each organism growing separately versus in co-culture, together with the impact of pollutant degradation. This would be a simple and clear way of illustrating the benefits of the mutualism.

Answer: Thanks for the reviewer’s comments. In the revised manuscript, we have

included the comparisons between model predictions and experiments on the growth and pollutant degradation of each organism growing separately versus in co-culture (Fig. 6a-e and Supplementary Fig. 11). We have also included a schematic diagram showing the metabolic cooperation for BO degradation (Fig. 6f). Please see our response to Comment 1 above.

Finally, it would be very helpful to provide more context on what is known about mutualism in soil microbiomes – for example, are there previous examples that detail mutualistic bacterial interactions in soils and how they benefit productivity or pollutant degradation? How do the interactions proposed in the manuscript here compare?

Answer: Thanks for the reviewer's comments. According to the reviewer's suggestions, we have included a paragraph in Discussion to provide the context on mutualism in soil microbiomes, and highlight functions of mutualistic bacterial interactions in improving survival and promoting pollutant degradation. We show that metabolic exchanges are ubiquitous in natural microbial communities, and metabolic modeling is a tool with application potential in exploring mutualism in natural environments which has been increasingly used recently.

On Lines 419-430, we write:

“Currently, the view that antagonistic interactions are ecologically more important than synergistic interactions (such as mutualism) in microbial communities is widely held⁷³. However, the function of microbial mutualism in natural environments might be underestimated as metabolic exchanges are difficult to assess within natural systems^{73,74}. In the present study, metabolic exchanges were detected between herbicide-degraders and keystone strains in soils by metabolic modeling, showing the application potential of metabolic modeling in exploring mutualism in natural environments. Recently, metabolic modeling has been increasingly used to explore microbial metabolic interactions in both free-living and host-associated natural communities⁷⁵⁻⁷⁷. These studies showed metabolic exchanges are ubiquitous in natural microbial communities⁷³, which is consistent with our results. Besides, we showed the metabolic exchanges enhanced pollutant-biodegrading capability of microbial communities. Similar results have been detected, showing cross-feeding in microbial communities not only improves survival but also promotes pollutant degradation⁷⁸⁻⁸⁰.”

Reference:

73. Kost, C., Patil, K. R., Friedman, J., Garcia, S. L. & Ralser, M. Metabolic exchanges are ubiquitous in natural microbial communities. *Nat. Microbiol.* **8**, 2244–2252 (2023).
74. LaSarre, B., McCully, A. L., Lennon, J. T. & McKinlay, J. B. Microbial mutualism dynamics governed by dose-dependent toxicity of cross-fed nutrients. *ISME J.* **11**, 337–348

- (2017).
75. Schäfer, M. *et al.* Metabolic interaction models recapitulate leaf microbiota ecology. *Science* **381**, eadf5121 (2023).
 76. Yu, J. S. L. *et al.* Microbial communities form rich extracellular metabolomes that foster metabolic interactions and promote drug tolerance. *Nat. Microbiol.* **7**, 542–555 (2022).
 77. Ryback, B., Bortfeld-Miller, M. & Vorholt, J. A. Metabolic adaptation to vitamin auxotrophy by leaf-associated bacteria. *ISME J.* **16**, 2712–2724 (2022).
 78. Ge, Z. B. *et al.* Two-tiered mutualism improves survival and competitiveness of cross-feeding soil bacteria. *ISME J.* **17**, 2090–2102 (2023).
 79. Wang, X. *et al.* Nitrogen transfer and cross-feeding between *Azotobacter chroococcum* and *Paracoccus aminovorans* promotes pyrene degradation. *ISME J.* **17**, 2169–2181 (2023).
 80. Zhao, Y. *et al.* Inter-bacterial mutualism promoted by public goods in a system characterized by deterministic temperature variation. *Nat. Commun.* **14**, 5394 (2023).

Reviewer #2 (Remarks to the Author):

I commend the authors for the additional experimental and analysis work on this manuscript. There are considerable improvements concerning clarification in the experimental procedures and data presentation. However, I still fear that the major driver of most observed results and conclusions stems from the high application of bacteria and not true domestication. I do not agree with the statement that “the biases caused by the high number of added cells do not influence the main conclusion of the study. On the contrary, the biases indicated the high efficiency of our domestication system, showing its application potential.” However, I believe that during the top-down part of the manuscript, it is simply important to clarify that a lot of the degradation is driven by the inoculated species (which the authors do in one of the sentences), and no further experimental data is required.

Answer: Thanks very much for the reviewer’s comments pointing out the improper use of “domestication”. Based on the reviewer’s suggestions, we compared the meaning of domestication for macro-flora/fauna and “domestication of microbiomes” here. The three typical pathways of domestication for macro-flora and fauna, including the commensal, prey, and directed pathways, are totally different to the “domestication of microbiomes” in the present study. The “domestication of microbiomes” here was intended to refer to “reassembly of microbial community after treatments of herbicide application and/or inoculation of herbicide-degrader”, and “domestication of microbiomes” here was just a description of reassembly process of microbial

communities, rather than the causes of reassembly. Therefore, the “domestication” was improperly used for description the process of reassembly process of microbial communities, which resulted in misunderstanding that “domestication” was the drivers of microbial community reassembly. In the revised manuscript, what we have shown is that exogenous application of herbicide and herbicide-degrader inoculation leads to the reassembly of microbial community, which is exactly same to the reviewer’s opinion. Consistently, many other studies have reported nutrition- or inoculation-driven reassembly of natural microbial communities, indicating that reassembly might be a common feature of microbiome under certain selective pressures. To avoid misunderstanding, we have removed the expression of “domestication of microbiomes” in the revised manuscript. Instead, we have used “application of herbicide and herbicide-degrader inoculation”, “herbicide application and herbicide-degrader inoculation”, “herbicide and inoculum applications”, “herbicide and inoculation treatments” or “treatments” in proper places.

We would like to explain the statement mentioned by the reviewer. In this study, we aimed to identify keystone strains for degraders to improve the performance of degraders by constructing a bacterial community. Bioaugmentation of organic contaminated soils is a commonly used strategy (please see the review by Gao et al. below), and the high-dose of inoculation ($>10^8$) has been frequently used for the bioaugmentation treatments (please see two examples below by Moreno-Forero et al. and Fida et al.) for high efficiency of degradation. Although few studies have focused on the effects of inoculum concentrations, we showed that the effects of inoculation on natural microbiomes seems to be dose-dependent (please see the discussion about the high inoculation rate in our response to the last Comments). Besides, the high-dose of inoculation could help to save the experimental time compared to the low-dose. Under this background, we got conclusion that the high-dose of inoculation was a practical solution for detecting the keystone strains in a relatively short time. We agreed with the review that the high-dose of inoculation might bring in biases of microbial community diversity and functions. Here, we made a trade-off between the experimental time and fineness. We successfully detected and isolated keystone strains, showing that the trade-off strategy worked.

Reference:

- Gao, D., *et al.* Current and emerging trends in bioaugmentation of organic contaminated soils: A review. *J. Environ. Manage.* **320**, 115799 (2022).
- Moreno-Forero S. K., *et al.* Genome-wide analysis of *Sphingomonas wittichii* RW1 behaviour during inoculation and growth in contaminated sand. *ISME J.* **9**, 150–165 (2015).
- Fida, T. T., *et al.* Physiological and transcriptome response of the polycyclic aromatic hydrocarbon degrading *Novosphingobium* sp. LH128 after inoculation in soil. *Environ. Sci. Technol.* **51**, 1570–1579 (2017)

According to the reviewer's suggestion, we have clarified that most of the degradation was driven by the inoculated species. Besides, by analyzing the taxonomic distribution of the key enzymes involved in BO degradation (Supplementary Fig. 6), we showed that 52%-100% of nitrilase (the key enzyme for BO degradation) in the treated microbiome from Day 30 samples was from 7D-2, indicating most of the degradation was driven by the inoculated bacteria, especially for the yellow cinnamon soil. The information has been included in the revised manuscript.

On Lines 166-169, we write:

“Notably, 52%-100% of nitrilase (the key enzyme for BO degradation) in the treated microbiome was from 7D-2, indicating most of the degradation was driven by the inoculated bacteria, especially for the yellow cinnamon soil (Supplementary Fig. 6).”

On Supplementary Fig. 6, we write:

“

Supplementary Fig. 6. *The taxonomic distribution of key enzymes involved in BO degradation in treated microbiome from Day 30 samples. The nitrilase (EC 3.5.5.1 and EC 3.5.5.6) involves in the transformation from bromoxynil to DBHB, which is the key reaction for BO degradation. The result showed most of the bromoxynil degradation was driven by the inoculated bacteria (i.e. Comamonas sp. 7D-2), especially for the yellow cinnamon soil.”*

Some suggestions (first from the response to authors, then general):

- **The authors have recalculated the metric from degradation rate in percent to concentrations which I applaud. However, why is there a lot of BO at the start of the experiment for the control (where no BO is added at all), same for Figure 2 groups 2 (where bacteria are added but no BO/DBHB)? In the methods, it states that 5 mg/kg is added, whereas all experiments show around 30 to 50 mg/kg residual DBHB or BO – could the authors clarify?**

Answer: Thanks for the reviewer's comments. In response, we would like to describe the experimental design. Two phases were set, including the soil treatments (phase I) and analysis of degrading ability by treated soils (phase II). In phase I, we added 5

mg/kg BO/DBHB (close to the concentration of field application) to soils for treatments. In phase II, we collected 0.5 g of treated soils from phase I to detect the degrading ability of BO/DBHB. For analysis of degrading ability, the 0.5 g of soil was brought into fresh MM media supplemented with BO/DBHB (50 mg/L) and cultured for 10 hours. Due to the different initial soil microbiome degradation capacity of herbicides, the initial concentration of BO/DBHB (day 0) were various for different soils, from 30 to 50 mg/kg. The information has been included in the revised manuscript. Besides, we have provided detailed description of Y-axis title in legend of Fig. 2 to avoid misunderstanding.

On Lines 449-452, we write:

“Soil samples were treated by BO or DBHB with four repeats. In the BO group, the microcosm treatments were set as follows: i) BO treatment (BO): adding 2.5 mL of methanol containing 3000 mg/L of BO (the final concentration of BO was 5 mg/kg soil, close to the concentration of field application); ii)...”

On Lines 465-467, we write:

“0.5 g of soil was collected to assess the degrading ability of BO or DBHB with high-performance liquid chromatography (HPLC). The detection methods are described in the Supplementary Methods.”

On Supplementary Methods, we write:

“To detect the degrading ability of BO or DBHB by microbial community with different treatments, 0.5 g of soil was collected from different treatments and brought into fresh MM media supplemented with BO or DBHB (50 mg/L). After incubating on a shaker at 30 °C for 10 h (160 rpm), 1 mL of the liquid medium was taken from the medium to detect the residues of BO, bromoxynil or DBHB. All treatments were performed in triplicate by HPLC.”

On legend of Fig. 2, we write:

“The degradation efficiency was analyzed by detecting residues of BO (a, c, e) or DBHB (b, d, f) in MM media supplemented with BO or DBHB (50 mg/L) after degradation by 0.5 g of treated soils for 10 hours.”

• SI Figure 2: The authors reverted back to the degradation rate as a percentage. I would change this again to the remaining herbicide as a concentration or degradation efficiency in percentage. Also, would the correct control experiments here not be a single inoculation but the same BO application and no inoculation but the same BO application?

Answer: Thanks for the reviewer’s comments. According to the reviewer’s suggestion, we have changed the degradation rate into the residues of BO or bromoxynil in the revised Supplementary Fig. 2. Besides, we have included other control treatments, including single inoculation but the same BO application, and no inoculation but the

same BO application in the revised Supplementary Fig. 2.

On Supplementary Fig. 2, we write:

“

Supplementary Fig. 2. Degradation ability of bromoxynil octanoate (BO) at day 30 by different treated microbiomes. Control, only regular water spray; Single and continuous herbicide application: adding 5 mg/kg of BO once (Single) or every three days for 10 times (continuous); Single and continuous inoculation: inoculating two strains X-1 and 7D-2 at a ratio of 1:1, with a final concentration of approximately 2×10^6 CFU/g soil for each strain, once (Single) or every three days for 10 times (continuous); Single bioaugmentation (low and high): inoculating two strains X-1 and 7D-2 at a ratio of 1:1, with a final concentration of approximately 2×10^6 (low) or 2×10^8 (high) CFU/g soil for each strain, and adding 5 mg/kg of BO only once; Continuous bioaugmentation (low and high): BO (5 mg/kg) and inoculating strains [2×10^6 (low) or 2×10^8 (high) CFU/g soil] were added repeatedly every three days for 10 times.”

• **In the initial paragraph of the results, introduce what the different strains are doing metabolically (i.e., why does BO need a synergistic community to degrade?). This will also help to understand Fig. 4b.**

Answer: Thanks for the reviewer’s comments. According to the reviewer’s suggestion, we have included a paragraph describing the properties of each strain and synergistic community used in the study. We have also included the BO-degradation pathway by X-1 and 7D-2, to make their relationship easy to understand. The information has been included in the revised manuscript.

On Lines 213-222, we write:

“The strain X-1 only transformed BO to bromoxynil, but it was not capable of degrading bromoxynil (Fig. 6a). Besides, the strain X-1 could not grow using BO or bromoxynil as the sole carbon source (Fig. 6b, c). The strain 7D-2 could degrade bromoxynil, but it was unable to degrade BO into bromoxynil (Fig. 6a, f). Therefore, the strain 7D-2 could not grow using BO as the sole carbon source, but it could grow using bromoxynil as the sole carbon source (Fig. 6b, c). However, the synergistic consortium of X-1 and

7D-2 could degrade BO completely by metabolic cooperation (Fig. 6a, f), and both strains grew well using BO as the sole carbon source (Fig. 6b). Besides, both strains 7D-2 and H8 could degrade DBHB and grew using DBHB as the sole carbon source (Fig. 6d, e). However, the combination of 7D-2 and H8 did not improve the DBHB-degrading efficiency or biomass of the consortium (Fig. 6d, e).”

On Fig. 6, we write:

“

Fig. 6 Simulations and experimental validations of metabolic interactions in two-member consortia. a Comparison of the degradation ability of BO by individual strains of 7D-2 or X-1 versus in co-cultures. The concentrations of BO and bromoxynil produced during BO degradation in the medium with BO as the sole carbon source are showed. b, c Comparison of cell growth of 7D-2 and X-1 growing separately versus in co-culture. The log transformed colony-forming unit (CFU) of each strain cultured in medium with BO (b) or bromoxynil (c) as the sole carbon source is showed. d, e The residues of DBHB (d) and cell biomass (e) during DBHB degradation by individual strains of 7D-2 and H8 or co-cultures. f BO degradation pathway by strains X-1, 7D-2 and H8. g Prediction of exchanged metabolites between strains 7D-2 and X-1....”

• The functional genes in Fig. 4 come from a great diversity of species/taxonomic classification. Is there a way to quantify how many actual cells in the day 30

samples come from the different taxons? And see how many are NOT from the inoculated proteobacteria? This would be interesting to see how much of the degradation is driven by the enriched, resident community.

Answer: Thanks for the reviewer's comments. We have included the taxonomic distribution of the key enzymes involved in BO degradation in the revised manuscript (Supplementary Fig. 6). The results showed that 52%-100% of nitrilase (the key enzyme for BO degradation) in the treated microbiome from Day 30 samples was from 7D-2, indicating most of the degradation was driven by the inoculated bacteria, especially for the yellow cinnamon soil. The information has been included in the revised manuscript. Please see our response to Comment 1 above.

• I would include a definition of what the authors mean by “domestication of microbiomes” in the introduction. This is important to understand the remaining manuscript. For macro-flora and fauna, there are three typical pathways of domestication, but a definition of the actual process would be good since it is used a lot.

Answer: We thank very much for the reviewer's comments. Based on the reviewer's suggestions, we compared the meaning of domestication for macro-flora/fauna and “domestication of microbiomes” used here. The three typical pathways of domestication for macro-flora and fauna, including the commensal, prey, and directed pathways, are different to the “domestication of microbiomes”. The “domestication of microbiomes” here was intended to refer to “reassembly of microbial community after treatments of herbicide application and/or inoculation of herbicide-degrader”, and “domestication of microbiomes” here was just a description of reassembly process of microbial communities. Therefore, the “domestication” was improper used for description the process of reassembly process of microbial communities in the manuscript. To avoid misunderstanding, we have removed the expression of “domestication of microbiomes” in the revised manuscript. Please see our response to Comment 1 above.

• Generally, I would add one paragraph to the discussion about the high inoculation rate and how this drives the community succession etc. in the initial top-down approach. I agree with the authors that this does not implicate the remaining aspect of the manuscript (bottom-up community design and validation) but could raise questions from the readership.

Answer: Thanks for the reviewer's comments. According to the reviewer's suggestions, we have added a paragraph to discuss about the high inoculation rate and the influences of different concentrations of inocula on the community succession.

On Lines 374-382, we write:

“We also showed that treatments with high-dose of inoculation had more remarkable influence on bacterial community compared to those with low-dose inoculation, indicating the effects of inoculation on natural microbiomes might be dose-dependent. Actually, for most bioaugmentation treatments, a final concentration of 10^8 CFU/g soil is usually used, and the high-dose of inoculation causes significant influences on natural microbiomes⁶⁴⁻⁶⁶. With decreased concentration (10^7 CFU/g soil), the inoculation causes temporary impacts⁶⁷, and the inoculation is not the main factor influencing the bacterial community structure compared to the herbicide application⁶⁸. The low-dose of inoculation (10^6 CFU/g soil) shows a relatively weak impact on the soil bacterial community⁶⁹.”

Reference

64. Liu, Y., Hou, Q., Liu, W., Meng, Y. & Wang, G. Dynamic changes of bacterial community under bioremediation with *Sphingobium* sp. LY-6 in buprofezin-contaminated Soil. *Bioprocess. Biosyst. Eng.* **38**, 1485–1493 (2015).
65. Wu, M. *et al.* Bacterial community shift and hydrocarbon transformation during bioremediation of short-term petroleum-contaminated soil. *Environ. pollut.* **223**, 657–664 (2017).
66. Liu, L. H. *et al.* Endophytic Phthalate-degrading *Bacillus subtilis* N-1-*gfp* colonizing in soil-crop system shifted indigenous bacterial community to remove di-n-butyl phthalate. *J. Hazard. Mater.* **449**, 130993 (2023).
67. Pacwa-Płociniczak, M., Czapla, J., Płociniczak, T. & Piotrowska-Seget, Z. The effect of bioaugmentation of petroleum-contaminated soil with *Rhodococcus erythropolis* strains on removal of petroleum from soil. *Ecotoxicol. Environ. Saf.* **169**, 615–622 (2019).
68. Chen, S. *et al.* Soil bacterial community dynamics following bioaugmentation with *Paenarthrobacter* sp. W11 in atrazine-contaminated soil. *Chemosphere* **282**, 130976 (2021).
69. Dai, Y., Li, N., Zhao, Q. & Xie, S. Bioremediation using *Novosphingobium* strain DY4 for 2, 4-dichlorophenoxyacetic acid-contaminated soil and impact on microbial community structure. *Biodegradation* **26**, 161–170 (2015).

Reviewer #1 (Remarks to the Author):

I thank the authors for their detailed response. However, my main concerns with the manuscript largely remain.

1. The proposed interactions are still not adequately explained from a physiological perspective. I realized that the authors predicted metabolite exchanges are shown in Fig 6 (e.g. X-1 consumes BO, produces octanoate while secreting hypoxanthine, glucosamine, bromoxynil, and proline; 7D-2 consumes secreted metabolites while providing xanthine, mannose, NH₄⁺, and glutamate to X-1 and producing HBr). Can the authors provide simple stoichiometric and energetic calculations to support this mutualism (for each organism and then the overall reaction)? Also, if the culture was fed only 50mg/L of BO, the mass balance around products (Fig 6h,j – for example, ~100 mg/L hypoxanthine alone) doesn't add up. Can the authors please explain this discrepancy?

2. Relatedly, I don't understand what the transcriptomic analysis confirmed. No actual results besides a Venn diagram and general descriptions are provided. If the author could provide a clear metabolic explanation together with a detailed explanation of how the gene expression analysis supports it (e.g., differential expression of specific genes related to the proposed interaction as well as the dataset) this would be very helpful.

3. Looking at Figure 6 I still see no clear comparison of what the model is predicting and what was experimentally measured. Can the authors make this clear in the figure and also include more discussion on where the model predictions were accurate (and where they weren't)? If the model prediction fits the data very well it should be relatively straightforward to propose a clear and testable explanation of the metabolite exchanges.

Reviewer #2 (Remarks to the Author):

[No further comments for authors]

We thank the editorial team and the referees for the positive feedback and the constructive report for our manuscript entitled "Engineering natural microbiomes toward enhanced bioremediation by microbiome modeling" (NCOMMS-22-11760B). In the revised version, we now address all the concerns raised by the referees. Please find below a point-by-point response providing a full account of our modifications (bold font style represents comments of reviewers, our response follows, and revised text is brought in italic where relevant).

REVIEWER COMMENTS

Reviewer #1 (Remarks to the Author):

I thank the authors for their detailed response. However, my main concerns with the manuscript largely remain.

Answer: Thanks for the reviewer's comments. In response to the main concerns raised, we have revised the manuscript to include a detailed physiological explanation and validation for the proposed interactions (Fig. 6, Supplementary Fig. 15, Supplementary Table 6, please see our detailed responses below). We believe that these explanations and validation adequately address the concerns raised by the reviewer.

Furthermore, we have taken note of the lack of validation and explanation of predicted metabolic interactions in previous studies (e.g., Schäfer et al., *Science*, 2023; Kost et al., *Nature Microbiology*, 2023; Dicenzo et al., *Nature Communications*, 2020; Ryback et al., *The ISME Journal*, 2022) and acknowledge the importance of addressing this issue. In the revised manuscript, we have made a process in the area of experimental validation and explanation of metabolic interactions obtained by community modeling.

References

- Schäfer M, et al. Metabolic interaction models recapitulate leaf microbiota ecology. *Science*, 2023, 381: 6653
- Kost C, et al. Metabolic exchanges are ubiquitous in natural microbial communities. *Nature Microbiology*, 2023, 8: 2244–2252.
- Dicenzo GC, et al. Genome-scale metabolic reconstruction of the symbiosis between a leguminous plant and a nitrogen-fixing bacterium. *Nature Communications*, 2020, 11: 2574.
- Ryback B, et al. Metabolic adaptation to vitamin auxotrophy by leaf-associated bacteria. *The ISME Journal*, 2022, 16: 2712-2724.

1. The proposed interactions are still not adequately explained from a physiological perspective.

Answer: Thanks for the reviewer's comments. In the revised manuscript, we have included a detailed description of the metabolic interactions and exchanges between the species together with the supportive experimental validation. We employed a combination of transcriptomics, metabolomics, and thermodynamic analysis to

elucidate and validate the physiological basis of these metabolic interactions. Briefly, to illustrate the metabolic pathways involved in the interactions, we have included a detailed metabolic network that being active in modeling. Furthermore, through transcriptomic analysis, we have demonstrated that both strains possess the necessary enzymes to catalyze these metabolic reactions, and we observed up-regulated expression of these genes in co-culture compared to single-culture. Moreover, our metabolomic analysis successfully detected the predicted secreted metabolites in the extracellular space (i.e., culture medium). Notably, no secreted metabolites were detected in the initial medium, and their detections in co-cultures validate the secretion of these metabolites by the strains. Additionally, we performed thermodynamic calculations for each metabolic reaction, providing further evidence for their occurrence within the cells during interactions. This information has been included in the revised manuscript (Lines 238-243, 249-259, Supplementary Fig. 13, and Supplementary Table 3). We sincerely appreciate your constructive feedback, which has significantly improved the quality of the manuscript.

On Lines 238-243, we wrote:

“The full map of predicted metabolic interactions between 7D-2&X-1 is detailed in Supplementary Fig. 13, describing the metabolic routes leading to production and consumption of the exchange metabolites specified in Fig 6b. To verify the predicted exchange fluxes, we used liquid chromatography–mass spectrometry (LC–MS) to detect the exchanged metabolites in co-cultures of these two strains (Supplementary Fig. 13). All the exchanged metabolites were successfully detected in LC–MS...”

On Lines 251-258, we wrote:

“To further validate the validity of the predicted metabolic pathways underlying the mutualism between strains X-1 and 7D-2, gene expression profiling of the two strains was compared in single-cultures versus co-cultures (Supplementary Fig. 13, Supplementary Table 3). The expression levels of most genes encoding enzymes that participate in the synthesis of the secreted metabolites were up-regulated in co-culture compared to single-culture. These results indicate that the strains 7D-2 and X-1 possess all the enzymes required for the metabolic interactions and genes coding these enzymes were expressed during metabolic interactions. Therefore, the results of transcriptomic profiling confirmed the predicted metabolic interaction.”

On Supplementary Fig. 13 and Supplementary Table 3, we write:

“|

Supplementary Fig. 13. Simplified metabolic network depicting metabolic interactions between strains X-1 and 7D-2. **a, c** The predicted reactions active in the strains X-1 (a) and 7D-2 (c). The codes starting with “rxn” represent metabolic reactions. Supplementary Table 3 provides detailed information on these reactions, including reactants, products, stoichiometry, direction, ΔG , and genes coding the enzymes required for the metabolic interactions. The heatmap shows gene expression levels of strains X-1 and 7D-2 in single-culture and co-culture, and each treatment had three replicates. Red indicates increased transcript abundance, while blue indicates decreased transcript abundance. The red arrows indicate up-regulated genes in co-cultures compared to single-cultures. **b** Prediction and experimental validation of exchanged metabolites between strains 7D-2 and X-1. The bar graph displays the concentration of each exchanged metabolite detected by LC–MS in the medium of co-cultures (BO&X-1&7D-2). Medium with BO as the only carbon and nitrogen source

was used for both simulation and experimental validation. The medium without inoculation of any strains (BO) was used as control. Notably, these exchanged metabolites were not initially present in the medium, and their detection validates the secretion of these metabolites by the strains.

Supplementary Table 3. Detailed information about the metabolic reactions related to the metabolic interactions between strain X-1 and 7D-2. The standard Gibbs energy of reaction for each reaction (Δ_rG^0) were calculated at pH 7.0, ionic strength of 0.25 M and temperature of 298.15 K by ModelSeed (<https://modelseed.org/biochem/reactions>) and eQuilibrator (<http://equilibrator.weizmann.ac.il>).

Reaction_ID	Δ_rG^0 (kcal/mol)	EC_ID	KEGG_ID	Formula	Gene_ID
X-1					
rxn01200	0.35	2.2.1.1	R01641	Glyceraldehyde3-phosphate + Sedoheptulose7-phosphate <=> ribose-5-phosphate + D-Xylulose5-phosphate	assembly_01277
rxn00770	0.19	2.7.6.1	R01049	ATP + ribose-5-phosphate => AMP + H+ + PRPP	assembly_03051
rxn00790	-12.98	2.4.2.14	R01072	PPi + L-Glutamate + H+ + 5-Phosphoribosylamine <=> H2O + L-Glutamine + PRPP	assembly_02969
rxn02895	-6.35	6.3.4.13	R04144	ATP + Glycine + 5-Phosphoribosylamine => ADP + Phosphate + H+ + GAR	assembly_00768
rxn03004	-6.68	2.1.2.2	R04325	10-Formyltetrahydrofolate + GAR => H+ + Tetrahydrofolate + N-Formyl-GAR	assembly_01536
rxn03005	-7.29	2.1.2.2	R04326	(2) H+ + Tetrahydrofolate + N-Formyl-GAR <=> H2O + 5-10-Methenyltetrahydrofolate + GAR	assembly_01536
rxn03084	-13.45	6.3.5.3	R04463	H2O + ATP + L-Glutamine + N-Formyl-GAR => ADP + Phosphate + L-Glutamate + H+ + 5'-Phosphoribosylformylglycinamide	assembly_03367
rxn02937	-20.66	6.3.3.1	R04208	ATP + 5'-Phosphoribosylformylglycinamide <=> ADP + Phosphate + H+ + AIR	assembly_01537
rxn05114	-13.69	6.3.4.18	R07404	ATP + H2CO3 + AIR => ADP + Phosphate + (2) H+ + 5-phosphoribosyl-5-carboxyaminoimidazole	assembly_02910
rxn05115	-7.71	5.4.99.1 8	R07405	H+ + 5-phosphoribosyl-5-carboxyaminoimidazole <=> 5'-Phosphoribosyl-4-carboxy-5-aminoimidazole	assembly_02911
...(the complete table is shown in Supplementary Information)					

”

I realized that the authors predicted metabolite exchanges are shown in Fig 6 (e.g. X-1 consumes BO, produces octanoate while secreting hypoxanthine, glucosamine, bromoxynil, and proline; 7D-2 consumes secreted metabolites while providing xanthine, mannose, NH₄⁺, and glutamate to X-1 and producing HBr). Can the authors provide simple stoichiometric and energetic calculations to support this mutualism (for each organism and then the overall reaction)?

Answer: Thanks for the reviewer's comments. The process from BO degradation to exchanged metabolite production is not a single metabolic reaction, as BO undergoes a series of metabolic transformations within the cells. The full set of active metabolic reactions is included in Response Fig. 1 below, which shows the complexity of the metabolism. Notably, most of these metabolites are used for cellular growth and metabolism. It is very challenging to determine the proportion of these metabolites utilized for growth, energy metabolism and exchanges as there are multiple routes enabling cellular plasticity and dynamics. Hence, it is virtually impossible to calculate the stoichiometric relations between BO (the input metabolites) and secreted metabolites. Furthermore, the process of metabolite exchanging between strains X-1 and 7D-2 occurred continuously and dynamically. In the experimental validation, we chose a specific time point (4 hours after co-culture) for sampling and measurement. Therefore, the amount of metabolites detected at this time point cannot reflect the total quantity of exchanged metabolites but only indicates that exchanged metabolites were secreted into the culture medium by the strains (as the initial medium did not contain these metabolites). Thus, the results of metabolite detection here should be considered as qualitative validation of metabolite exchanging, but they cannot provide an accurate stoichiometric coefficient from BO to exchanged metabolites. Thirdly, we used community modeling to predict the flux of metabolic exchange. It is important to note that the predicted metabolic flux is not unique and alternative solutions may exist. Therefore, the predictions cannot provide an accurate stoichiometric coefficient from BO to the exchanged metabolites either. More generally, this limitation is consistent with other works where exchange metabolites were predicted and confirmed qualitatively (Succurro et al., 2017; Venturelli et al., 2018; Xu et al., 2019; Goyal et al., 2021;).

As mentioned above, metabolic interactions are not a well-defined metabolic reaction, and thus, clear stoichiometric relations cannot be established, making energy calculations unfeasible either. Instead, we employed the tool ModelSeed (<https://modelseed.org/biochem/reactions>) and eQuilibrator (<http://equilibrator.weizmann.ac.il>) to estimate the Gibbs free energies for all involved reactions (Supplementary Fig. 15, Supplementary Table 12). The $\Delta_r G^\circ$ of most reactions was < 0 , and we also detected some reactions with $\Delta_r G^\circ > 0$. Notably, for the strain X-1, the $\sum \Delta_r G^\circ = -100.12$ (kcal/mol), and for the strain 7D-2, $\sum \Delta_r G^\circ = -160.51$ (kcal/mol). Besides, the concentration of BO in cells was far larger than that of exchanged metabolites. Therefore, these reactions are feasible to occur inside cells, which are also supported by the increased expressions of related genes (Supplementary Fig. 15). Actually, the secretion of these substances has also been reported, such as xanthine/hypoxanthine (Rinas et al., 1995; Brauer et al., 2006; Link et al., 2015),

mannose (Hu et al., 2016), glutamate (Chao et al., 1959; Musílková et al., 1966; Zareian et al., 2012; Ghazanfari et al., 2023), glucosamine (Liu et al., 2013), and proline (Rancourt et al., 1984; Krämer, 1994)

References

Succurro A, et al. A diverse community to study communities: integration of experiments and mathematical models to study microbial consortia. *Journal of Bacteriology*, 2017, 199: 10.1128/jb.00865-16.

Venturelli OS, et al. Deciphering microbial interactions in synthetic human gut microbiome communities. *Molecular Systems Biology*, 2018, 14: e8157.

Xu X, et al. Modeling microbial communities from atrazine contaminated soils promotes the development of biostimulation solutions. *The ISME Journal*, 2019, 13: 494-508.

Goyal A, et al. Ecology-guided prediction of cross-feeding interactions in the human gut microbiome. *Nature Communications*, 2021, 12: 1335.

Rinas U, et al. Entry of *Escherichia coli* into stationary phase is indicated by endogenous and exogenous accumulation of nucleobases. *Applied and Environmental Microbiology*, 1995, 61: 4147-4151.

Brauer MJ, et al. Conservation of the metabolomic response to starvation across two divergent microbes. *Proceedings of the National Academy of Sciences*. 2006, 103: 19302-19307.

Link H, et al. Real-time metabolome profiling of the metabolic switch between starvation and growth. *Nature Methods*, 2015, 12:1091-1097.

Hu X, et al. D-Mannose: Properties, production, and applications: An overview. *Comprehensive Reviews in Food Science and Food Safety*, 2016, 15: 773-785.

Chao KC & Foster JW. A glutamic acid-producing *Bacillus*. *Journal of Bacteriology*, 1959, 77: 715-725.

Musílková M, et al. Isolation of a glutamate-producing bacterial strain. *Folia Microbiologica*, 1966, 11: 301-303.

Zareian M, et al. A glutamic acid-producing lactic acid bacteria isolated from Malaysian fermented foods. *International Journal of Molecular Sciences*, 2012, 13: 5482-5497.

Ghazanfari N, et al. Optimization of fermentation culture medium containing food waste for l-glutamate production using native lactic acid bacteria and comparison with industrial strain. *LWT*, 2023, 184: 114871.

Liu L, et al. Microbial production of glucosamine and N-acetylglucosamine: advances and perspectives. *Applied Microbiology and Biotechnology*, 2013, 97: 6149-6158.

Rancourt DE, et al. Proline excretion by *Escherichia coli* K12. *Biotechnology and Bioengineering*, 1984, 26: 74-80.

Krämer R. Secretion of amino acids by bacteria: physiology and mechanism. *FEMS Microbiology Reviews*, 1994, 13: 75-93.

Response Fig. 1. Overview of the metabolic network and simulated flux distribution of different reactions through modeling. a. Strain 7D-2. b. Strain X-1. Ellipses represent metabolites, while rectangles represent metabolic reactions. The blue or green lines indicate the direction of the reactions and the magnitude of flux, with thicker lines indicating higher flux. To simplify the metabolic network, only metabolic reactions

with flux > 0.1 are shown, while small molecules and cofactors such as H⁺, H₂O, O₂, CO₂, ATP, NAD, NADH, NADPH, NADP, CoA, Pi, PPi, and NH₄⁺ have been eliminated from the networks. The detailed information of metabolites and reactions are provided in Supplementary Models of the manuscript.

Also, if the culture was fed only 50mg/L of BO, the mass balance around products (Fig 6h,j – for example, ~100 mg/L hypoxanthine alone) doesn't add up. Can the authors please explain this discrepancy?

Answer: We are grateful for the reviewer for pointing this out before publication. Compared to the added carbon sources (BO here), we anticipated that the concentration of exchanged metabolites would be very low before performing the experiments. Therefore, when detecting the exchanged metabolites, we concentrated the culture medium (200-fold). Therefore, the ~100 mg/L hypoxanthine here is equivalent to 0.5 mg/L of the original concentration. To avoid any misunderstanding, we have changed the concentration of all substances according to the initial concentration (Supplementary Fig. 11 and Supplementary Fig. 13).

2. Relatedly, I don't understand what the transcriptomic analysis confirmed. No actual results besides a Venn diagram and general descriptions are provided. If the author could provide a clear metabolic explanation together with a detailed explanation of how the gene expression analysis supports it (e.g., differential expression of specific genes related to the proposed interaction as well as the dataset) this would be very helpful.

Answer: Thanks for the reviewer's comments. According to the reviewer's suggestion, we have included a clear metabolic pathway with detailed gene expressions related to each metabolic reaction (Supplementary Fig. 13 and Supplementary Table 3). We showed the clear metabolic pathway from BO to the exchanged metabolites, which are directly related to the metabolic interactions, thus providing a clear metabolic explanation of the interactions. We also compared the expression of genes related to these reactions between single-culture and co-culture, which showed increased gene expression levels in co-culture compared to single-culture. The gene expression pattern indicated that these reactions occurred in both cells, thus confirmed the proposed interaction. Besides, to make the transcriptomic analysis concise and focused on the metabolic interactions, we have deleted the Venn diagram and non-related treatments in the revised manuscript.

3. Looking at Figure 6 I still see no clear comparison of what the model is predicting and what was experimentally measured. Can the authors make this clear in the figure and also include more discussion on where the model predictions were accurate (and where they weren't)? If the model prediction fits the data very well it should be relatively straightforward to propose a clear and testable explanation of the metabolite exchanges.

Answer: Thanks for the reviewer's comments. We have revised Fig. 6 to show a clear comparison between simulations and experimental validation. To make the figure

concise and focus, we only showed the interactions between X-1 and 7D-2 in the revised Fig. 6, and the interactions between H8 and 7D-2 is showed in Supplementary Fig. 11. In the revised Fig. 6, a total of 10 testable simulations were predicted based on modeling, which were experimental validation (Fig. 6c-g, Supplementary Fig. 13). For examples, the predictions of no growth of X-1 or 7D-2 for single-culture but growth of both strains for co-cultured were verified by the growth experiment (Fig. 6c); the predictions of no BRO assimilation by X-1 were verified by the growth experiment with BRO as the carbon source (Fig. 6d); the predictions of BO degradation by X-1 and no BO degradation by 7D-2 were verified by the BO-degrading experiment (Fig. 6e, f); the predictions of secretions of exchanged metabolites were verified by detecting the exchanged metabolites in medium of co-cultures (Supplementary Fig. 13). All these predictions were supported by experimental validations. The information has been included in the revised manuscript (Lines 213-227). Besides, an additional discussion about the accurate of model predictions has been included (Lines 428-435).

On Lines 213-227, we write:

“To determine detailed metabolic interactions between strains in the inoculated consortium, including the synergistic (7D-2&X-1) and competitive consortia (7D-2&H8), a two-strain community model was constructed and analyzed (Fig. 6, Supplementary Fig. 11). By simulation, the strain X-1 only transformed BO to bromoxynil, thus could not grow using BO or bromoxynil as the sole carbon source; the strain 7D-2 could degrade bromoxynil, but it was unable to degrade BO into bromoxynil (Fig. 6a). Therefore, the strain 7D-2 could not grow using BO as the sole carbon source, but it could grow using bromoxynil as the sole carbon source (Fig. 6b, Supplementary Fig. 12). However, the synergistic consortium of X-1 and 7D-2 could degrade BO completely by metabolic cooperation, and both strains grew well using BO as the sole carbon source (Fig. 6b, Supplementary Fig. 12). These predictions were supported by experimental validations (Fig. 6c-g). For example, the predictions of no growth of X-1 or 7D-2 for single-culture but growth of both strains for co-cultured were verified by the growth experiment (Fig. 6c). Both strains 7D-2 and H8 could degrade DBHB and grew using DBHB as the sole carbon source (Supplementary Fig. 11). However, the combination of 7D-2 and H8 did not improve the DBHB-degrading efficiency or biomass of the consortium (Supplementary Fig. 11).”

On Lines 428-435, we write:

“Exploring and validating metabolic interactions is a challenging task. We developed a modeling tool to predict metabolic interactions and provided a series of testable predictions for experimental validation. These include: 1) the predicted exchanged metabolites could be detected in the medium of co-culture; 2) the strain growth could be improved by the exchanged metabolites in single-culture; 3) SIP experiments could be employed to validate the assimilation of the exchanged metabolites; 4) transcriptome profiling could be used to test the expression of genes required for metabolic reactions. The modeling tool combined with the validation strategy could greatly facilitate the application of metabolic modeling.”

On Fig.6 (Lines 936-947) and Supplementary Fig. 11, we write:

“

Fig. 6 | Simulations and experimental validations of metabolic interactions in two-member consortia. a, b Predicted metabolic interactions between strains 7D-2 and X-1 in single-culture (a) and co-culture (b) by community modeling. Ten testable predictions were provided for experimental verification of the simulations. The letters in parentheses followed the predictions refers to the panels (c-h) supporting the corresponding predictions. BRO, bromoxynil. **c, d** Comparison of cell growth of 7D-2 and X-1 growing separately versus in co-culture. The log transformed colony-forming unit (CFU) of each strain cultured in medium with BO (b) or bromoxynil (c) as the sole carbon source is showed. **e-g** The degradation ability of BO by X-1 (e), 7D-2 (f) and co-cultures (g). The concentrations of BO and bromoxynil produced during BO degradation in the medium with BO as the sole carbon source are showed. **h** Identification of the predicted exchanged metabolites via LC-MS in a co-culture of strains 7D-2 and X-1. Medium with BO as the only carbon and nitrogen source was used for both simulation and experimental validation.

Supplementary Fig. 11. Simulations and experimental validations of metabolic interactions between strain 7D-2 and H8. *a, b* Predicted metabolic interactions between strains 7D-2 and H8 in single-culture (*a*) and co-culture (*b*) by community modeling. Five testable predictions were provided for experimental validation of the simulations. The letters in parentheses followed the predictions refers to the panels (*c-e*) supporting the corresponding predictions. *c* The cell growth of 7D-2 and H8 growing separately versus in co-culture. The log transformed colony-forming unit (CFU) of each strain cultured in medium with DBHB as the sole carbon source is shown. *d* The residues of DBHB during DBHB degradation by individual strains of 7D-2 and H8 or co-cultures. *e* Identification of the predicted exchanged metabolites (EMs) via LC-MS in a co-culture of strains 7D-2 and H8. Medium with DBHB and NH₄⁺ as the carbon and nitrogen sources was used for both simulation and experimental validation.

”

Reviewer #1 (Remarks to the Author):

I thank the reviews for their detailed response. Most of my concerns are addressed. My remaining comment relates to the overall thermodynamics of the proposed interactions. While supplementary table 3 provides standard free energy values for individual reactions, there is no supporting calculation on whether the overall stoichiometry of net energy and growth reactions are thermodynamically feasible. Such calculations are routine in environmental microbiology/biotechnology (see Rittmann and McCarty Environmental Biotech textbook or equivalent). The author's claim that it is impossible to track all the reactions and metabolites, but this is simply not true (or even necessary) – the stoichiometry of extracellular substrate uptake and metabolite production should come directly out of the FBA solution which can be used for the calculations (For example, a stoichiometric equation could be written for the interactions shown in Fig 6a and 6b that form the basis of the overall bioenergetic calculations). I think such a calculation would be a very useful addition to determine whether the proposed interactions violate basic laws of physics.

REVIEWER COMMENTS

Reviewer #1 (Remarks to the Author):

I thank the reviews for their detailed response. Most of my concerns are addressed. My remaining comment relates to the overall thermodynamics of the proposed interactions. While supplementary table 3 provides standard free energy values for individual reactions, there is no supporting calculation on whether the overall stoichiometry of net energy and growth reactions are thermodynamically feasible. Such calculations are routine in environmental microbiology/biotechnology (see Rittmann and McCarty Environmental Biotech textbook or equivalent). The author's claim that it is impossible to track all the reactions and metabolites, but this is simply not true (or even necessary) – the stoichiometry of extracellular substrate uptake and metabolite production should come directly out of the FBA solution which can be used for the calculations (For example, a stoichiometric equation could be written for the interactions shown in Fig 6a and 6b that form the basis of the overall bioenergetic calculations). I think such a calculation would be a very useful addition to determine whether the proposed interactions violate basic laws of physics.

Answer: Thanks for the reviewer's very helpful comments. According to the reviewer's suggestions, we have included the thermodynamic calculations showing that the overall stoichiometry of net energy and growth reactions are thermodynamically feasible (Supplementary Table 4, Supplementary Table 5). These calculations were conducted individually for each organism, X-1 and 7D-2, as well as their combination. In brief, we firstly used the FBA solution to determine the stoichiometry of metabolite production and uptake. Subsequently, this stoichiometry was used to derive the reactions for energy generation and cell synthesis, as well as their overall reactions following the methods provided in Rittmann & McCarty Environmental Biotechnology: Principles and Applications (Second Edition). Finally, based on these reactions, the overall energy generation and energy requirement for cell synthesis were calculated, confirming the thermodynamic feasibility of the proposed interaction (Supplementary Table 4, Supplementary Table 5).

On Lines 241-243, we wrote:

“The thermodynamic analysis of the predicted metabolic interaction confirmed its thermodynamic feasibility (Supplementary Table 4, Supplementary Table 5).”